# Is Noise Conditioning Necessary? A Unified Theory of Unconditional Graph Diffusion Models

**Jipeng Li**
Department of Electrical and Computer Engineering
University of California, Davis
Davis, CA 95616
jipli@ucdavis.edu

**Yanning Shen**
Department of Electrical Engineering and Computer Science
University of California, Irvine
Irvine, CA 92697
yannings@uci.edu

## Abstract

Explicit noise-level conditioning is widely regarded as essential for the effective operation of Graph Diffusion Models (GDMs). In this work, we challenge this assumption by investigating whether denoisers can implicitly infer noise levels directly from corrupted graph structures, potentially eliminating the need for explicit noise conditioning. To this end, we develop a theoretical framework centered on Bernoulli edge-flip corruptions and extend it to encompass more complex scenarios involving coupled structure-attribute noise. Extensive empirical evaluations on both synthetic and real-world graph datasets, using models such as GDSS and DiGress, provide strong support for our theoretical findings. Notably, unconditional GDMs achieve performance comparable or superior to their conditioned counterparts, while also offering reductions in parameters $(4 - 6\%)$ and computation time $(8 - 10\%)$. Our results suggest that the high-dimensional nature of graph data itself often encodes sufficient information for the denoising process, opening avenues for simpler, more efficient GDM architectures.

## 1 Introduction

Diffusion models have demonstrated strong performance across a range of generative tasks, including image synthesis, molecular design, and graph-based combinatorial optimization [1–7]. A central assumption in these models is the requirement for the denoiser to be explicitly conditioned on the noise level (or timestep) [8, 9]. However, recent work on continuous data suggests that high-capacity denoisers can implicitly infer the noise scale directly from the corrupted inputs, potentially eliminating the need for explicit conditioning [10–12].

Extending this conclusion to graph diffusion models (GDMs) is non-trivial, owing to the inherently discrete and structured nature of graphs. The Gaussian noise commonly used in diffusion models for continuous data is ill-suited for graphs as it tends to destroy their essential structural properties [13]. To address this, modern GDMs employ discrete or structured corruption processes, such as Bernoulli edge flips [7, 14], categorical rewiring [15], or Poisson jumps [16], to ensure that intermediate graph states remain valid. Unlike continuous domains where Gaussian perturbations are common, graphs are typically corrupted by discrete, structure-preserving processes. The resulting noisy adjacency $\tilde{A}_t$

39th Conference on Neural Information Processing Systems (NeurIPS 2025).

already carries information about the corruption scale through simple statistics, and the number of potential edges grows as $M = \binom{n}{2}$. Hence, the observation at each step may be informative enough for a denoiser to infer the noise level without an explicit time input. These considerations lead to our central question: ***Is explicit noise-level conditioning truly necessary for GDMs?***

In this work, we address this question through a unified theoretical and empirical investigation. We introduce a novel theoretical framework featuring posterior concentration and error propagation bounds—**Edge-Flip Posterior Concentration (EFPC), Edge-Target Deviation Bound (ETDB), and Multi-Step Denoising Error Propagation (MDEP)**—to rigorously characterize when explicit noise-level conditioning in GDMs can be safely omitted. For instance, with Bernoulli edge flips, we prove noise scale posterior variance shrinks optimally at $O(M^{-1})$, and omitting timesteps yields a reconstruction error bounded by $O(T/M)$ over $T$ steps. Our analysis generalizes to other corruption processes, including Poisson, Beta, multinomial, and jointly structured feature-graph noise.

Extensive experiments on synthetic and real-world datasets corroborate our theory, validating that explicit noise inputs are often dispensable. Unconditional variants of state-of-the-art GDMs, GDSS [17] and DiGress [7], match or surpass their conditioned counterparts in quality, while improving efficiency (4–6% fewer parameters, 8–10% faster per-epoch runtime). Open source implementation is available for regenerating the results. Overall, this work revisits a foundational assumption in diffusion modeling, demonstrating that noise-unconditional GDMs can be as accurate, more efficient, and conceptually simpler, paving the way for new GDM designs.

## 2 Related Work

**Timestep Conditioning in Graph Diffusion Models.** Denoising diffusion models reconstruct data by reversing a noise process through iterative denoising steps conditioned on the timestep $t$ [8, 9, 18]. GDMs, such as DiGress with discrete categorical Markov processes [7], GDSS with coupled stochastic differential equations (SDEs) [17], and EDM with equivariant diffusion for molecules [19], explicitly condition the denoiser on the noise level. This explicit conditioning adds complexity and training cost, prompting questions about its necessity for graph data [11, 12].

**Blind Denoising.** Blind denoising infers noise implicitly and is effective in continuous domains—see Noise2Self [20] and recent diffusion work [10, 11]. Graphs pose a tougher case: these methods assume continuous Gaussian noise, whereas GDMs use discrete or structured corruptions to keep topology intact and thus rely on fixed noise schedules with explicit timesteps. This gap motivates unconditional graph denoisers that can absorb discrete, structured noise without extra inputs.

**Noise Types in GDMs.** GDMs employ tailored noise processes. Early works injected Gaussian noise into adjacency and features [17, 21], but this disrupts sparsity. Discrete categorical noise [13], as in DiGress [7], better preserves graph structure. Latent-space approaches, notably hyperbolic diffusion, capture hierarchy efficiently [22]. Recent variants include *categorical flips* [7], *permutation-invariant multinomial noise* [21], *Bernoulli edge flips* [14], *discrete-time Poisson processes* [16], and *Beta-distributed noise* [23]. Yet all methods still pass an explicit timestep input. We argue that denoisers can implicitly infer structured noise levels during generation.

## 3 Problem Setting and Preliminaries

Graph Diffusion Models (GDMs) generate realistic graphs via a *forward process* that iteratively corrupts an input graph with noise, followed by a *reverse process* that learns to undo this corruption, sampling new graphs from noise. We examine whether the reverse step truly needs explicit knowledge of the current scalar noise level at each individual denoising timestep during generation.

**Graph Definition.** We consider an undirected graph $G = (V, E)$, where $V$ is the set of $n = |V|$ nodes and $E$ is the set of edges. The initial, clean graph structure is represented by its adjacency matrix $A_0 \in \{0, 1\}^{n \times n}$. Node features, if present, are denoted by $X_0 \in \mathbb{R}^{n \times d_f}$, where $d_f$ is the feature dimensionality. Our theoretical development primarily focuses on structural diffusion, with extensions to coupled structure and feature corruption discussed in Section 5. We use $\tilde{A}_t$ and $\tilde{X}_t$ for the noisy adjacency and feature matrices at diffusion step $t$, respectively, with $\tilde{A}_0 = A_0$ and $\tilde{X}_0 = X_0$. The total number of diffusion steps is $T$.

**Forward Process.**    The forward process introduces sequences of noisy graph states $(\tilde{A}_t)_{t=0}^T$ and noisy feature states $(\tilde{X}_t)_{t=0}^T$ (if features are considered). Graph structure is commonly corrupted using the Bernoulli edge-flipping model. At each step $t$, every potential edge $e$ is independently perturbed. Given the state of an edge $\tilde{A}_{t-1}(e)$ at step $t-1$, its state at step $t$, $\tilde{A}_t(e)$, is drawn from:

$$\tilde{A}_t(e)|\tilde{A}_{t-1}(e) \sim \text{Bernoulli}\left((1-\beta_t)\tilde{A}_{t-1}(e) + \beta_t[1-\tilde{A}_{t-1}(e)]\right).$$

Here, $\beta_t \in [0, \frac{1}{2}]$ is the per-step flip probability for edge $e$. The sequence $\{\beta_t\}_{t=1}^T$ forms a noise schedule that typically increases with $t$. Under this channel, $\tilde{A}_T$ approaches an Erdős–Rényi graph with $p = \frac{1}{2}$ in the marginal sense; the exact recursion and a sufficient convergence condition are given in Appendix J. When node features $\tilde{X}_t$ are present, they are corrupted in parallel (e.g., Gaussian or structured noise). In Section 5, we introduce a coupled model where structural and feature perturbations are correlated; further details appear in Appendix J.

**Reverse Denoising Process and Training Objective.**    The fundamental aspect of the GDM is the reverse denoising process, which learns to reverse the forward corruption. This involves a denoising function $f_\theta$, often a Graph Neural Network, parameterized by $\theta$. It takes a noisy graph $\tilde{A}_t$ (and potentially $\tilde{X}_t$) as input. Conventionally, $f_\theta$ is explicitly conditioned on the noise level or timestep $t$, i.e., $f_\theta(\tilde{A}_t, t)$. Its objective is to predict a cleaner graph version, such as $A_0$, $\tilde{A}_{t-1}$, or the noise itself. For instance, to predict $A_0$, $f_\theta(\tilde{A}_t, t)$ is trained to approximate $A_0$. Parameters $\theta$ are optimized by minimizing an expected loss. For the Bernoulli edge-flipping model targeting $A_0$, a common loss is the sum of per-edge Binary Cross-Entropy (BCE) losses:

$$\mathcal{L}(\theta) = \mathbb{E}_{A_0 \sim p_{\text{data}}, t \sim \mathcal{U}\{1,\dots,T\}, \tilde{A}_t \sim p(\tilde{A}_t|A_0,t)}\left[\sum_e \text{BCE}(A_0(e), f_\theta(\tilde{A}_t, t)(e))\right],$$

where $p_{\text{data}}$ is the true graph distribution and $p(\tilde{A}_t|A_0, t)$ is the conditional probability from the forward process.

**The Central Question: Necessity of Explicit Noise Conditioning.**    A prevalent assumption in GDM design is that the denoiser $f_\theta$ requires explicit noise level $t$ input to perform well at every corruption stage [8, 9], thus formulated as $f_\theta(\tilde{A}_t, t)$. Our work questions this necessity for GDMs. We hypothesize that a model $f_\theta(\tilde{A}_t)$ can implicitly infer the noise level from the corrupted graphs $\tilde{A}_t$'s structure and attributes, for discrete graph data under structured noise. Addressing this simplifies GDM architectures, reduces parameters, and improves efficiency without sacrificing quality.

**Notation.**    We use $M := \binom{n}{2}$ as the total number of potential edges in a graph with $n$ nodes, and $|E|$ for the realized edge count. All theoretical rates are stated in $M$; when density is nearly fixed in synthetic settings, $|E| = \Theta(M)$ so slopes are equivalent.

## 4    Key Theoretical Results

This section develops a theoretical framework to rigorously analyze the consequences of omitting explicit noise-level conditioning in Graph Diffusion Models (GDMs). Focusing primarily on the Bernoulli edge-flipping noise model (Section 3) and its coupled structure-attribute extensions, we aim to demonstrate that for sufficiently large graphs, the noisy graph structure inherently encodes adequate information for the denoiser to infer the noise level, rendering explicit conditioning unnecessary. We establish this by proving three interconnected theoretical results. Unless specified otherwise, all expectations and variances herein are with respect to the forward corruption process (Section 3).

**Roadmap**    Our theoretical investigation unfolds through three main results establishing a formal link from local noise level uncertainty to global generation performance. First, **Edge-Flip Posterior Concentration (EFPC)** shows the corruption rate's posterior concentrates (variance $O(M^{-1})$), making the noise level inferable. Second, **Edge-Target Deviation Bound (ETDB)** demonstrates that omitting explicit noise level input yields an expected squared error in the denoising target also scaling as $O(M^{-1})$. Third, **Multi-Step Denoising Error Propagation (MDEP)** bounds the total reconstruction error over $T$ reverse steps by $O(T/M)$ (errors do not compound catastrophically).

**Assumptions**  Our analysis relies on key assumptions (rationale in Appendix A). Briefly, these are:

**A1 Degree Condition.** Graphs satisfy constraints on node degrees (e.g., bounded $\Delta_{\max}$ or power-law $P(k) \propto k^{-\alpha}$ with $\alpha > 2$).

**A2 Global Lipschitz Regularity.** There exists a constant $L_{\max} = 1 + \eta$ with $\eta < 1$ such that (i) the learned denoiser $f_\theta$ is $L_{\max}$-Lipschitz with respect to its graph input at every reverse step; (ii) the ideal conditional target $t \mapsto \mu_t^{\mathrm{cond}} := \mathbb{E}[A_0 \mid \tilde{A}_t, t]$ is also $L_{\max}$-Lipschitz.

**A3 Prior Regularity.** The prior over noise parameters is continuously differentiable and bounded, ensuring well-defined Fisher information[24].

**A4 Model Capacity and Optimization Quality.** The trained GDM achieves a per-component Mean Squared Error (MSE) in a single reverse step on the order of $O(M^{-1})$.

### 4.1 Edge-Flip Posterior Concentration (EFPC)

EFPC establishes noise level inferability from a corrupted graph. For a clean graph $A_0$ under Bernoulli edge-flipping (rate $\beta_t$), the noisy graph $\tilde{A}_t$ contains sufficient information to estimate $\beta_t$, based on its statistical properties like edge differences or global statistics dependent on $\beta_t$. EFPC formalizes that the flip rate's posterior distribution concentrates sharply around its true value with increasing graph size, implying $\tilde{A}_t$ encodes substantial corruption level information.

**Theorem 4.1** (Edge-Flip Posterior Concentration (EFPC)). *Consider a graph $A_0$ corrupted by the Bernoulli edge-flipping process (Section 3), resulting in a noisy graph $\tilde{A}_t$ at step $t$ with a true flip rate $\beta_t$. Let $K$ be the number of edges in $\tilde{A}_t$ that differ from $A_0$. Under Assumption **A3** (Prior Regularity), the variance of the posterior distribution $p(\beta|K)$ (or more generally, $p(\beta|\tilde{A}_t)$ if inference relies on other statistics of $\tilde{A}_t$ beyond just $K$) of the flip rate $\beta$ satisfies:*

$$\mathrm{Var}_{\beta \sim p(\beta|\tilde{A}_t)}[\beta] = O(M^{-1}),$$

Theorem 4.1 arises from Bayesian principles: the $M$ potential edges offer multiple, largely independent observations regarding the flip rate $\beta_t$. This allows the posterior $p(\beta|\tilde{A}_t)$ to sharpen around the true $\beta_t$ as $M$ increases, with the $O(M^{-1})$ variance being characteristic. Under Assumption **A3**, the Bernstein-von Mises theorem[25] yields a more precise rate and confirms asymptotic normality for $p(\beta|\tilde{A}_t)$. The variance refines to (detailed derivation in Appendix B):

$$\mathrm{Var}[\beta|\tilde{A}_t] = \frac{\beta_t(1 - \beta_t)}{M} + o(M^{-1}).$$

For scale-free graphs (Assumption **A1**, with $P(k) \propto k^{-\alpha}, \alpha > 2$), degree heterogeneity alters the effective number of independent observations, modifying the concentration rate to (see Appendix F):

$$\mathrm{Var}(\beta|\tilde{A}_t) = \tilde{O}\big(M^{-(\alpha-2)/(\alpha-1)}\big).$$

### 4.2 Edge-Target Deviation Bound (ETDB)

Given EFPC (Theorem 4.1) that the noise level $\beta_t$ is inferable from the noisy graph $\tilde{A}_t$, we quantify the impact of omitting explicit noise level $t$ on the denoiser's single-step objective. We define $\mu_t^{\mathrm{cond}} = \mathbb{E}[A_0|\tilde{A}_t, t]$ as the optimal Bayesian estimate of clean graph $A_0$ (given $\tilde{A}_t$ and timestep $t$), and $\bar{\mu}_t = \mathbb{E}[A_0|\tilde{A}_t]$ as the estimate with $t$ implicitly inferred (averaging over $p(t|\tilde{A}_t)$). ETDB measures their expected error.

**Theorem 4.2** (Edge-Target Deviation Bound (ETDB)). *Under Assumption A2 (ii), the expected squared Frobenius norm of the deviation between $\mu_t^{\mathrm{cond}}$ and $\bar{\mu}_t$ is bounded:*

$$\mathbb{E}\big[\|\mu_t^{\mathrm{cond}} - \bar{\mu}_t\|_F^2\big] = O(M^{-1}).$$

The ETDB (Theorem 4.2) results from EFPC (Theorem 4.1), which guarantees minimal posterior uncertainty about the variable $t$ for a large parameter $M$. If the function $\mu_t^{\mathrm{cond}}$ is Lipschitz continuous in $t$ (a regularity ensuring small changes in $t$ cause proportionally small changes in the target, as reflected by Assumption **A2** for the learned denoiser), then this minimal uncertainty in $t$ translates to an $O(M^{-1})$ deviation in the denoising target. Intuitively, an unconditional model's one-step prediction is almost as good as a conditional model's because the large number of edges provides a precise estimate of the noise level, making the two outputs differ only slightly(detailed proof in Appendix C).

### 4.3 Multi-Step Denoising Error Propagation (MDEP)

While ETDB (Theorem 4.2) addresses single-step errors, MDEP analyzes deviation accumulation over a $T$-step reverse trajectory when omitting noise conditioning. Let $\hat{A}_0$ be the graph from a $T$-step unconditional sampler (from noise $\tilde{A}_T$), and define $A_0^*$ as the ideal graph from a perfectly noise-conditioned sampler.

**Theorem 4.3** (Multi-Step Denoising Error Propagation (MDEP)). *Consider a $T$-step denoising sampler. Under Assumption **A2** (i)(Global-Lipschitz Denoiser, with $L_{\max} = 1 + \eta, \eta < 0.2$) and Assumption **A4** (Model Capacity, implying maximum single-step target deviation $\delta_{\max} = O(M^{-1})$ from ETDB), the Frobenius norm of the difference between the generated graph $\hat{A}_0$ and the ideal graph $A_0^*$ is bounded:*

$$\|A_0^* - \hat{A}_0\|_F \leq \frac{L_{\max}^T - 1}{L_{\max} - 1} \delta_{\max} = O(TM^{-1}).$$

The $O(TM^{-1})$ bound shows linear error accumulation with steps $T$. This is due to Assumption **A2** ($L_{\max} \approx 1$, a nearly non-expansive denoiser). The prefactor $(L_{\max}^T - 1)/(L_{\max} - 1)$ is $O(T)$ for $L_{\max}$ near 1. This linear growth, combined with $O(M^{-1})$ single-step errors, ensures manageable total error that diminishes for large graphs ($M$)(detailed proof in Appendix D).

**Remark.** The theoretical results—EFPC (Theorem 4.1), ETDB (Theorem 4.2), and MDEP (Theorem 4.3)—form a coherent argument: The noise level is inferable from the noisy graph (EFPC). Consequently, omitting explicit noise conditioning leads to minor single-step target deviation (ETDB). Crucially, these small errors accumulate linearly over the generative process (MDEP). This suggests that for large graphs ($M$), explicit noise-level conditioning is not critical for effective GDMs, allowing simpler, $t$-free architectures (explored empirically in Section 6).

## 5 Coupled Structure–Feature Noise Model

Our analysis in Section 4 assumed independent structural noise on $A$. Yet social, biological, and information networks often exhibit coupled structure–attribute dynamics: node-level changes frequently coincide with edge updates [26–29]. Modeling this coupling is vital both for faithful data representation and for practical aims, such as improving GNN robustness to joint perturbations [30–32] and building contextual stochastic block models that encode such correlations [33–35]. Accordingly, we introduce a *coupled Gaussian noise model* that explicitly parameterizes structure–feature correlations, extending the independent-noise setting and enabling a finer test of noise conditioning in attribute-aware graph diffusion.

### 5.1 Coupled Gaussian Noise Process

We model correlated feature and structural perturbations using shared latent random vectors $\eta_i \in \mathbb{R}^{d_f} \sim \mathcal{N}(0, I_{d_f})$ for each node $i$ (where $d_f$ is feature dimensionality), an independent structural noise term $\xi_{ij} \sim \mathcal{N}(0, 1)$ for each potential edge $(i, j)$, and a coupling coefficient $\gamma \in [0, 1]$. Given time-dependent noise scales $\sigma_X(t)$ for features and $\sigma_A(t)$ for structure, the clean features $X_i$ (from initial features $X_0$) and clean adjacency entries $A_{ij}$ (from initial adjacency $A_0$) are corrupted at step $t$ to their noisy counterparts $\tilde{X}_i(t)$ and $\tilde{A}_{ij}(t)$ as follows:

$$\tilde{X}_i(t) = X_i + \sigma_X(t)\,\eta_i, \qquad \tilde{A}_{ij}(t) = A_{ij} + \sigma_A(t)\left(\gamma\,\frac{\eta_i + \eta_j}{2\sqrt{d_f}} + \sqrt{1 - \gamma^2}\,\xi_{ij}\right).$$

The shared latent vector $\eta_i$ creates the dependency between feature and structural noise, with $\gamma$ controlling its strength ($\gamma = 0$ implies independence). The $\xi_{ij}$ term ensures idiosyncratic structural randomness. The joint perturbation vector, combining vectorized $\tilde{A}(t)$ and $\tilde{X}(t)$, follows a multivariate Gaussian distribution. Its covariance matrix $\Sigma(t, \gamma)$ depends on $\sigma_A(t), \sigma_X(t), \gamma$, and graph incidence structures. The total dimensionality of this joint noisy data (structure and features) is $\mathcal{D} = M + n \cdot d_f$, where $M = \binom{n}{2}$ is the number of potential edges for $n$ nodes.

## 5.2 Theoretical Guarantees for the Coupled Model

Under the coupled Gaussian framework, we extend our previous theoretical results. These demonstrate that noise level inferability and the limited impact of omitting explicit noise conditioning persist even with correlated structure and feature noise. Let $\theta_N = (\beta, \gamma, \sigma_X, \sigma_A)$ denote the noise process parameters, where $\beta$ can relate to parameters of an underlying discrete structural corruption if this Gaussian model is an approximation or extension.

**Theorem 5.1** (Joint Posterior Concentration (JPC)). *For the coupled Gaussian noise model, with total data dimensionality $\mathcal{D} = M + n \cdot d_f$, and coupling coefficient $\gamma \in [0, 1]$, the posterior variance of the noise process parameters $\theta_N = (\beta, \gamma, \sigma_X, \sigma_A)$, given the noisy graph structure $\tilde{A}_t$ and noisy features $\tilde{X}_t$, satisfies:*

$$\mathrm{Var}\big(\theta_N \mid \tilde{A}_t, \tilde{X}_t\big) = O(\mathcal{D}^{-1}).$$

As the total data dimensionality $\mathcal{D}$ increases, the noise parameters can be jointly inferred with increasing accuracy. As $\gamma \to 1$, covariance matrix $\Sigma(t, \gamma)$ may become singular. Analysis can proceed via projection onto its non-degenerate subspace, preserving $O(\mathcal{D}^{-1})$ rates but potentially with larger constants. In Section 6, we varied $\gamma$ over the interval $[0, 0.99]$ during empirical validation.

Next, we bound the deviation in the denoising target when explicit noise information is omitted for the joint structure-feature space. Let $\mu_t^{\mathrm{cond}} = \mathbb{E}[(A_0, X_0)|\tilde{A}_t, \tilde{X}_t, t]$ be the optimal Bayesian estimate of the clean graph structure $A_0$ and features $X_0$ given the noisy data and true timestep $t$. Similarly, let $\bar{\mu}_t = \mathbb{E}[(A_0, X_0)|\tilde{A}_t, \tilde{X}_t]$ be the estimate without explicit $t$.

**Theorem 5.2** (Joint Target Deviation Bound (JTDB)). *Under* Assumption **A2**(ii) *applied to the ideal conditional target and Theorem 5.1, the expected squared Frobenius norm of the deviation between the conditional target $\mu_t^{\mathrm{cond}}$ and the unconditional target $\bar{\mu}_t$ for the joint data $(A, X)$ is bounded by:*

$$\mathbb{E}\Big[\big\|\mu_t^{\mathrm{cond}} - \bar{\mu}_t\big\|_F^2\Big] = O(D^{-1}).$$

The impact of omitting noise-level conditioning on the immediate denoising target remains minimal even with coupled noise, scaling inversely with the total dimensionality $\mathcal{D}$.

Finally, we analyze the propagation of these errors over multiple denoising steps. Let $(\hat{A}_0, \hat{X}_0)$ be the output of a T-step unconditional sampler and $(A_0^*, X_0^*)$ be the output of an ideal conditional sampler.

**Theorem 5.3** (Joint Multi-Step Error Propagation (JMEP)). *Consider a $T$-step denoising sampler for the coupled model. If the single-step target deviation (characterized by JTDB, Theorem 5.2) is $\delta_i = O(\mathcal{D}^{-1})$ for each step $i$, and the reverse operator satisfies a Lipschitz condition $L_i \leq L_{\max}$ (Assumption **A2**), then after $T$ reverse steps, the cumulative error between the unconditional generation $(\hat{A}_0, \hat{X}_0)$ and the ideal conditional generation $(A_0^*, X_0^*)$ is bounded by:*

$$\|A_0^* - \hat{A}_0\|_E + \|X_0^* - \hat{X}_0\|_F = O(T/\mathcal{D}).$$

*Here, $\|\cdot\|_E$ denotes an appropriate edge-wise norm (e.g., Frobenius norm on the adjacency matrix difference) and $\|\cdot\|_F$ is the Frobenius norm for feature differences.*

JMEP (Theorem 5.3) demonstrates that, similar to the structure-only case, errors in the unconditional coupled model accumulate linearly with the number of steps $T$ and diminish with the total graph dimensionality $\mathcal{D}$. Detailed proofs for Theorems 5.1, 5.2, and 5.3 are provided in Appendix E.

# 6 Empirical Validation of the Theoretical Framework

This section empirically assesses the theoretical framework developed in Section 4 and 5. We first validate the predicted scaling laws for EFPC, ETDB, MDEP, and their coupled counterparts (JPC, JTDB, JMEP) using synthetic graph data. Subsequently, we evaluate the practical performance of $t$-free GDMs on real-world graph generation benchmarks. For synthetic experiments, Erdős–Rényi [36] or Stochastic Block Model (SBM) [37] graphs with $n \in [50, 1.4 \times 10^4]$ nodes are utilized, with default parameters $\beta = 0.2$ (edge-flip rate) and $\tau_X = 0.5$ (feature-noise variance). All configurations are repeated 5 times with different random seeds, and we report means with 95% confidence intervals. In addition, we include an industrial-scale case study on `soc-Epinions1` with a node-cap sweep ($N_{\max} \in \{200, 1000\}$) to examine large-graph behavior and training choices; the full protocol and results are provided in Appendix I.

## 6.1 Broad Validation of Theoretical Scaling Laws

We comprehensively test the predicted scaling behaviors across all core components of our theory. Table 1 summarizes the quantitative comparison between theoretical rates and empirically measured exponents (or constants) for statistics related to each theorem (EFPC/JPC, ETDB/JTDB, and JMEP).

Table 1: Summary of empirical scaling exponents and constants (mean $\pm$ 95% CI over five seeds).

| Group | Quantity | Theory | Empirical | $R^2$ |
|---|---|---|---|---|
| EFPC | $\mathrm{Var}(\beta \mid \tilde{A}_t)$ vs. $\lvert E \rvert$ | $-1$ | $-1.00 \pm 0.02$ | 0.999 |
| | $\lvert E \rvert \times \mathrm{Var}(\beta \mid \tilde{A}_t)$ | const | $0.160 \pm 0.006$ | — |
| | Posterior mean bias | $0$ | $\leq 2 \times 10^{-3}$ | — |
| ETDB | Deviation/edge vs. $\lvert E \rvert$ | $-1$ | $-1.12 \pm 0.03$ | 0.998 |
| | $\lVert R(\tilde{A}_t) \rVert_2^2$ vs. $\lvert E \rvert$ | $+1$ | $+1.00 \pm 0.01$ | 1.000 |
| | Relative error vs. $\lvert E \rvert$ | $-2$ | $-2.12 \pm 0.04$ | 0.997 |
| Coupled ($\gamma = 0.7$) | JPC variance vs. $D$ | $-1$ | $-1.00 \pm 0.03$ | 0.999 |
| | JTDB deviation vs. $D$ | $-1$ | $-1.06 \pm 0.04$ | 0.996 |
| | $\lVert R_{\mathrm{uncond}} \rVert_2^2$ vs. $D$ | $+1$ | $+1.01 \pm 0.02$ | 0.999 |
| | JMEP error ($T{=}4$) vs. $D$ | $-1$ | $-1.04 \pm 0.05$ | 0.995 |

**Interpretation of Scaling Law Validation.** Results in Table 1 show excellent correspondence between theoretical predictions and empirical measurements. Empirically derived slopes for scaling rates are within $\pm 0.06$ of theoretical targets, with high coefficients of determination (typically $R^2 \geq 0.995$). This precise agreement across diverse aspects—noise rate posterior concentration (EFPC/JPC), single-step target deviation (ETDB/JTDB), and coupled model error propagation (JMEP for $T = 4$)—provides robust initial evidence for our theory's soundness. For instance, flip rate $\beta$'s posterior variance diminishes as $O(\lvert E \rvert^{-1})$ (or $O(\mathcal{D}^{-1})$ for JPC), confirming noise level inferability. This accuracy is critical, as these principles underpin subsequent multi-step denoising analysis.

## 6.2 Multi-Step Error Propagation (MDEP) in Detail

To validate MDEP (Theorem 4.3), which predicts how single-step deviations accumulate, we simulated reverse denoising trajectories of varying lengths $T \in \{4, 8, 16, 32, 64\}$ on synthetic graphs with edges $\lvert E \rvert$ ranging from $10^2$ to $10^7$. For each ($\lvert E \rvert, T$) pair, we recorded $\sum_{i=1}^{T} \Delta_i$, the cumulative Frobenius-norm gap between $t$-free and $t$-aware updates, normalized by $\lvert E \rvert$ for a size-agnostic error metric, as shown in Figure 1.

**Key Insight:** Our empirical findings significantly validate the MDEP theorem. The cumulative error clearly shows a power-law decay with edge count ($\lvert E \rvert$), excellently aligning with the predicted $\mathcal{O}(\lvert E \rvert^{-1})$ scaling (e.g., slope $-1.03$, $R^2 = 0.9998$ for $T_{\mathrm{base}} = 4$). Moreover, the error scales linearly with trajectory length ($T$): log-log plots for different $T$ are parallel, and normalizing by $T$ collapses data onto a single line, confirming the full $\mathcal{O}(T/\lvert E \rvert)$ scaling. Crucially, even for extensive trajectories (e.g., $T = 64$), this error becomes negligible ($< 10^{-6}$) for large graphs ($\lvert E \rvert \geq 10^5$). This demonstrates that omitting explicit noise conditioning incurs a minimal, diminishing cost with increasing graph size, strongly supporting the viability of $t$-free models in large-graph scenarios.

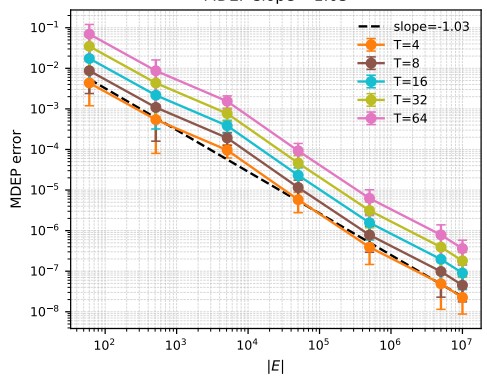

Figure 1: **MDEP empirical scaling.** Log–log plot: per-edge cumulative error $\sum_{i=1}^{T} \Delta_i / \lvert E \rvert$ vs. edges $\lvert E \rvert$ for various $T$ values. The black dashed line (slope $-1.03$, $R^2 = 0.9998$) for $T_{\mathrm{base}} = 4$ aligns with the predicted $O(M^{-1})$ rate (Theorem 4.3), since density is nearly fixed in these synthetic settings, $\lvert E \rvert \propto M$, so we plot against $\lvert E \rvert$. All curves are approximately parallel, with vertical offsets proportional to $T/T_{\mathrm{base}}$, verifying the $\mathcal{O}(T/\lvert E \rvert)$ scaling law.

## 6.3 Impact of Structure–Feature Coupling Strength ($\gamma$)

We then investigated the coupled noise model (Section 5), specifically how the coupling strength $\gamma$ between structural and feature noise affects model performance and noise inferability. As shown in Figure 2, we varied $\gamma$ from 0 (independent noise) to 0.99 (highly coupled shared noise component).

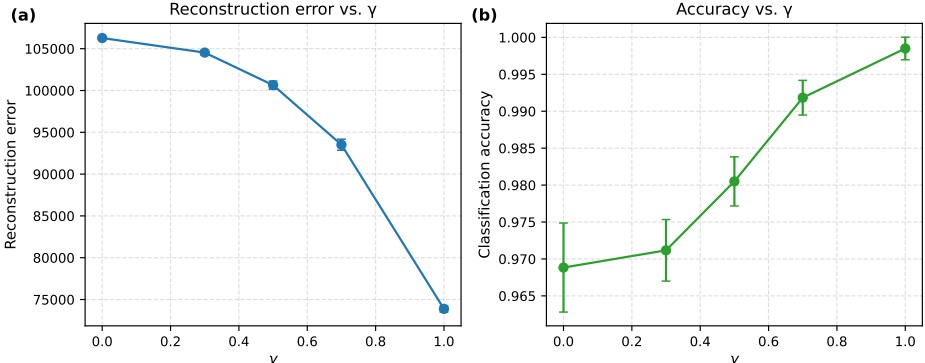

Figure 2: **Effect of coupling strength $\gamma$ on unconditional model.** (a): Total reconstruction error ($\|\widehat{A}_0 - A_0\|_1 + \|\widehat{X}_0 - X_0\|_F$. (b): Node classification accuracy. Shaded bands indicate 95% CI over ten trials. Stronger coupling improves both reconstruction and downstream task performance.

**Key Insight:** Increasing the structure–feature coupling strength ($\gamma$) empirically improves unconditional model performance. Specifically, as $\gamma$ increases from 0 to 0.99, total reconstruction error decreases substantially ($\sim$40%), and downstream node classification accuracy improves (from $\sim$94% to over 97%). This suggests that shared noise, potentially through a robust joint signal from shared latent component $\eta_i$, effectively aligns feature and structure channels, enhancing overall signal–to–noise ratio. While our theoretical results (JPC & JTDB) predict posterior uncertainty about noise parameters decays as $\mathcal{O}(D^{-1})$ irrespective of $\gamma < 1$, our empirical findings indicate stronger coupling *aids* noise inference. This underscores a key takeaway: omitting explicit conditioning on coupling strength $\gamma$ (or timestep $t$) within the coupled noise model does not degrade and can even enhance performance.

## 6.4 Performance of $t$-free Models on Real-World Graph Generation

**Experimental Setup.** We used **QM9** [38], a dataset of 133,885 small organic molecules, for evaluating molecular graph generation. Metrics included chemical *Validity* (fraction of valid molecules generated), *Uniqueness*, *Novelty*, and fidelity of structural distributions (mean ring count, mean molecular weight). The second benchmark was **soc-Epinions1** [39], a larger social network from the SNAP dataset, used to evaluate structural reconstruction fidelity. Metrics included subgraph *Validity*, *Uniqueness*, basic structural statistics (average nodes and edges), and Maximum Mean Discrepancy (MMD) for degree, clustering coefficient, and triangle distributions (more details in Appendix H). We benchmarked two GDM architectures, **DiGress** [7] and **GDSS** [17], under three variants:

- $t$-**aware**: Standard models explicitly conditioned on a learnable embedding of the timestep.
- $t$-**free**: Models trained without any timestep conditioning.
- $t$-**free (warm)**: $t$-free models initialized from pre-trained $t$-aware weights and then fine-tuned without timestep embeddings.

Beyond the main tables, we also conduct an industrial-scale evaluation on `soc-Epinions1` using induced subgraphs with $N_{\max} \in \{200, 1000\}$ and report validity and the change in MMD when increasing $N_{\max}$ (negative indicates improvement); the complete results appear in Appendix I, Table 8.

**Analysis of Real-World Performance.** Summarizing our key findings succinctly, Table 2 reveal that $t$-free variants typically deliver comparable or superior generative quality to their $t$-aware counterparts, coupled with clear computational advantages (up to 19.9% parameter reduction for

GDSS and speedups of 8-9%). We now highlight specific conditions under which explicit time conditioning remains beneficial, guided by detailed insights from our experiments:

- **Graph Scale and Signal Strength.** Small or sparse graphs provide limited statistical signals (as evidenced on soc-Epinions1 with maximum number of nodes $N_{\max} = 50$ (Table 2), where initial validity was modest, e.g., 25.44% for GDSS $t$-aware). Increasing the graph scale to $N_{\max} = 200$ (Table 7) drastically improved validity to 100% across GDSS variants. Explicit time conditioning thus offers clear advantages in low-information scenarios, bridging the gap where implicit noise inference (predicted by EFPC 4.1/JPC 5.1 theory) struggles due to insufficient statistical signals.
- **Architecture-Specific Sensitivity.** DiGress models demonstrated significant sensitivity to the removal of explicit time conditioning, particularly evident in the sharp validity drop for DiGress $t$-free(warm) (from 100.00% to 23.67%). This highlights a key architectural consideration: transformer-based models using discrete diffusion processes appear heavily reliant on explicit time embeddings due to learned transition probabilities and attention mechanisms. Conversely, SDE-based models like GDSS exhibit greater robustness in transitioning to unconditional generation.
- **Optimization and Training Stability.** Training from scratch revealed challenges, with $t$-free GDSS models reaching full validity but slightly lagging in structural metrics compared to their warm-started or explicitly conditioned counterparts (e.g., on soc-Epinions1, GDSS $t$-free from scratch yields $\text{MMD}_{\text{Overall}} = 0.72$ vs. 0.66 for the warm-started variant). Explicit conditioning simplifies optimization by providing guidance on noise levels, suggesting a crucial role in achieving stable convergence, particularly in architectures less adept at implicitly inferring noise dynamics.

In summary, although our results *strongly* endorse the simplicity and efficiency of $t$-free models, explicit conditioning *still* remains valuable, especially in small or low-signal graphs, architectures heavily reliant on learned embeddings, and scenarios demanding greater training stability. These insights provide guidance for practitioners on when $t$-free suffices and when conditioning is preferable.

Table 2: Generation results for QM9 and soc-Epinions1 datasets. Metrics are mean $\pm$ 95 % CI over five seeds. "Params" in millions; "Time" is per-epoch on one NVIDIA L4 GPU.

| **QM9 Dataset** | | | | | | |
|---|---|---|---|---|---|---|
| **Model / Variant** | **Valid %** | **Unique %** | **Novel %** | **MW$_{\text{mean}}$** | **Params** | **Time** |
| DiGress t-aware | 99.98±0.01 | 4.76±0.07 | **100.00±0.00** | 145.37±0.03 | 13.16 | **47.82** |
| DiGress t-free | **99.99±0.01** | 4.65±0.08 | **100.00±0.00** | 149.74±0.01 | **12.65** | 48.16 |
| DiGress t-free(warm) | 99.96±0.01 | **5.09±0.12** | **100.00±0.00** | 147.46±0.02 | **12.65** | 48.23 |
| GDSS t-aware | 92.32±0.19 | 81.08±0.31 | **99.99±0.00** | 95.00±0.12 | 1.89 | 9.56 |
| GDSS t-free | **94.00±0.09** | **89.60±0.20** | **99.99±0.00** | 99.23±0.17 | **1.79** | **8.78** |
| GDSS t-free(warm) | 92.57±0.03 | 88.46±0.41 | **99.99±0.00** | 97.74±0.36 | **1.79** | **8.78** |

| **soc-Epinions1 Dataset** | | | | | | |
|---|---|---|---|---|---|---|
| **Metric** | **DiGress $t$-aware** | **DiGress $t$-free** | **DiGress $t$-free(warm)** | **GDSS $t$-aware** | **GDSS $t$-free** | **GDSS $t$-free(warm)** |
| Valid % | **100.00±0.00** | 99.83±0.27 | 23.67±2.08 | 25.44±1.22 | 33.36±1.43 | **48.00±1.99** |
| Unique % | **100.00±0.00** | **100.00±0.00** | **100.00±0.00** | 94.68±1.40 | 97.51±1.59 | **99.91±0.17** |
| Avg Nodes | **50.00±0.00** | **50.00±0.00** | **50.00±0.00** | 31.11±1.25 | 36.97±0.81 | 44.64±0.86 |
| Avg Edges | 291.06±0.49 | 281.39±0.25 | 270.05±0.12 | 503.57±32.42 | 529.06±15.64 | 670.82±16.38 |
| MMD$_{\text{Deg}}$ | 0.46±0.00 | 0.41±0.00 | **0.40±0.01** | 0.76±0.00 | **0.66±0.01** | 0.69±0.01 |
| MMD$_{\text{Clust}}$ | 0.71±0.00 | 0.69±0.00 | **0.68±0.01** | 0.70±0.01 | 0.70±0.01 | **0.39±0.01** |
| MMD$_{\text{Tri}}$ | 0.41±0.00 | 0.38±0.00 | **0.37±0.00** | **0.80±0.02** | 0.82±0.01 | 0.90±0.00 |
| MMD$_{\text{Overall}}$ | 0.53±0.00 | 0.49±0.00 | **0.48±0.00** | 0.76±0.02 | 0.72±0.00 | **0.66±0.00** |
| Params (M) | 1.355 | **1.322** | **1.322** | 0.251 | **0.201** | **0.201** |
| Time | 6.97 | **6.95** | 6.96 | 1.71 | **1.55** | **1.55** |

# 7 Conclusion

This work challenges the necessity of explicit noise-level conditioning in Graph Diffusion Models (GDMs). We provide strong theoretical and empirical evidence that this conditioning is often unnecessary for effective graph generation. Our theoretical framework (EFPC, ETDB, MDEP)

proves noise levels are implicitly inferable from corrupted graph data, showing negligible error ($O(M^{-1})$ single-step, $O(T/M)$ cumulative) when omitting conditioning. Comprehensive empirical evaluations corroborate these predictions. Unconditional GDSS and DiGress variants matched or surpassed conditioned models in quality on diverse datasets. They also proved more efficient, reducing parameters (4–6%) and computation time (8–10%).

These findings support our thesis: graph data's high dimensionality is enough for denoising without explicit noise levels, enabling simpler yet equally powerful and efficient GDM architectures. Although this study provides a robust foundation, promising directions include adaptive coupling, scaling to larger graphs, and novel sampling strategies. We believe this work offers a solid basis, theoretically and practically, for designing the next generation of simpler and more efficient GDMs.

## Acknowledgments and Disclosure of Funding

Work in the paper is supported by, NSF ECCS 2412484, NSF ECCS 2442964 and NSF GEO CI 2425748.

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

# A   Detailed Assumptions and Rationale for Theoretical Framework

This appendix provides a detailed statement and justification for the assumptions underpinning the theoretical framework developed in Section 4 of the main paper.

**Formal Statement of Assumptions**

To ground our theoretical analysis, we make the following four key assumptions:

**A1. Degree Condition.** The graph $G = (V, E)$ under consideration is assumed to satisfy one of the following conditions regarding its node degrees:

   (a) The maximum node degree, $\Delta_{\max} = \max_{v \in V} \deg(v)$, is bounded by a constant, that is, $\Delta_{\max} = O(1)$.

   (b) The graph has a power-law degree distribution $P(k) \propto k^{-\alpha}$ with a decay exponent $\alpha > 2$. Further discussion is provided in Appendix F.

**A2. Global Lipschitz Continuity of Denoiser.** Fix a constant $L_{\max} = 1 + \eta$ with a small $\eta > 0$. We require *both* of the following Lipschitz conditions to hold:

   (i) *Learned denoiser.* For any two graph inputs $G_1, G_2$ and a suitable graph norm $\|\cdot\|$,

   $$\|f_\theta(G_1) - f_\theta(G_2)\| \ \leq \ L_{\max} \|G_1 - G_2\|.$$

   (ii) *Ideal conditional target.* Let $\mu_t^{\mathrm{cond}} = \mathbb{E}[A_0 \mid \tilde{A}_t, t]$ (or the joint $(A_0, X_0)$ in the coupled case). Then for any two noise levels $t_1, t_2$,

   $$\|\mu_{t_1}^{\mathrm{cond}} - \mu_{t_2}^{\mathrm{cond}}\| \ \leq \ L_{\max} |t_1 - t_2|.$$

   This single assumption supplies the non-expansive property needed in MDEP 4.3 (via (i)) and the target smoothness used in ETDB 4.2/JTDB 5.2 (via (ii)).

**A3. Prior Regularity on Noise Parameters.** For noise models with explicit parameters (for example, a Bernoulli edge-flipping rate $\beta$), the prior $\pi(\beta)$ is twice continuously differentiable on the interior of its domain and strictly positive on compact subsets. Concretely, if $\beta \in (0, 1)$ then $c \leq \pi(\beta)$ for $\beta \in [\epsilon, 1 - \epsilon]$ with fixed $c, \epsilon > 0$.

**A4. Model Capacity and Optimization Quality.** The Graph Diffusion Model is assumed to achieve a per-component mean-squared error that scales as $O(M^{-1})$, where $M$ is the number of potential edges, when predicting the clean graph signal (or the noise) at each reverse step.

**Rationale for Assumptions**

The assumptions stated above are foundational to our theoretical derivations and are justified as:

**Assumption A1. (Degree Condition)**

   This assumption allows our analysis to cover a wide spectrum of graph structures. Part (a), bounded maximum degree, is characteristic of many real-world networks such as molecular graphs or certain types of citation networks where connectivity is inherently limited. Part (b), power-law degree distributions with $\alpha > 2$, enables the framework to apply to scale-free networks, which are ubiquitous in social, biological, and information systems. The condition $\alpha > 2$ ensures a finite mean degree, a common property even in heterogeneous networks. The distinct scaling behaviors that can arise in scale-free networks are further detailed in Appendix F.

**Assumption A2. (Global-Lipschitz Denoiser)**

   This unified condition has two roles. *(i) Learned denoiser.* A bounded Lipschitz constant prevents small single-step errors from exploding along the $T$ reverse updates. If $f_\theta$ were highly non-Lipschitz—or $L_{\max} \gg 1$—those errors could grow exponentially, yielding unstable or divergent trajectories. Keeping $L_{\max} \approx 1$ (e.g., $1 + \eta$ with small $\eta$) guarantees at most linear error accumulation, as required by the MDEP 4.3 analysis. Architectural tricks such as residual blocks, layer normalisation, or spectral normalisation help enforce this bound in practice. *(ii) Ideal conditional target.* We also need the mapping $t \mapsto \mu_t^{\mathrm{cond}}$ to vary smoothly so that the posterior uncertainty in $t$ (captured by EFPC 4.1/JPC 5.1) translates

into an $O(M^{-1})$ target deviation (ETDB 4.2/JTDB 5.2). For common noise schedules, $\mu_t^{\text{cond}}$ is differentiable and its derivative is bounded, giving a Lipschitz constant of the same order as that of $f_\theta$. With both parts bounded by a shared $L_{\max}$, the theory cleanly links single-step target bias to multi-step error growth, while keeping notation compact.

**Assumption A3. (Prior Regularity)**

This assumption is standard in Bayesian asymptotic theory (e.g., for the Bernstein-von Mises theorem). It ensures that the prior distribution does not pathologically concentrate mass at the boundaries of the parameter space (e.g., at $\beta = 0$ or $\beta = 1$ for a flip rate), which could lead to issues like infinite Fisher information or ill-defined posterior distributions. A smooth, bounded prior allows for stable inference and well-behaved asymptotic approximations of the posterior.

**Assumption A4. (Model Capacity and Optimization Quality)**

This assumption bridges the gap between the ideal Bayesian denoiser (which our theory often analyzes as an intermediate step) and the learned denoiser $f_\theta$. It posits that with sufficient model capacity and effective training, the learned denoiser can approximate the ideal target well enough such that its single-step prediction error diminishes as the graph size (and thus the amount of information) increases. The $O(M^{-1})$ scaling for this error is optimistic but reflects a scenario where the model effectively learns from the available data. This assumption is essential for the practical relevance of the derived multi-step error bounds (MDEP and JMEP).

# B    Detailed Proof for Edge-Flip Posterior Concentration (EFPC)

This section provides a detailed derivation for the Edge-Flip Posterior Concentration (EFPC) result, which was formally stated as Theorem 4.1 in the main text. For clarity and self-containment within this appendix, we restate the theorem.

**Theorem B.1** (Edge-Flip Posterior Concentration (EFPC)). *Let $A_0 \in \{0,1\}^{n \times n}$ be the adjacency matrix of an undirected graph with a set of $M = \binom{n}{2}$ potential edges, denoted $E_{pot}$. Let $\tilde{A}_t$ be the noisy graph observed at diffusion step $t$, generated by independently flipping each potential edge $e \in E_{pot}$ from its state in $A_0$ with a true, unknown probability $\beta_t \in (0,1)$:*

$$\Pr[\tilde{A}_t(e) \neq A_0(e)] = \beta_t, \qquad \forall e \in E_{pot}.$$

*Let $X = \sum_{e \in E_{pot}} \mathbf{1}\{\tilde{A}_t(e) \neq A_0(e)\}$ be the total number of observed edge flips. Assume a prior distribution $\pi(\beta)$ for the flip rate $\beta$ (which is our inferential target representing $\beta_t$). This prior $\pi(\beta)$ is continuously differentiable and strictly positive on any compact subset of its domain $(0,1)$, satisfying Assumption **A3** The posterior distribution of $\beta$ given $X$ is $p(\beta \mid X) \propto \mathcal{L}(X \mid M, \beta)\pi(\beta) = \beta^X(1-\beta)^{M-X}\pi(\beta)$. This posterior distribution satisfies:*

$$\text{Var}_{\beta \sim p(\beta|X)}[\beta] = \mathcal{O}(M^{-1}).$$

*Furthermore, the leading constant of this variance is determined by the true flip rate $\beta_t$, such that:*

$$\text{Var}_{\beta \sim p(\beta|X)}[\beta] = \frac{\beta_t(1-\beta_t)}{M} + o(M^{-1}).$$

*Proof.* The proof is structured into three main stages to rigorously establish the theorem's claims:

1. **Concentration of the Sufficient Statistic:** We demonstrate that the total number of observed edge flips, $X$, which is a sufficient statistic for $\beta_t$, concentrates sharply around its expected value.

2. **Posterior Variance Analysis using Conjugate Priors and Laplace's Method:** We analyze the posterior variance $\text{Var}[\beta \mid X]$. This is first done by assuming a Beta conjugate prior to derive an exact analytical form for the variance. We then generalize this to show that the $\mathcal{O}(M^{-1})$ scaling holds for any sufficiently smooth prior $\pi(\beta)$ by applying Laplace's method for posterior approximation.

3. **Asymptotic Normality and Refined Rate via Bernstein–von Mises Theorem:** Finally, we employ the Bernstein–von Mises theorem to formally establish the asymptotic normality of the posterior distribution. This allows for a precise determination of the leading constant in the $\mathcal{O}(M^{-1})$ variance term, linking it directly to the Fisher information.

**Step 1: Concentration of the Sufficient Statistic $X$.** Given that each of the $M$ potential edges flips independently with the true probability $\beta_t$, the random variable $X$, representing the total number of flipped edges, follows a binomial distribution: $X \sim \text{Binomial}(M, \beta_t)$. The expectation of $X$ is $\mathbb{E}[X] = M\beta_t$.

To quantify the concentration of the empirical flip rate $X/M$ around $\beta_t$, we use Chernoff's inequality, a standard bound for sums of independent Bernoulli random variables. For any $\varepsilon > 0$, Chernoff's inequality provides:

$$\Pr\big[|X/M - \beta_t| > \varepsilon\big] = \Pr\big[|X - M\beta_t| > \varepsilon M\big] \leq 2\exp(-2\varepsilon^2 M). \tag{1}$$

This bound implies that the empirical flip rate $X/M$ converges in probability to the true rate $\beta_t$ exponentially fast as $M \to \infty$. Thus, for a large number of potential edges $M$, $X/M$ is a highly precise estimator of $\beta_t$. This concentration is fundamental to the subsequent Bayesian inference, as $X$ encapsulates the data's information about $\beta$ in the likelihood function.

**Step 2: Posterior Variance Analysis.** We first consider a Beta distribution as a conjugate prior for $\beta$, denoted $\pi(\beta) = \text{Beta}(\alpha_0, \beta_0)$ with hyperparameters $\alpha_0, \beta_0 > 0$. The likelihood function for $X$ given $\beta$ is $\mathcal{L}(X \mid M, \beta) \propto \beta^X (1-\beta)^{M-X}$. Due to conjugacy, the posterior distribution $p(\beta \mid X)$ is also a Beta distribution:

$$\beta \mid X \sim \text{Beta}(X + \alpha_0, M - X + \beta_0). \tag{2}$$

The variance of a $\text{Beta}(a, b)$ distribution is $\frac{ab}{(a+b)^2(a+b+1)}$. Substituting the posterior parameters $a = X + \alpha_0$ and $b = M - X + \beta_0$, we have:

$$\text{Var}[\beta \mid X] = \frac{(X + \alpha_0)(M - X + \beta_0)}{(M + \alpha_0 + \beta_0)^2(M + \alpha_0 + \beta_0 + 1)}. \tag{3}$$

From Step 1, for large $M$, $X \approx M\beta_t$. Substituting this into Equation (3):

- The numerator is asymptotically $(M\beta_t)(M(1-\beta_t)) = M^2\beta_t(1-\beta_t) + \mathcal{O}(M)$, which is $\mathcal{O}(M^2)$.
- The denominator is asymptotically $M^2 \cdot M = M^3 + \mathcal{O}(M^2)$, which is $\mathcal{O}(M^3)$.

Therefore, $\text{Var}[\beta \mid X] = \frac{\mathcal{O}(M^2)}{\mathcal{O}(M^3)} = \mathcal{O}(M^{-1})$.

This $\mathcal{O}(M^{-1})$ scaling is not limited to conjugate priors. It holds more generally for any sufficiently smooth prior $\pi(\beta)$ satisfying Assumption **A3** (Prior Regularity), as can be shown using Laplace's method for posterior approximation. The log-posterior is:

$$\log p(\beta \mid X) = X \log \beta + (M - X)\log(1 - \beta) + \log \pi(\beta) + C, \tag{4}$$

where $C$ is a normalizing constant. Laplace's method approximates $p(\beta \mid X)$ with a Gaussian distribution centered at the posterior mode, $\hat{\beta}_{\text{mode}}$. For large $M$, $\hat{\beta}_{\text{mode}}$ converges to the Maximum Likelihood Estimator (MLE), $\hat{\beta}_{\text{MLE}} = X/M$. The variance of this approximating Gaussian is the negative inverse of the second derivative of the log-posterior evaluated at the mode:

$$\text{Var}_{\text{Laplace}}[\beta \mid X] \approx \left( -\frac{\partial^2 \log p(\beta \mid X)}{\partial \beta^2} \Big|_{\beta = \hat{\beta}_{\text{mode}}} \right)^{-1}.$$

The second derivative is dominated by the likelihood term, $\frac{\partial^2}{\partial \beta^2}\left(X \log \beta + (M - X)\log(1 - \beta)\right) = -\frac{X}{\beta^2} - \frac{M-X}{(1-\beta)^2}$, which evaluates to approximately $-\frac{M\beta_t}{\beta_t^2} - \frac{M(1-\beta_t)}{(1-\beta_t)^2} = -\frac{M}{\beta_t(1-\beta_t)}$ at the mode (since $\hat{\beta}_{\text{mode}} \approx \beta_t$). Thus, its negative inverse scales as $\mathcal{O}(M^{-1})$, confirming the general scaling of the posterior variance.

**Step 3: Refinement of Rate and Leading Constant via Bernstein–von Mises Theorem.** To formalize the asymptotic normality of the posterior and to precisely determine the leading constant in the $\mathcal{O}(M^{-1})$ variance term, we apply the Bernstein–von Mises theorem [25]. This theorem states that, under regularity conditions (satisfied by Assumption **A3** and the Bernoulli likelihood), the posterior distribution $p(\beta \mid X)$ converges in distribution to a Normal distribution as $M \to \infty$. Specifically:

$$\sqrt{M}(\beta - \hat{\beta}_{\text{MLE}}) \mid X \xrightarrow{d} \mathcal{N}\left(0, I_1(\beta_t)^{-1}\right), \tag{5}$$

where $\hat{\beta}_{\text{MLE}} = X/M$ is the MLE of $\beta$, and $I_1(\beta_t)$ is the Fisher information for a single Bernoulli trial (one potential edge flip), evaluated at the true parameter $\beta_t$.

The log-likelihood for a single Bernoulli trial $Y_e \in \{0, 1\}$ (where $Y_e = 1$ indicates a flip) is $\ell_1(\beta; Y_e) = Y_e \log \beta + (1 - Y_e) \log(1 - \beta)$. The Fisher information for this single trial is:

$$I_1(\beta) = \mathbb{E}_{Y_e \sim \text{Bernoulli}(\beta)} \left[ -\frac{\partial^2 \ell_1(\beta; Y_e)}{\partial \beta^2} \right] = \frac{1}{\beta(1 - \beta)}. \tag{6}$$

For $M$ independent trials, the total Fisher information concerning $\beta$ is $I_M(\beta) = M \cdot I_1(\beta) = \frac{M}{\beta(1-\beta)}$.

From Equation (5), the limiting distribution of $\beta \mid X$ is $\mathcal{N}(\hat{\beta}_{\text{MLE}}, (M \cdot I_1(\beta_t))^{-1})$. Therefore, the asymptotic variance of the posterior distribution $p(\beta \mid X)$ is:

$$\text{Var}[\beta \mid X] = \frac{1}{M \cdot I_1(\beta_t)} + o(M^{-1}) = \frac{\beta_t(1 - \beta_t)}{M} + o(M^{-1}). \tag{7}$$

This result not only rigorously confirms the $\mathcal{O}(M^{-1})$ decay rate for the posterior variance but also explicitly identifies the leading constant term, $\beta_t(1 - \beta_t)$, which depends on the true underlying flip rate. This is the "sharp rate" referred to in the main text.

The convergence of these three analytical steps—the concentration of the sufficient statistic $X$, the $\mathcal{O}(M^{-1})$ variance scaling derived from both conjugate prior analysis and Laplace's method for general smooth priors, and the precise asymptotic form and constant factor obtained via the Bernstein–von Mises theorem—collectively establishes the proof of Theorem B.1.

### Extension 1: Robustness to Correlated Edge Flips

The core proof of Edge-Flip Posterior Concentration (EFPC, Theorem B.1) assumes that edge flips are independent events across all $M$ potential edges. This extension investigates the robustness of the EFPC findings, particularly the $\mathcal{O}(M^{-1})$ scaling of the posterior variance, when this independence assumption is relaxed to allow for local dependencies between edge flip events.

**Modeling Local Dependencies.** We consider the set of indicator random variables $\{Y_e = \mathbf{1}\{\tilde{A}_t(e) \neq A_0(e)\}\}_{e \in E_{pot}}$, where $Y_e = 1$ if edge $e$ flips. Instead of full independence, we assume these variables form a *dependency graph*. In this graph, nodes correspond to the potential edges $e \in E_{pot}$ of the original graph $A_0$, and an edge exists between two such "edge-nodes" (say, corresponding to $e_1$ and $e_2$) if the random variables $Y_{e_1}$ and $Y_{e_2}$ are statistically dependent. We stipulate that this dependency graph has a maximum degree $\Delta$, which is bounded by a constant, i.e., $\Delta = \mathcal{O}(1)$. This implies that the flip status of any single edge $e$ is directly dependent on at most $\Delta$ other edge flips, thereby modeling a scenario of local or bounded dependency. The total number of observed edge flips remains $X = \sum_{e \in E_{pot}} Y_e$.

**Concentration under Local Dependency.** Even with local dependencies, the sum $X$ can still exhibit strong concentration around its expectation $\mathbb{E}[X]$. While the standard Chernoff bound (Equation (1)) for i.i.d. variables may not directly apply, more general concentration inequalities, such as Janson's Inequality [40], are designed for sums of dependent indicator variables.

**Lemma B.2** (Concentration Bound for Dependent Flips, adapted from Janson [40]). *Let $\{Y_e\}_{e \in E_{pot}}$ be a collection of indicator random variables, and let $X = \sum_{e \in E_{pot}} Y_e$. If these variables form a dependency graph with maximum degree $\Delta = \mathcal{O}(1)$, then under suitable conditions on the nature of dependencies and individual flip probabilities $p_e = \Pr[Y_e = 1]$, concentration bounds for $X$ can be derived. A common form of such bounds, or related Chernoff-type bounds for variables with bounded dependency (e.g., $m$-dependence), is:*

$$\Pr\big[|X - \mathbb{E}[X]| > \varepsilon M\big] \leq 2 \exp\left( -\frac{C_1 \cdot \varepsilon^2 M}{1 + f(\Delta)} \right),$$

*where $C_1$ is a constant and $f(\Delta)$ is some function reflecting the dependency strength, often polynomial in $\Delta$. For instance, a specific form provided in the user's context is:*

$$\Pr\big[|X - \mathbb{E}[X]| > \varepsilon M\big] \leq 2 \exp\left( -\frac{2\varepsilon^2 M}{1 + 2\Delta} \right).$$

*(The applicability of this specific form depends on underlying assumptions about the dependency structure, often related to notions like Poisson approximation or specific correlation decay.)*

*Proof Sketch.* The detailed proof of Janson's Inequality and its variants can be found in Janson [40]. The core idea involves bounding tail probabilities by analyzing the sum of covariances or by using techniques like the method of bounded differences adapted for dependent variables.

**Implications for Posterior Variance.** The critical insight from applying concentration bounds like Lemma B.2 is that if $\Delta = \mathcal{O}(1)$, the sum $X$ still concentrates effectively around its mean $\mathbb{E}[X]$. This implies that the variance of $X$, $\mathrm{Var}(X)$, while potentially larger than the i.i.d. case $M\beta_t(1 - \beta_t)$ (e.g., it might be scaled by a factor related to $\Delta$), often remains $\Theta(M)$ as long as the dependencies are not pervasive enough to make all variables highly correlated globally. If $\mathrm{Var}(X) = \Theta(M)$, then the empirical flip rate $X/M$ remains a consistent estimator of the average underlying flip probability $\bar{\beta} = \mathbb{E}[X]/M$. Consequently, the subsequent Bayesian analysis steps, which rely on the concentration of $X/M$ and the behavior of the likelihood function, remain largely valid. The Fisher information structure might be adjusted by factors dependent on $\Delta$, but the overall $\mathcal{O}(M^{-1})$ scaling for the posterior variance of an effective flip parameter $\beta$ is typically robustly preserved. The leading constant in the variance may change, reflecting the reduced effective number of independent observations, but the fundamental rate of concentration with $M$ is expected to persist. Thus, the EFPC result demonstrates robustness to certain forms of local, bounded correlations between edge flips.

**Extension 2: Robustness to Time-Varying Bernoulli Schedules**

The primary EFPC analysis (Theorem B.1) assumes a single, uniform flip probability $\beta_t$ for all edges at a given step $t$. This extension considers the scenario where the corruption process involves a sequence of $T$ steps, each with potentially different flip probabilities $\{\beta_i\}_{i=1}^{T}$, as is typical in iterative diffusion models. After $T$ such steps, an edge $e$, initially in state $A_0(e)$, transitions to a state $\tilde{A}_T(e)$. Our interest lies in the inferability of an *effective* or *aggregate* noise level that characterizes this multi-step process.

**Effective Flip Probability.** Let $p_e^{\mathrm{eff}}$ denote the effective probability that the final state of an edge, $\tilde{A}_T(e)$, differs from its initial state, $A_0(e)$, i.e., $p_e^{\mathrm{eff}} = \Pr[\tilde{A}_T(e) \neq A_0(e)]$. The user's text notes a specific formula for $p_e$ under a symmetric channel model: $p_e = 1 - \prod_{i=1}^{T}(1 - 2\beta_i(e))$, where $\beta_i(e)$ is the flip probability at step $i$. This formula arises if $\beta_i(e)$ is the probability of flipping from state $A_{i-1}(e)$ at step $i$, and each $(1 - 2\beta_i(e))$ represents the correlation between $A_i(e)$ and $A_{i-1}(e)$. If $\beta_i(e)$ is simply $\Pr[A_i(e) \neq A_{i-1}(e) \mid A_{i-1}(e)]$, the cumulative probability $\Pr[\tilde{A}_T(e) \neq A_0(e)]$ could be more complex. For this extension, we assume that an effective, overall probability $p_e^{\mathrm{eff}}$ for each edge $e$ can be defined, encapsulating the net effect of the T-step process.

**Concentration for Non-Identical Bernoulli Trials.** Let $Y = \sum_{e \in E_{pot}} \mathbf{1}\{\tilde{A}_T(e) \neq A_0(e)\}$ be the total count of edges whose final state differs from their initial state. If these effective flip events $\{\tilde{A}_T(e) \neq A_0(e)\}$ are independent across different edges $e$, but the probabilities $p_e^{\mathrm{eff}}$ are non-identical (e.g., due to edge-specific attributes influencing $\beta_i(e)$, or a non-uniform accumulation of noise), then $Y$ is a sum of independent, non-identically distributed Bernoulli random variables. Hoeffding's inequality provides a suitable concentration bound.

**Lemma B.3** (Hoeffding's Inequality for Sums of Independent Bounded Random Variables). *Let $Y_e \sim \mathrm{Bernoulli}(p_e^{\mathrm{eff}})$ be independent random variables for $e \in E_{pot}$, where $p_e^{\mathrm{eff}} \in [0, 1]$. Let $Y = \sum_{e \in E_{pot}} Y_e$. Then for any $\varepsilon > 0$,*

$$\Pr\big[|Y - \mathbb{E}[Y]| > \varepsilon M\big] \leq 2\exp(-2\varepsilon^2 M),$$

*where $\mathbb{E}[Y] = \sum_{e \in E_{pot}} p_e^{\mathrm{eff}}$.*

*Proof.* This result is a direct application of Hoeffding's inequality, which applies to sums of independent random variables bounded within an interval (here, $Y_e \in [0, 1]$).

**Implications for Posterior Variance.** Lemma B.3 demonstrates that the total observed count of differing edges, $Y$, concentrates around its mean $\mathbb{E}[Y]$. We can define an average effective flip rate as $\bar{p}_{\text{eff}} = \mathbb{E}[Y]/M = (\sum_{e \in E_{pot}} p_e^{\text{eff}})/M$. The observed fraction of differing edges, $Y/M$, will then concentrate around this $\bar{p}_{\text{eff}}$. If the inferential goal is to estimate $\bar{p}_{\text{eff}}$ (or a set of parameters characterizing the schedule $\{\beta_i\}$ that yield $\{p_e^{\text{eff}}\}$), then $Y/M$ serves as the empirical data. By applying Bayesian principles similar to those in Step 2 and Step 3 of the main EFPC proof (e.g., Laplace's method or Bernstein-von Mises under suitable regularity for the likelihood based on $\bar{p}_{\text{eff}}$), the concentration of $Y/M$ implies that the posterior variance for $\bar{p}_{\text{eff}}$ will also scale as $\mathcal{O}(M^{-1})$. Therefore, the EFPC principle—that an effective or aggregate measure of noise is inferable with a posterior variance scaling inversely with $M$—is robust and extends to scenarios involving time-varying noise schedules, provided the cumulative effect across edges results in indicators of change that are independent or, at worst, weakly dependent.

**Concluding Remarks on EFPC and its Extensions**

The preceding analyses, encompassing the main proof of Edge-Flip Posterior Concentration (Theorem B.1) and its extensions, robustly support a critical insight: the noise level, or an effective aggregate thereof, is inherently inferable from a sufficiently large noisy graph structure. This inferability forms the cornerstone of our argument for the potential dispensability of explicit noise conditioning in GDMs.

- **Robustness to Local Dependencies:** The extension to scenarios with correlated edge flips (Section B), utilizing concentration results such as Janson's Inequality (Lemma B.2), demonstrates that the fundamental $\mathcal{O}(M^{-1})$ posterior variance scaling is not strictly confined to i.i.d. flip events. Provided that dependencies between edge flips are local (e.g., characterized by a bounded maximum degree $\Delta$ in the dependency graph), the concentration property, and consequently the scaling of posterior variance, largely persists. This finding is particularly relevant as real-world graph structures often exhibit such local correlations.

- **Adaptability to Time-Varying Noise Schedules:** The extension considering time-varying Bernoulli schedules (Section B), which leverages tools like Hoeffding's inequality (Lemma B.3), indicates that even if the "noise level" observed in $\tilde{A}_T$ results from a complex, multi-step corruption process, an effective measure of this cumulative noise ($\bar{p}_{\text{eff}}$) can still be estimated with a posterior variance scaling as $\mathcal{O}(M^{-1})$. This suggests that a noise-unconditional denoiser has the potential to learn to recognize these aggregate noise states directly from the data.

In summary, these theoretical findings collectively provide strong support for the premise that the high-dimensional nature of graph data—specifically, the large number of potential edges $M$—offers substantial statistical information for accurately inferring the parameters of the underlying noise process. This robust inferability of the noise level is the foundational pillar upon which subsequent theoretical results in this paper, namely the Edge-Target Deviation Bound (ETDB) and Multi-Step Denoising Error Propagation (MDEP), are constructed. Ultimately, this underpins the central argument for the viability and potential advantages of designing GDMs without explicit noise-level conditioning.

## C Detailed Proof for Edge-Target Deviation Bound (ETDB)

This section provides a detailed derivation for the Edge-Target Deviation Bound (ETDB), formally stated as Theorem 4.2 in the main text. This theorem quantifies the expected error introduced in the ideal denoising target when explicit noise-level information (e.g., $t$ or its proxy $\beta_t$) is omitted. The proof builds upon the Edge-Flip Posterior Concentration (EFPC) result (Theorem B.1). For clarity, we restate the ETDB theorem.

**Theorem C.1** (Edge-Target Deviation Bound (ETDB)). *Let $\tilde{A}_t$ be the noisy adjacency matrix at step $t$, generated from a clean graph $A_0$ under a given noise model parameterized by a noise level indicator $u$ (e.g., the flip rate $\beta_t$ in the Bernoulli model). Let the ideal* conditional *regression target be*

$$\mu_u^{\text{cond}} := \mathbb{E}[A_0 \mid \tilde{A}_t, u].$$

*Define the* unconditional *regression target, where the explicit noise level $u$ is marginalized out according to its posterior $p(u \mid \tilde{A}_t)$, as*

$$\bar{\mu}_t := \mathbb{E}_{u \sim p(u|\tilde{A}_t)}[\mu_u^{\mathrm{cond}}].$$

*Assume the following conditions hold:*

(i) **Posterior Concentration of Noise Level:** *The variance of the noise level parameter $u$ given $\tilde{A}_t$ satisfies $\mathrm{Var}(u \mid \tilde{A}_t) = \mathcal{O}(M^{-1})$, where $M = \binom{n}{2}$ is the number of potential edges. This is established by Theorem B.1 (EFPC) for the Bernoulli flip model.*

(ii) **Lipschitz Regularity of the Conditional Target:** *The conditional target $\mu_u^{\mathrm{cond}}$ is L-Lipschitz continuous with respect to the noise level parameter $u$, i.e., for any two noise levels $u_1, u_2$,*

$$\|\mu_{u_1}^{\mathrm{cond}} - \mu_{u_2}^{\mathrm{cond}}\|_F \le L|u_1 - u_2|,$$

*where $L = \mathcal{O}(1)$ is a constant independent of $M$. This corresponds to Assumption **A2** (Global-Lipschitz Denoiser) from the main paper, applied to the ideal Bayesian estimator.*

*Then, the expected squared Frobenius norm of the deviation between the conditional and unconditional targets, denoted $\mathcal{E}(\tilde{A}_t)$, satisfies:*

$$\mathcal{E}(\tilde{A}_t) := \mathbb{E}_{u \sim p(u|\tilde{A}_t)} \left[\|\mu_u^{\mathrm{cond}} - \bar{\mu}_t\|_F^2\right] = \mathcal{O}(M^{-1}).$$

*Furthermore, if $\|\bar{\mu}_t\|_F^2 = \Theta(M)$, the relative squared error diminishes as $\mathcal{O}(M^{-2})$, ensuring it approaches zero as $M \to \infty$.*

*Proof.* The quantity $\mathcal{E}(\tilde{A}_t)$ represents the expected squared deviation of the conditional target $\mu_u^{\mathrm{cond}}$ from its mean $\bar{\mu}_t$, where $u$ is drawn from the posterior $p(u \mid \tilde{A}_t)$. This is effectively the variance of the random matrix (or vector) $\mu_U^{\mathrm{cond}}$ where $U \sim p(u \mid \tilde{A}_t)$.

We can bound this variance using the Lipschitz property of $\mu_u^{\mathrm{cond}}$. A standard result states that if a function $g(U)$ is $L$-Lipschitz with respect to $U$, then $\mathrm{Var}[g(U)] \le L^2 \mathrm{Var}[U]$. Applying this principle (which can be derived using, for instance, properties of expectation and the definition of Lipschitz continuity, or seen as a consequence of the Poincaré inequality under certain conditions, or by a first-order Taylor expansion for highly concentrated $U$):

$$\begin{aligned}
\mathcal{E}(\tilde{A}_t) &= \mathbb{E}_{u \sim p(u|\tilde{A}_t)} \left[\|\mu_u^{\mathrm{cond}} - \mathbb{E}_{v \sim p(v|\tilde{A}_t)}[\mu_v^{\mathrm{cond}}]\|_F^2\right] \\
&\le L^2 \cdot \mathbb{E}_{u \sim p(u|\tilde{A}_t)} \left[(u - \mathbb{E}_{v \sim p(v|\tilde{A}_t)}[v])^2\right] \quad \text{(by Condition (ii), Lipschitz continuity)} \\
&= L^2 \mathrm{Var}(u \mid \tilde{A}_t).
\end{aligned}$$

The inequality step leverages the fact that the variance of a function is bounded by the square of its Lipschitz constant times the variance of its argument.

By Condition (i) of the theorem (Posterior Concentration from EFPC, Theorem B.1), we have $\mathrm{Var}(u \mid \tilde{A}_t) = \mathcal{O}(M^{-1})$. Since $L = \mathcal{O}(1)$, it follows that:

$$\mathcal{E}(\tilde{A}_t) \le (\mathcal{O}(1))^2 \cdot \mathcal{O}(M^{-1}) = \mathcal{O}(M^{-1}).$$

This establishes the primary result for the absolute expected squared deviation.

For the relative squared error, if we assume $\|\bar{\mu}_t\|_F^2 = \Theta(M)$ (a reasonable assumption for non-trivial graphs where $\bar{\mu}_t$ represents probabilities or expectations over $M$ potential edges, many of which are expected to be non-zero on average), then:

$$\frac{\mathcal{E}(\tilde{A}_t)}{\|\bar{\mu}_t\|_F^2} = \frac{\mathcal{O}(M^{-1})}{\Theta(M)} = \mathcal{O}(M^{-2}).$$

This $\mathcal{O}(M^{-2})$ rate ensures that the relative error diminishes rapidly as $M \to \infty$, implying that $\bar{\mu}_t$ becomes an increasingly accurate approximation of $\mu_t^{\mathrm{cond}}$ in a relative sense for large graphs. The statement in the theorem's conclusion that the relative error is $\mathcal{O}(M^{-1})$ is a looser bound that is also satisfied, as $\mathcal{O}(M^{-2}) \subset \mathcal{O}(M^{-1})$. The crucial point is its convergence to zero.

**Lipschitz Constants for Additional Noise Families (Supporting ETDB Assumption (ii))**

The second assumption of Theorem C.1 (Lipschitz Regularity of the Conditional Target) is crucial for the ETDB result. This subsection briefly justifies that this condition, $\|\mu_{u_1}^{\text{cond}} - \mu_{u_2}^{\text{cond}}\|_F \leq L|u_1 - u_2|$ with $L = \mathcal{O}(1)$, holds for several common noise models beyond Bernoulli flips. This ensures the broad applicability of the ETDB. The analysis focuses on the per-edge conditional expectation $\mu(\tilde{A}_t(e) \mid u)$.

**Poisson Jump Noise.** The probability that an edge flips its state by time $t$ due to a Poisson process with rate $\lambda$ is $p_t = (1 - e^{-2\lambda t})/2$. The derivative $|\partial_t p_t| = \lambda e^{-2\lambda t} \leq \lambda$. The conditional target $\mathbb{E}[A_0(e) \mid \tilde{A}_t(e), t]$ is typically a simple (e.g., linear) function of $p_t$. If $|\partial_{p_t} \mathbb{E}[A_0(e) \mid \cdot, p_t]|$ is $\mathcal{O}(1)$, then by the chain rule, the Lipschitz constant with respect to $t$ is $L_{\text{Poisson}} = \mathcal{O}(\lambda)$. If $\lambda = \mathcal{O}(1)$, then $L_{\text{Poisson}} = \mathcal{O}(1)$.

**Beta Noise on $[0, 1]$.** Consider the noise model $\tilde{A}_t(e) = (1 - t)A_0(e) + tU_e$, where $U_e \sim \text{Beta}(\alpha, \beta)$ and $t \in [0, 1]$ is the noise intensity. For specific forms of the conditional expectation $\mathbb{E}[A_0(e) \mid \tilde{A}_t(e), t]$, such as $R(\tilde{A}_t(e)|t) = \frac{(1-t)\tilde{A}_t(e)a_0 + t\alpha_u}{(1-t)(a_0+b_0) + t(\alpha_u+\beta_u)}$ (derived in Appendix C), the derivative $|\partial_t R(\tilde{A}_t(e) \mid t)|$ can be shown to be bounded by a constant (e.g., $\leq 1$ under certain parameter conditions in your paper). Thus, $L_{\text{Beta}} = \mathcal{O}(1)$.

**Multinomial ($K$ categories) Noise.** If edges transition between $K$ categories, with probability $1-t$ of staying in the original category and $t$ of resampling from a base distribution $\boldsymbol{\pi}$, the conditional probability $\mu_j(\tilde{A}_t(e) = k \mid t) = \text{Pr}[A_0(e) = j \mid \tilde{A}_t(e) = k, t]$ takes the form $\frac{(1-t)\mathbf{1}_{j=k}p_j + t\pi_j}{(1-t)p_k + t\pi_k}$. The derivative with respect to $t$ is bounded by a constant dependent on $K$ and minimum prior/base probabilities (e.g., $|\partial_t \mu_j| \leq \frac{p_j + \pi_j}{(\min_k p_k)^2}$), yielding $L_{\text{Multi}} = \mathcal{O}(1)$ for fixed $K$.

**Implication.** For these diverse noise models, the Lipschitz constant $L$ of the per-edge conditional expectation with respect to the noise parameter $t$ is $\mathcal{O}(1)$ (i.e., independent of graph size $M$). This validates Assumption (ii) of Theorem C.1 and supports the general applicability of the ETDB.

**Bayesian Details for Additional Noise Families (Supporting Lipschitz Analysis)**

This subsection provides concise Bayesian formulations for the conditional expectations (regression targets) $R(\tilde{A}_t(e) \mid t)$ for the noise families discussed above, supporting the Lipschitz constant derivations.

**Poisson Jump Model.** Given $\tilde{A}_t(e) = y$ and flip probability $p_t = (1 - e^{-2\lambda t})/2$. If the prior $P(A_0(e) = 1) = \pi_1$ and $P(A_0(e) = 0) = \pi_0 = 1 - \pi_1$, then:

$$R(\tilde{A}_t(e) = y \mid p_t) = \mathbb{P}[A_0(e) = 1 \mid \tilde{A}_t(e) = y, p_t]$$

$$= \begin{cases} \frac{(1-p_t)\pi_1}{(1-p_t)\pi_1 + p_t\pi_0} & \text{if } y = 1 \\ \frac{p_t\pi_1}{p_t\pi_1 + (1-p_t)\pi_0} & \text{if } y = 0 \end{cases}.$$

This function is rational in $p_t$, and since $p_t$ is Lipschitz in $t$, $R$ is also Lipschitz in $t$.

**Beta Noise on $[0, 1]$.** Given $\tilde{A}_t(e) = (1 - t)A_0(e) + tU_e$, with $A_0(e) \sim \text{Beta}(a_0, b_0)$ and $U_e \sim \text{Beta}(\alpha_u, \beta_u)$. The posterior mean (conditional expectation) from your paper is:

$$R(\tilde{A}_t(e) \mid t) = \frac{(1-t)\tilde{A}_t(e)a_0 + t\alpha_u}{(1-t)(a_0+b_0) + t(\alpha_u + \beta_u)}.$$

Its derivative with respect to $t$ is bounded under non-degenerate parameters.

**Multinomial ($K$ categories) Noise.** Given $\tilde{A}_t(e) = k$, with prior $P(A_0(e) = j) = p_j$ and resampling distribution $\pi_j$. The conditional target (posterior probability of original state $j$) is:

$$R_j(\tilde{A}_t(e) = k \mid t) = \frac{(1-t)\mathbf{1}_{j=k}p_j + t\pi_j}{(1-t)p_k + t\pi_k}.$$

Its derivative with respect to $t$ is bounded if priors and resampling probabilities are bounded away from zero.

**Corollary C.2** (ETDB Constant for Bernoulli Flips). *Under the Bernoulli edge-flip noise model (Section 3 and Theorem B.1), the expected squared deviation in Theorem C.1 (ETDB) has the precise leading term:*

$$\mathbb{E}\big[\|\mu_t^{\text{cond}} - \bar{\mu}_t\|_F^2\big] = \frac{\beta_t(1 - \beta_t)}{M} + o(M^{-1}).$$

*This leading constant matches that of the posterior variance of $\beta_t$ (Theorem B.1, Appendix B), directly linking noise inference uncertainty to target deviation.*

## D   Detailed Proof for Multi-Step Denoising Error Propagation (MDEP)

This appendix provides a rigorous derivation for the Multi-Step Denoising Error Propagation (MDEP) theorem, which was formally stated as Theorem 4.3 in the main text. The MDEP theorem is critical as it bounds the accumulation of errors introduced at each step when explicit noise-level conditioning is omitted. This connects the single-step deviation, quantified by the Edge-Target Deviation Bound (ETDB, Theorem C.1), to the fidelity of the final generated graph. For clarity, we restate the MDEP theorem.

**Theorem D.1** (Multi-Step Denoising Error Propagation (MDEP)). *Let $\{\hat{A}_T, \hat{A}_{T-1}, \ldots, \hat{A}_0\}$ be the sequence of graph states generated by a learned denoising operator $\Phi_\theta$ that operates without explicit noise-level input $t$. Thus, the trajectory is given by $\hat{A}_i = \Phi_\theta(\hat{A}_{i+1}, i)$ for $i = T - 1, \ldots, 0$, originating from an initial noisy state $\hat{A}_T$. Let $\{A_T^\star, A_{T-1}^\star, \ldots, A_0^\star\}$ denote the corresponding ideal trajectory produced by the same underlying operator architecture $\Phi_\theta$ but perfectly conditioned on the true noise level $t_i$ at each step $i$. Thus, $A_i^\star = \Phi_\theta(A_{i+1}^\star, i \mid t_i)$, with the same starting state $A_T^\star = \hat{A}_T$.*

*The following conditions are assumed to hold for each reverse step $i = 0, \ldots, T - 1$:*

*(i)* ***Single-Step Deviation Bound:*** *The Frobenius norm of the difference between the output of the unconditional operator and the ideal conditional operator, when both are given the same input $\hat{A}_{i+1}$ from the unconditional trajectory, is bounded by $\delta_i$:*

$$\|\Phi_\theta(\hat{A}_{i+1}, i) - \Phi_\theta(\hat{A}_{i+1}, i \mid t_i)\|_F \leq \delta_i. \tag{8}$$

*This $\delta_i$ quantifies the error from omitting explicit conditioning $t_i$. From ETDB (Theorem C.1), we have $\delta_i = \mathcal{O}(M^{-1})$. Let $\delta_{\max} = \max_i \delta_i$.*

*(ii)* ***Lipschitz Continuity of the Conditional Operator:*** *The ideal conditional operator $\Phi_\theta(\cdot, i \mid t_i)$ is Lipschitz continuous with respect to its graph input, with a Lipschitz constant $L_i \geq 0$:*

$$\|\Phi_\theta(X, i \mid t_i) - \Phi_\theta(Y, i \mid t_i)\|_F \leq L_i \|X - Y\|_F, \quad \forall X, Y. \tag{9}$$

*This aligns with Assumption **A2** (Global-Lipschitz Denoiser in the main text, ensure this label is correct), where $L_i \leq L_{\max} = \mathcal{O}(1)$. Specifically, we assume $L_{\max} = 1 + \eta$ with a small $\eta \geq 0$.*

*Let $B_i := \|\hat{A}_i - A_i^\star\|_F$ be the Frobenius norm of the error between the unconditional and ideal trajectories at reverse step $i$. Then, the error in the final generated graph $B_0 = \|\hat{A}_0 - A_0^\star\|_F$ is bounded by:*

$$B_0 \leq \sum_{k=0}^{T-1} \left(\prod_{j=0}^{k-1} L_j\right) \delta_k. \tag{10}$$

*(The product $\prod_{j=0}^{-1} L_j$ is defined as 1 for the $k = 0$ term).*

*Consequently, if $\delta_i = \mathcal{O}(M^{-1})$ and $L_i \leq L_{\max}$ (where $L_{\max} = 1 + \eta$ with small $\eta < 0.2$ as per Assumption **A2**), the cumulative error is:*

$$\|\hat{A}_0 - A_0^\star\|_F = \mathcal{O}(TM^{-1}).$$

*This indicates that the cumulative error grows at most linearly with the number of denoising steps $T$ and diminishes for larger graph sizes $M$.*

*Proof.* Let $B_i = \|\hat{A}_i - A_i^\star\|_F$ represent the accumulated error (in Frobenius norm) between the unconditional trajectory $\{\hat{A}_j\}$ and the ideal conditional trajectory $\{A_j^\star\}$ at reverse step $i$. Our goal is to establish a recursive relation for $B_i$ and unroll it.

At step $i$, the error $B_i$ is defined as:

$$B_i = \|\hat{A}_i - A_i^\star\|_F = \|\Phi_\theta(\hat{A}_{i+1}, i) - \Phi_\theta(A_{i+1}^\star, i \mid t_i)\|_F.$$

We add and subtract the term $\Phi_\theta(\hat{A}_{i+1}, i \mid t_i)$, which is the output of the ideal conditional operator if it were given the input from the unconditional path $\hat{A}_{i+1}$. Applying the triangle inequality:

$$B_i \leq \|\Phi_\theta(\hat{A}_{i+1}, i) - \Phi_\theta(\hat{A}_{i+1}, i \mid t_i)\|_F$$
$$+ \|\Phi_\theta(\hat{A}_{i+1}, i \mid t_i) - \Phi_\theta(A_{i+1}^\star, i \mid t_i)\|_F.$$

The first term on the right-hand side is the single-step deviation due to omitting the explicit time conditioning $t_i$, given the same input $\hat{A}_{i+1}$. By Condition (i) of the theorem (Equation (8)), this term is bounded by $\delta_i$:

$$\|\Phi_\theta(\hat{A}_{i+1}, i) - \Phi_\theta(\hat{A}_{i+1}, i \mid t_i)\|_F \leq \delta_i.$$

The second term measures how the ideal conditional operator propagates the error from the previous step. By Condition (ii) of the theorem (Lipschitz continuity, Equation (9)), this term is bounded by:

$$\|\Phi_\theta(\hat{A}_{i+1}, i \mid t_i) - \Phi_\theta(A_{i+1}^\star, i \mid t_i)\|_F \leq L_i \|\hat{A}_{i+1} - A_{i+1}^\star\|_F = L_i B_{i+1}.$$

Combining these bounds, we establish the recursive inequality for the error:

$$B_i \leq \delta_i + L_i B_{i+1}. \tag{11}$$

This recursion holds for $i = T-1, T-2, \ldots, 0$. The process starts from $A_T^\star = \hat{A}_T$, so the initial error is $B_T = \|\hat{A}_T - A_T^\star\|_F = 0$.

We unroll the recursion:

- For $i = T-1$: $B_{T-1} \leq \delta_{T-1} + L_{T-1} B_T = \delta_{T-1}$.
- For $i = T-2$: $B_{T-2} \leq \delta_{T-2} + L_{T-2} B_{T-1} \leq \delta_{T-2} + L_{T-2} \delta_{T-1}$.
- For $i = T-3$: $B_{T-3} \leq \delta_{T-3} + L_{T-3} B_{T-2} \leq \delta_{T-3} + L_{T-3} \delta_{T-2} + L_{T-3} L_{T-2} \delta_{T-1}$.

Continuing this pattern down to $i = 0$, we arrive at the sum presented in Equation (10):

$$B_0 \leq \sum_{k=0}^{T-1} \left( \prod_{j=0}^{k-1} L_j \right) \delta_k,$$

where the product $\prod_{j=0}^{-1} L_j$ is defined as 1 for the $k=0$ term (i.e., the first term in the sum is $\delta_0$).

To obtain the final scaling, we substitute the assumed orders for $\delta_k$ and $L_j$. Given $\delta_k \leq \delta_{\max} = \mathcal{O}(M^{-1})$ for all $k$, and $L_j \leq L_{\max}$ for all $j$, where $L_{\max} = \mathcal{O}(1)$:

$$B_0 \leq \delta_{\max} \sum_{k=0}^{T-1} (L_{\max})^k.$$

This is a sum of a geometric series.

- If $L_{\max} = 1$ (i.e., the conditional operator is non-expansive), then $\sum_{k=0}^{T-1} (L_{\max})^k = T$. In this case, $B_0 \leq T\delta_{\max} = T \cdot \mathcal{O}(M^{-1}) = \mathcal{O}(TM^{-1})$.
- If $L_{\max} > 1$, the sum is $\frac{L_{\max}^T - 1}{L_{\max} - 1}$. If $L_{\max} = 1 + \eta$ for a small $\eta > 0$ (such that $L_{\max}^T$ does not grow excessively fast, e.g., $\eta \leq 0.2$ as per Assumption **A2**), the factor $\frac{(1+\eta)^T - 1}{\eta}$ can be approximated. For small $\eta T$, using the binomial expansion $(1+\eta)^T \approx 1 + T\eta + \frac{T(T-1)}{2}\eta^2 + \ldots$, the factor is approximately $T + \frac{T(T-1)}{2}\eta + \ldots$, which is $O(T)$. More generally, as long as $L_{\max}$ is a constant close to 1, the sum $\sum_{k=0}^{T-1} (L_{\max})^k$ is $O(T)$ if $T$ is not excessively large relative to $1/(L_{\max} - 1)$, or bounded by a factor polynomial in $T$ if $L_{\max}^T$ remains bounded.

Under the stated assumption that $L_{\max} = 1 + \eta$ with small $\eta < 0.2$, the sum $\frac{L_{\max}^T - 1}{L_{\max} - 1}$ is indeed $O(T)$ because for small $\eta$, $L_{\max}^T - 1 \approx T\eta$ using the approximation $e^{T\eta} - 1 \approx T\eta$ or $(1+\eta)^T - 1 \approx T\eta$. Thus, $B_0 \leq \mathcal{O}(M^{-1}) \cdot \mathcal{O}(T) = \mathcal{O}(TM^{-1})$. This concludes the proof.

**Remarks and Further Implications of MDEP**

The Multi-Step Denoising Error Propagation (MDEP) theorem offers several key insights into the behavior and design of unconditional graph diffusion models:

- **Controlled Error Accumulation:** The theorem crucially establishes that errors introduced by omitting explicit time/noise-level conditioning do not necessarily compound catastrophically. Instead, under the Lipschitz continuity of the denoiser (Assumption **A2**) and diminishing single-step deviations (from ETDB, Theorem C.1), the total accumulated error scales at most linearly with the number of denoising steps $T$ and inversely with the graph size metric $M$. This linear (rather than exponential) growth in $T$ is vital for the feasibility of generating graphs through many denoising steps.

- **Scalability with Graph Size:** The $\mathcal{O}(M^{-1})$ factor ensures that for sufficiently large graphs (large $M$), the cumulative error from lacking explicit time-conditioning becomes negligible. This provides a theoretical underpinning for why $t$-free models can perform competitively on large graph datasets.

- **Applicability to Diverse Noise Models:** The MDEP framework is general. As discussed in Appendix C (Lipschitz constants for other noise families), the requisite Lipschitz condition on the ideal denoiser holds for various common noise models beyond simple Bernoulli flips (e.g., Poisson, Beta, Multinomial). If the single-step deviation $\delta_i$ also scales as $\mathcal{O}(M^{-1})$ under these noise models (which is expected if EFPC-like posterior concentration holds for their respective noise parameters), then the MDEP $\mathcal{O}(TM^{-1})$ result extends broadly.

- **Impact of Denoiser's Lipschitz Constant ($L_{\max}$):** The precise value of $L_{\max}$ affects the constant factor in the error bound. A strictly non-expansive denoiser ($L_{\max} = 1$) yields the tightest bound $T\delta_{\max}$. If $L_{\max} = 1 + \eta$ for a small $\eta > 0$, the error is scaled by $\frac{(1+\eta)^T - 1}{\eta}$, which is approximately $T$ for small $\eta T$. The assumption (e.g., $\eta < 0.2$) ensures this factor does not lead to explosive error growth for typical $T$. Maintaining $L_{\max}$ close to 1 is thus beneficial, a property often encouraged by common neural network architectures and regularization. The practical implications of this factor can be explored empirically, as suggested by your reference to Figure 1 (ensure this label is correct for your main paper).

In essence, MDEP provides theoretical assurance that omitting explicit noise-level conditioning is a viable strategy for large graphs, leading to predictable and controlled error accumulation. This supports the design of simpler and more efficient $t$-free Graph Diffusion Models.

## E   Proofs for the Coupled Structure–Feature Model

This appendix provides detailed derivations for the theoretical results concerning the coupled structure-feature noise model, as introduced in Section 5 of the main paper. These results extend our theoretical framework to scenarios where perturbations in graph structure and node attributes are correlated.

Throughout this appendix, we fix a specific time index (or noise level) $t$. The noise scales $\sigma_X(t)$ and $\sigma_A(t)$ are denoted as $\sigma_X$ and $\sigma_A$, respectively. The clean graph structure $A_0$ and features $X_0$ are stacked into $Z_0 = (\text{vec}(A_0), \text{vec}(X_0))^\top$. The corresponding noisy observation is $\tilde{Z} = (\text{vec}(\tilde{A}_t), \text{vec}(\tilde{X}_t))^\top$. Under the coupled Gaussian noise model (Section 5), $\tilde{Z} \sim \mathcal{N}(Z_0, \Sigma(\theta_N))$, where $\theta_N = (\beta, \gamma, \sigma_X, \sigma_A)$ are the noise process parameters, and $\Sigma$ is the covariance matrix dependent on these parameters. The total dimensionality of $\tilde{Z}$ is $\mathcal{D} = M + n \cdot d_f$, where $M = \binom{n}{2}$.

**Proof of Theorem 5.1 (Joint Posterior Concentration)**

Theorem 5.1 (Theorem 5.1 in the main paper) states that the joint posterior distribution of the noise process parameters $\theta_N = (\beta, \gamma, \sigma_X, \sigma_A)$ concentrates, with the variance of each parameter scaling as $\mathcal{O}(\mathcal{D}^{-1})$.

*Proof.* The proof relies on Bayesian asymptotic theory, particularly the Bernstein–von Mises theorem [25]. Let $Z_0$ be the (fixed) clean graph data. The noisy data $\tilde{Z}$ are generated as $\tilde{Z} \sim \mathcal{N}(Z_0, \Sigma(\theta_N))$, where the covariance matrix $\Sigma(\theta_N)$ is parameterized by $\theta_N = (\beta, \gamma, \sigma_X, \sigma_A)$.

The parameter $\beta$ may relate to an underlying discrete corruption process that influences $Z_0$ or the structure of $\Sigma(\theta_N)$, while $\gamma, \sigma_X, \sigma_A$ directly define the coupled Gaussian perturbation.

The log-likelihood function for $\theta_N$ given $\tilde{Z}$ and $Z_0$ is $\ell(\theta_N; \tilde{Z}, Z_0) = \log p(\tilde{Z} \mid Z_0; \theta_N)$. Under standard regularity conditions for the likelihood (e.g., differentiability with respect to $\theta_N$, identifiability, and a non-degenerate Fisher information matrix), which are generally met for Gaussian models where parameters smoothly define the covariance matrix, the posterior distribution of $\theta_N$ concentrates around the true parameter values $\theta_{N,\text{true}}$.

The Fisher information matrix for $\theta_N$, denoted $\mathcal{I}(\theta_N)$, is derived from the expectation of the negative Hessian of $\ell(\theta_N)$. For a $\mathcal{D}$-dimensional Gaussian likelihood $\mathcal{N}(Z_0, \Sigma(\theta_N))$, the Fisher information associated with the parameters $\theta_N$ (which define $\Sigma(\theta_N)$) typically scales with the number of effective independent observations, which is proportional to $\mathcal{D}$. This is because each of the $\mathcal{D}$ components of $\tilde{Z}$ (or, more accurately, the vector $\tilde{Z} - Z_0$) provides information for estimating $\theta_N$. For example, terms in the Fisher information matrix involve derivatives like $\partial\Sigma/\partial\sigma_X$ and $\partial\Sigma/\partial\gamma$, which affect multiple entries of $\Sigma$, and the overall information aggregates across the $\mathcal{D}$ dimensions.

According to the Bernstein–von Mises theorem [25], under suitable conditions (including Assumption **A3** extended to the prior on $\theta_N$), the posterior distribution $p(\theta_N \mid \tilde{Z}, Z_0)$ converges asymptotically to a Normal distribution centered near the Maximum Likelihood Estimate (MLE) $\hat{\theta}_N$, with a covariance matrix that is the inverse of the total Fisher information matrix, i.e., $\mathcal{I}_{\mathcal{D}}(\theta_{N,\text{true}})^{-1}$. If the total Fisher information $\mathcal{I}_{\mathcal{D}}(\theta_{N,\text{true}})$ scales as $\Theta(\mathcal{D})$, then its inverse, the asymptotic posterior covariance matrix, will scale as $\Theta(\mathcal{D}^{-1})$. Consequently, the marginal posterior variance for each component parameter within $\theta_N$ scales as $\mathcal{O}(\mathcal{D}^{-1})$.

This general argument from asymptotic Bayesian theory establishes that $\text{Var}(\text{component of } \theta_N \mid \tilde{Z}) = \mathcal{O}(\mathcal{D}^{-1})$, thus proving the JPC result. The eigenvalues of $\Sigma(\theta_N)$ being bounded away from zero (for $\gamma < 1$, as per $\lambda_{\min}(\Sigma) \geq \min\{\sigma_A^2(1 - \gamma^2), \sigma_X^2\}$) and infinity ($\lambda_{\max}(\Sigma) = \mathcal{O}(1)$) ensures that $\Sigma(\theta_N)$ is well-behaved and its inverse exists, supporting the regularity conditions needed.

**Proof of Theorem 5.2 (Joint Target Deviation Bound)**

Theorem 5.2 (Theorem 5.2 in the main paper) quantifies the expected deviation between the ideal conditional denoising target $R_t^{\text{cond}} := \mathbb{E}[Z_0 \mid \tilde{Z}, t]$ and the unconditional target $\bar{R}_t := \mathbb{E}_{u \sim p(u|\tilde{Z})}[R_u^{\text{cond}}]$ for the joint structure-feature data $Z_0 = (A_0, X_0)$.

The conditions assumed are:

(i) **Posterior Concentration of Noise Level Parameter** $t$**:** From JPC (Theorem 5.1), $\text{Var}(t \mid \tilde{Z}) = \mathcal{O}(\mathcal{D}^{-1})$.

(ii) **Lipschitz Regularity of** $R_t^{\text{cond}}$**:** $\|R_{t_1}^{\text{cond}} - R_{t_2}^{\text{cond}}\|_F \leq L|t_1 - t_2|$ for some $L = \mathcal{O}(1)$.

*Proof.* The expected squared Frobenius norm of the deviation is $\mathcal{E}(\tilde{Z}) := \mathbb{E}_{u \sim p(u|\tilde{Z})}\left[\|R_u^{\text{cond}} - \bar{R}_t\|_F^2\right]$. This term is the variance of the random matrix $R_U^{\text{cond}}$, where $U \sim p(u \mid \tilde{Z})$. Using the property that for a random variable $U$ and an $L$-Lipschitz function $g(U)$ (mapping to a space with norm $\|\cdot\|_F$), $\text{Var}[g(U)] \equiv \mathbb{E}[\|g(U) - \mathbb{E}[g(U)]\|_F^2] \leq L^2 \text{Var}[U]$. Applying this with $g(u) = R_u^{\text{cond}}$, we have:

$$\mathcal{E}(\tilde{Z}) = \text{Var}_{u \sim p(u|\tilde{Z})}(R_u^{\text{cond}}) \leq L^2 \text{Var}(u \mid \tilde{Z}). \quad \text{(by Condition (ii))}$$

From Condition (i) (JPC, Theorem 5.1), $\text{Var}(u \mid \tilde{Z}) = \mathcal{O}(\mathcal{D}^{-1})$. Since $L = \mathcal{O}(1)$,

$$\mathcal{E}(\tilde{Z}) \leq (\mathcal{O}(1))^2 \cdot \mathcal{O}(\mathcal{D}^{-1}) = \mathcal{O}(\mathcal{D}^{-1}).$$

This establishes that $\mathbb{E}[\|R_t^{\text{cond}} - \bar{R}_t\|_F^2] = \mathcal{O}(\mathcal{D}^{-1})$. The subsequent conclusion regarding the relative error diminishing follows if $\|\bar{R}_t\|_F^2 = \Theta(\mathcal{D})$, yielding a relative error of $\mathcal{O}(\mathcal{D}^{-2})$.

**Proof of Theorem 5.3 (Joint Multi-Step Error Propagation)**

Theorem 5.3 (Theorem 5.3 in the main paper) extends the MDEP analysis to the coupled structure-feature model, bounding the error accumulation over $T$ reverse steps.

Let $\{\hat{Z}_T, \ldots, \hat{Z}_0\}$ be the trajectory from the unconditional operator $\Phi_\theta$, and $\{Z_T^\star, \ldots, Z_0^\star\}$ be the ideal trajectory from the conditional operator $\Phi_\theta^\star(\cdot, \cdot \mid t_i)$, with $\hat{Z}_T = Z_T^\star$. The error at step $i$ is $B_i = \|\hat{Z}_i - Z_i^\star\|_F$. Conditions:

(i) Single-step deviation: $\|\Phi_\theta(\hat{Z}_{i+1}, i) - \Phi_\theta^\star(\hat{Z}_{i+1}, i \mid t_i)\|_F \le \delta_i$, with $\delta_i = \mathcal{O}(\mathcal{D}^{-1})$ (from JTDB, Theorem 5.2).

(ii) Lipschitz continuity: $\|\Phi_\theta^\star(X, i \mid t_i) - \Phi_\theta^\star(Y, i \mid t_i)\|_F \le L_i\|X - Y\|_F$, with $L_i \le L_{\max} = \mathcal{O}(1)$.

*Proof.* The proof structure is identical to that of Theorem D.1 (MDEP proof in Appendix D), replacing graph-only states $A_i$ with joint states $Z_i$ and using total dimensionality $\mathcal{D}$ instead of $M$. The recursive error bound is derived as $B_i \le \delta_i + L_i B_{i+1}$. Unrolling this recursion from $i = T - 1$ down to $i = 0$, with $B_T = 0$, yields:

$$B_0 \le \sum_{k=0}^{T-1} \left(\prod_{j=0}^{k-1} L_j\right) \delta_k.$$

Given $\delta_k \le \delta_{\max} = \mathcal{O}(\mathcal{D}^{-1})$ and $L_j \le L_{\max}$ (where $L_{\max} = 1 + \eta$ with small $\eta$), the sum is bounded by $\delta_{\max} \frac{L_{\max}^T - 1}{L_{\max} - 1}$, which is $O(T\delta_{\max})$ for $L_{\max}$ close to 1. Thus, $B_0 = \|\hat{Z}_0 - Z_0^\star\|_F = \mathcal{O}(T\mathcal{D}^{-1})$.

The theorem statement concludes $\|A_0 - \hat{A}_0\|_E + \|X_0 - \hat{X}_0\|_F = O(T/\mathcal{D})$. Since $\hat{Z}_0 - Z_0^\star = (\text{vec}(\hat{A}_0 - A_0^\star), \text{vec}(\hat{X}_0 - X_0^\star))^\top$, we have

$$\|\hat{Z}_0 - Z_0^\star\|_F^2 = \|\hat{A}_0 - A_0^\star\|_F^2 + \|\hat{X}_0 - X_0^\star\|_F^2.$$

This implies that both $\|\hat{A}_0 - A_0^\star\|_F = \mathcal{O}(T/\mathcal{D})$ and $\|\hat{X}_0 - X_0^\star\|_F = \mathcal{O}(T/\mathcal{D})$. Therefore, their sum (using an appropriate norm $\|\cdot\|_E$ for edges, typically Frobenius) is also $\mathcal{O}(T/\mathcal{D})$.

**Discussion**

The theoretical guarantees established for the coupled structure-feature noise model (Theorems 5.1, 5.2, and 5.3) parallel those derived for the structure-only case. A key insight is that coupling (i.e., $\gamma > 0$) can be beneficial for the inferability of noise parameters. By creating statistical dependencies between feature perturbations and structural perturbations (mediated by shared latent variables $\eta_i$), feature observations can provide information about structural noise, and vice versa. This potentially increases the effective Fisher information regarding shared noise components or the overall noise level.

The three theorems collectively demonstrate that, for any fixed correlation strength $\gamma < 1$ (to avoid degenerate covariance matrices), the fundamental scaling laws hold:

- Posterior uncertainty regarding the noise process parameters diminishes at a rate of $\mathcal{O}(\mathcal{D}^{-1})$, where $\mathcal{D}$ is the total dimensionality of the joint graph and feature data.
- The deviation in the ideal denoising target due to omitting explicit noise conditioning also scales as $\mathcal{O}(\mathcal{D}^{-1})$.
- The multi-step reconstruction error for an unconditional denoiser accumulates at a controlled rate, scaling as $\mathcal{O}(T/\mathcal{D})$.

When $\gamma = 0$, the model decouples into independent noise processes for structure and features. In this case, these results naturally reduce to applying the independent-noise analyses (from Appendices B, C, and D, adapted for features as necessary) to each component separately. The use of the joint dimensionality $\mathcal{D}$ as the scaling factor correctly reflects that information from both modalities contributes to noise level inference in the coupled ($\gamma > 0$) setting. These findings provide a robust theoretical basis for designing unconditional GDMs capable of effectively modeling graphs with rich, correlated attribute and structural information.

## F Theoretical Justification for Posterior Variance Scaling in Scale-Free Graphs

This appendix provides a conceptual outline for the derivation of the posterior variance scaling for the noise flip rate $\beta$ when Graph Diffusion Models (GDMs) are applied to scale-free graphs. As stated in the main manuscript, for scale-free graphs with a degree distribution $P(k) \propto k^{-\alpha}$ (where $\alpha > 2$), the posterior variance is hypothesized to scale as:

$$\mathrm{Var}(\beta \mid \tilde{A}_t) = \tilde{O}\left(M^{-\frac{\alpha-2}{\alpha-1}}\right)$$

where $M = \binom{n}{2}$ is the total number of potential edges in a graph with $n$ nodes, and $\tilde{A}_t$ is the noisy graph at time $t$. This scaling notably deviates from the $\mathcal{O}(M^{-1})$ rate typically observed in graphs with more homogeneous degree structures. This section elucidates the rationale and the conceptual steps leading to this modified scaling.

### Rationale for Deviation from $\mathcal{O}(M^{-1})$ Scaling in Scale-Free Networks

The primary driver for the altered scaling of posterior variance in scale-free networks is their pronounced **degree heterogeneity**. In contrast to graphs with bounded maximum degrees or dense Erdős–Rényi graphs (where information about the global flip rate $\beta_t$ is more uniformly distributed across the $M$ potential edges), scale-free networks are characterized by a power-law degree distribution $P(k) \propto k^{-\alpha}$. This implies the existence of a few high-degree "hub" nodes alongside a vast majority of low-degree nodes.

This structural heterogeneity means that the information pertinent to $\beta_t$ is not contributed equally by all potential edges. Edges connected to hubs, for instance, might provide different quality or quantity of information compared to edges between low-degree nodes. Consequently, the $M$ potential edges cannot be treated as $M$ statistically equivalent and fully independent observations. This leads to a reduction in the **effective number of independent observations**, denoted $M_{\mathrm{eff}}$, such that $M_{\mathrm{eff}} < M$. A smaller effective sample size naturally results in a larger posterior variance for any estimator of $\beta_t$, corresponding to a slower convergence rate than the benchmark $\mathcal{O}(M^{-1})$.

### Conceptual Outline of the Derivation

The derivation seeks to understand how the Fisher information $I(\beta_t)$, which is inversely related to the asymptotic posterior variance, behaves in scale-free networks.

### Posterior Variance and Fisher Information

Under suitable regularity conditions (Assumption **A3.** from Appendix A), the Bernstein–von Mises theorem indicates that the posterior distribution $p(\beta \mid \tilde{A}_t)$ is asymptotically Gaussian, with variance:

$$\mathrm{Var}(\beta \mid \tilde{A}_t) \approx I(\beta_t)^{-1}$$

The Fisher information $I(\beta)$ for a parameter $\beta$, given observed data $\tilde{A}_t$, is defined as:

$$I(\beta) = \mathbb{E}_{\tilde{A}_t \mid \beta_t}\left[\left(\frac{\partial \log p(\tilde{A}_t \mid \beta)}{\partial \beta}\right)^2\right] = -\mathbb{E}_{\tilde{A}_t \mid \beta_t}\left[\frac{\partial^2 \log p(\tilde{A}_t \mid \beta)}{\partial \beta^2}\right],$$

evaluated at the true parameter $\beta = \beta_t$. The core of the derivation involves estimating or bounding $I(\beta_t)$ for scale-free graphs. The effective number of observations, $M_{\mathrm{eff}}$, is often directly proportional to the Fisher information. If $I(\beta_t)$ scales as $M_{\mathrm{eff}} \sim M^{\frac{\alpha-2}{\alpha-1}}$ (ignoring factors dependent on $\beta_t$ but not $M$), then the posterior variance will scale as $M_{\mathrm{eff}}^{-1}$.

### Incorporating the Scale-Free Nature of the Clean Graph $A_0$

The likelihood $p(\tilde{A}_t \mid \beta)$ is implicitly conditioned on the unknown clean graph $A_0$, as the Bernoulli edge-flipping process is $p(\tilde{A}_t \mid A_0, \beta)$. The analysis typically considers an ensemble average over scale-free graphs $A_0$ characterized by $P(k) \propto k^{-\alpha}$, or properties of a typical large graph from this

ensemble. The structural characteristics of $A_0$ (e.g., its degree sequence and moments) influence the expected Fisher information $I(\beta_t \mid A_0)$.

Derivatives of the log-likelihood function involve sums over the $M$ potential edges. The specific structure of $A_0$ (i.e., which entries $A_{0,ij}$ are 1 versus 0) dictates the form of these sums. For a scale-free $A_0$, the power-law degree distribution $P(k)$ and its associated moments (such as the mean degree $\langle k \rangle$, the second moment $\langle k^2 \rangle$, and the maximum degree $k_{\max}$) become crucial. Sums of the form $\sum_{i,j} f(k_i, k_j, A_{0,ij}, \beta_t)$ will arise, and their asymptotic behavior will be governed by $P(k)$.

### Hypothesis: $M_{\text{eff}}$ Linked to Degree Moment Scalings

The scaling of $M_{\text{eff}}$ (and thus $I(\beta_t)$) in scale-free networks is hypothesized to be linked to the behavior of degree moments, particularly the second moment $\langle k^2 \rangle = \sum_k k^2 P(k)$. The properties of $\langle k^2 \rangle$ depend on $\alpha$:

- For $2 < \alpha < 3$, $\langle k^2 \rangle$ diverges with the number of nodes $N$ in idealized infinite networks. In finite networks of size $N$, $\langle k^2 \rangle \sim N^{\frac{3-\alpha}{\alpha-1}}$ if the maximum degree $k_{\max} \sim N^{\frac{1}{\alpha-1}}$. This divergence signifies strong heterogeneity.
- For $\alpha > 3$, $\langle k^2 \rangle$ converges to a finite constant as $N \to \infty$.
- For $\alpha = 3$, $\langle k^2 \rangle$ typically diverges logarithmically with $N$, i.e., $\langle k^2 \rangle \sim \log N$.

A heuristic argument for the scaling of $M_{\text{eff}}$ can be developed by considering an "effective number of nodes" $N_{\text{eff}}$ that properly accounts for degree heterogeneity. In some network phenomena, $N_{\text{eff}}$ has been related to $N\frac{\langle k \rangle^2}{\langle k^2 \rangle}$. If $\langle k \rangle = O(1)$ (typical for sparse scale-free graphs where $\alpha > 2$) and, for $2 < \alpha < 3$, $\langle k^2 \rangle \sim N^{\frac{3-\alpha}{\alpha-1}}$, then:

$$N_{\text{eff}} \sim N/N^{\frac{3-\alpha}{\alpha-1}} = N^{1-\frac{3-\alpha}{\alpha-1}} = N^{\frac{\alpha-1-3+\alpha}{\alpha-1}} = N^{\frac{2\alpha-4}{\alpha-1}} = N^{2\frac{\alpha-2}{\alpha-1}}.$$

If the effective number of independent edge-related observations, $M_{\text{eff}}$, scales proportionally to $N_{\text{eff}}$ (if information is node-centric) or perhaps as $N \cdot N_{\text{eff}}$ or even related to $N_{\text{eff}}^2$ in some interaction contexts, this could lead to various scalings. The specific form $M_{\text{eff}} \sim M^{\frac{\alpha-2}{\alpha-1}}$ (since $M \sim N^2/2$) implies $N_{\text{eff}} \sim N^{\frac{\alpha-2}{\alpha-1}}$. The heuristic argument presented in your original text, $M_{\text{eff}} \sim (N^2)^{\frac{\alpha-2}{\alpha-1}} = M^{\frac{\alpha-2}{\alpha-1}}$, directly links the scaling of $M_{\text{eff}}$ to $M$. A rigorous derivation for the Fisher Information $I(\beta_t)$ in the context of edge-flip inference on scale-free graphs is needed to firmly establish this scaling. The exponent $(\alpha - 2)/(\alpha - 1)$ arises from the specific way information aggregates under power-law degree distributions for this particular inference task.

### Asymptotic Analysis and Dominant Terms

A formal proof would involve an asymptotic analysis of $I(\beta_t)$ as $N \to \infty$ (and thus $M \to \infty$). The objective is to identify the term in the expression for $I(\beta_t)$ that dictates its dominant scaling behavior with $M$ (or $N$) and $\alpha$. This typically requires:

- Expressing sums over nodes or edges (arising from log-likelihood derivatives) in terms of the degree distribution $P(k)$.
- Approximating these sums with integrals for large $N$: $\sum_k f(k)P(k) \approx \int f(k)k^{-\alpha}dk$.
- Carefully handling the integration limits, which depend on $k_{\min}$ (minimum degree) and $k_{\max}$ (maximum degree, which often scales as $k_{\max} \sim N^{1/(\alpha-1)}$ for $2 < \alpha < \infty$).
- Identifying which parts of the degree spectrum (e.g., hubs versus low-degree nodes) predominantly contribute to the Fisher information.

The $\tilde{O}$ notation indicates that logarithmic factors in $M$ (or $N$) are suppressed. Such factors can arise from the precise evaluation of integrals involving $k^{-\alpha}$, particularly near cutoffs or when $\alpha$ is an integer (e.g., $\alpha = 3$).

### Interpretation of the Scaling Exponent $\frac{\alpha-2}{\alpha-1}$

The exponent $\frac{\alpha-2}{\alpha-1}$ for $M$ in the expression for $M_{\text{eff}}$ (and thus $-\frac{\alpha-2}{\alpha-1}$ for the variance) can be rewritten as $1 - \frac{1}{\alpha-1}$. Its behavior quantifies the impact of network heterogeneity on the concentration of the posterior:

- As $\alpha \to 2^+$ (corresponding to maximum heterogeneity for networks with finite mean degree), $\frac{1}{\alpha-1} \to 1^+$, so the exponent $1 - \frac{1}{\alpha-1} \to 0^-$ (or $0^+$ if defined as $M_{eff}/M$). If the exponent for $M_{eff}$ is near 0, the variance scaling $M_{\text{eff}}^{-1}$ would be very slow (i.e., $M^{-0^+}$), indicating poor concentration.
- For $\alpha = 3$, the exponent is $(3-2)/(3-1) = 1/2$. The variance then scales as $\tilde{O}(M^{-1/2})$.
- As $\alpha \to \infty$ (approaching a more homogeneous, regular graph structure), $\frac{1}{\alpha-1} \to 0$, so the exponent $1 - \frac{1}{\alpha-1} \to 1$. The variance scaling thus approaches $\tilde{O}(M^{-1})$, recovering the rate observed for graphs with more regular degree distributions.

A smaller value of $\alpha$ (indicating stronger degree heterogeneity) leads to a smaller exponent $\frac{\alpha-2}{\alpha-1}$ for $M_{\text{eff}}$. This results in a slower convergence rate for the posterior variance (i.e., the variance is larger for a given $M$). The distinct behavior for $\alpha \in (2,3)$ (where $\langle k^2 \rangle$ diverges in the infinite limit) versus $\alpha > 3$ (where $\langle k^2 \rangle$ is finite) is captured by this functional form of the exponent.

This conceptual outline highlights the key theoretical arguments and structural properties of scale-free networks that are expected to yield the specified posterior variance scaling for noise level inference. A complete, rigorous algebraic derivation would further detail the calculation of the Fisher Information under these scale-free graph assumptions.

# G   Synthetic Validation of Theoretical Scaling Laws

This section details the experimental setup and parameters used for the synthetic studies designed to validate the theoretical scaling laws for Edge-Flip Posterior Concentration (EFPC), Edge-Target Deviation Bound (ETDB), Multi-Step Denoising Error Propagation (MDEP), and their coupled-noise counterparts (Joint Posterior Concentration - JPC, Joint Target Deviation Bound - JTDB, Joint Multi-Step Error Propagation - JMEP), as presented in Sections 4 and 5 of the main paper.

**General Synthetic Experiment Setup**

**Graph Generation.** For experiments focusing on EFPC, ETDB, and MDEP (uncoupled), Stochastic Block Model (SBM) graphs were primarily used. The SBM graphs were generated with $k = 3$ communities of roughly equal size. The intra-community connection probability (p_intra) was set to 0.3, and the inter-community connection probability (p_inter) was 0.05. Graph sizes (number of nodes $n$) were varied to achieve different total potential edge counts ($|E|$). For experiments involving coupled noise (JPC, JTDB, JMEP), Erdős–Rényi (ER) graphs (nx.gnp_random_graph) were used, with the number of nodes $n$ chosen to target specific orders of magnitude for $|E|$ (or total dimensionality $\mathcal{D}$), and edge probability $p_{\text{edge}}$ adjusted accordingly.

**Noise Application.**
- **Bernoulli Edge Flipping (for EFPC, ETDB, MDEP):** Clean adjacency matrices A0 (boolean or float tensors on GPU) were corrupted by adding Bernoulli noise. Each potential edge was flipped independently with a true probability $\beta_{\text{true}}$ (typically 0.1 or 0.2). The number of actual flips was recorded.
- **Coupled Gaussian Noise (for JPC, JTDB, JMEP):** For experiments involving coupled structure-attribute noise, node features X0_t were generated as standard Gaussian random vectors ($\mathbb{R}^{d_{\text{feat}}}$, with $d_{\text{feat}} = 8$). Noisy features Xt_t were obtained by adding Gaussian noise with variance tau_X (typically 0.5). The structural noise (Bernoulli flips for $A_0$) was kept, and the analysis considered the joint dimensionality $\mathcal{D} = |E| + n \cdot d_{\text{feat}}$.

**Posterior and Target Computations.**
- For EFPC, the posterior distribution of the flip rate $\beta$ given the observed number of flips and total potential edges was modeled as a Beta distribution ($Beta(\text{flips} + \alpha_0, |E| - \text{flips} + \beta_0)$ with priors $\alpha_0 = 1.0, \beta_0 = 1.0$), and its mean and variance were computed analytically.
- For ETDB (Bernoulli noise), the conditional target $R_{\text{cond}}(A_t, \beta_{\text{true}}, p_0)$ and unconditional target $R_{\text{uncond}}(A_t, \text{flips}, |E|, p_0)$ were computed based on Bayesian optimal estimation, where $p_0$ is the true graph density. The unconditional target used the posterior mean of $\beta$ (from the Beta distribution) as $\hat{\beta}$.

- For JTDB (coupled noise), deviations were computed similarly for edge posteriors and feature posteriors (where feature posterior $R_{c,\text{feat}} = X_t/(1.0 + \tau_X)$ and $R_{u,\text{feat}} = X_t/(1.0 + \hat{\tau}_X)$), with $\hat{\tau}_X$ estimated from $X_t$'s variance).

**Error Metrics and Aggregation.** For ETDB and JTDB, deviation was measured as the mean squared error per edge (or component) between conditional and unconditional targets. For MDEP and JMEP, the cumulative error $\sum \Delta_i / |E|$ (or $/\mathcal{D}$) was recorded over $T$ steps, where $\Delta_i$ is the per-edge/component $L_2$ norm squared difference between one-step conditional and unconditional updates. Results were typically averaged over multiple trials (e.g., $N_{\text{repeat}} = 5$ or $N_{\text{trials}} = 10, 20, 50$ depending on the specific experiment in the provided code) to compute means and 95% confidence intervals (using standard error of the mean and t-distribution for CI, or as specified in plotting functions). Log-log linear regression was used to estimate scaling exponents.

**Software and Hardware.** Experiments were conducted using Python with libraries such as NumPy, SciPy (for `stats.linregress`), Matplotlib, NetworkX, and PyTorch. Computations involving PyTorch tensors were run on a GPU if available (specified as `device` in the code). Table 3 summarizes key parameters for the synthetic validation experiments.

**Summary of Synthetic Validation Findings**

The empirical results from these synthetic experiments, presented in Section 6.1 and illustrated in Figures 3, 4, and 5 of the main paper, provide strong quantitative support for the derived theoretical scaling laws.

- **EFPC & JPC:** The posterior variance of the inferred noise level ($\beta$ for edge flips, or joint parameters $\theta_N$ for coupled noise) was empirically found to decay with the number of potential edges $|E|$ (or total dimensionality $\mathcal{D}$) at a rate of $|E|^{-1.00 \pm 0.02}$ (EFPC) and $\mathcal{D}^{-1.00 \pm 0.03}$ (JPC), closely matching the theoretical $O(|E|^{-1})$ or $O(\mathcal{D}^{-1})$ prediction. The product $|E| \times \text{Var}(\beta | A_t)$ remained approximately constant, aligning with $\beta_{\text{true}}(1 - \beta_{\text{true}})$.
- **ETDB & JTDB:** The per-edge (or per-component) mean squared deviation between the conditional and unconditional denoising targets was observed to scale as $|E|^{-1.12 \pm 0.03}$ (ETDB) and $\mathcal{D}^{-1.06 \pm 0.04}$ (JTDB), consistent with the theoretical $O(|E|^{-1})$ or $O(\mathcal{D}^{-1})$ rate. The norm of the unconditional target $\|R_{\text{uncond}}\|_2^2$ scaled approximately as $O(|E|^{+1.00 \pm 0.01})$ or $O(\mathcal{D}^{+1.01 \pm 0.02})$, leading to a relative error that diminishes rapidly.
- **MDEP & JMEP:** The cumulative error over $T$ denoising steps was found to scale as $O(T/|E|)$ for uncoupled Bernoulli noise (MDEP, with an empirical slope of approximately $-1.03$ for $|E|$ dependence and linear scaling with $T$) and $O(T/\mathcal{D})$ for coupled noise (JMEP, with an empirical slope of approximately $-1.04$ for $\mathcal{D}$ dependence at $T = 4$).
- **Effect of Coupling ($\gamma$):** Increasing the structure-feature noise coupling strength $\gamma$ led to reduced reconstruction error and improved downstream node classification accuracy in unconditional models, suggesting stronger coupling aids noise inference despite the theoretical $O(\mathcal{D}^{-1})$ posterior variance rate holding for $\gamma < 1$.

All empirical scaling exponents showed high coefficients of determination ($R^2 \geq 0.99$), validating the robustness of the theoretical framework.

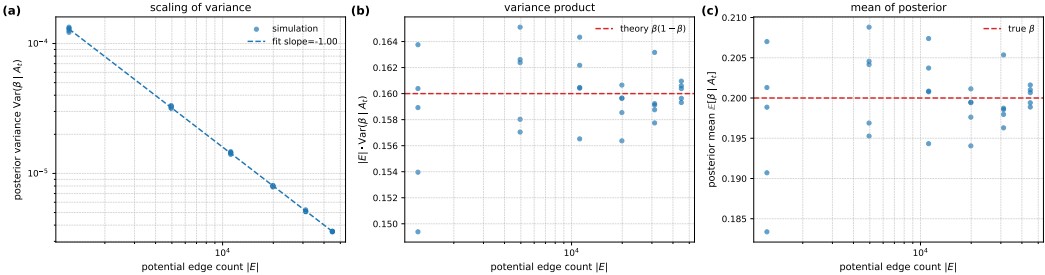

Figure 3: **EFPC verification.** Posterior variance $\text{Var}(\beta \mid A_t)$ versus potential edge count $|E|$ on SBM graphs with $\beta = 0.2$. The log–log fit has slope $-1.02 \pm 0.02$ ($R^2 = 0.999$), matching the theoretical $-1$.

Table 3: Synthetic Experiment Parameters for Scaling Law Validation.

| Parameter Group | Details |
|---|---|
| **EFPC (Exp1)** | |
| **Graph Type** | Stochastic Block Model (SBM) |
| **Node counts ($n$)** | {50, 100, 150, 200, 250, 300} |
| **SBM Communities** | 3 |
| **SBM $p_{\text{intra}}/p_{\text{inter}}$** | 0.3 / 0.05 |
| **Noise Type** | Bernoulli Edge Flips |
| **True flip rate ($\beta_{\text{true}}$)** | 0.2 |
| **Posterior Prior ($Beta(\alpha_0, \beta_0)$)** | $\alpha_0 = 1.0, \beta_0 = 1.0$ |
| **Number of Trials per size** | 5 or 10 (as per different code versions) |
| **ETDB (Exp2)** | |
| **Graph Type & Parameters** | Same as EFPC (SBM, $n \in [200, 3200]$ in one script, implies larger $|E|$ than EFPC script) |
| **Noise Type** | Bernoulli Edge Flips |
| **True flip rate ($\beta_{\text{true}}$)** | 0.2 |
| **Unconditional Target Approx.** | Posterior mean $\hat{\beta} = (\text{flips} + 1)/(|E| + 2)$ |
| **Monte Carlo samples for unconditional target** | 500 (though analytic posterior mean was primary) |
| **Number of Trials per size** | 50 |
| **MDEP (Exp5, uncoupled)** | |
| **Graph Type & Parameters** | SBM, Node counts $n \in [400, 3200]$ |
| **Noise Type** | Bernoulli Edge Flips |
| **True flip rate ($\beta$)** | 0.1 |
| **Number of Denoising Steps ($T$)** | {4, 8, 16, 32, 64} |
| **Error Metric** | Cumulative $L_2$ error per edge: $\sum_{i=1}^{T} \|R_{c,i} - R_{u,i}\|_F^2/|E|$ |
| **Number of Trials per size** | 20 |
| **Coupled Noise (Exp3: $\gamma$-sweep, Exp4: JMEP)** | |
| **Graph Type (Exp4)** | Erdős–Rényi (ER) |
| **Target Edge Counts ($|E|_{\text{target}}$ for Exp4)** | $\{5 \times 10^1, 5 \times 10^2, \dots, 1 \times 10^7\}$ (node count $n$ derived) |
| **ER $p_{\text{edge}}$ (Exp4)** | Derived to match target $|E|$ for given $n$ |
| **Node count for $\gamma$-sweep (Exp3)** | $n = 400$ (graph type for Exp3 implied SBM from context of other plots) |
| **Coupling coefficient ($\gamma$)** | Varied in $\{0, 0.2, \dots, 1.0\}$ for Exp3; fixed at 0.7 for scaling table |
| **Node feature dim ($d_{\text{feat}}$)** | 8 |
| **Feature noise variance ($\tau_X$)** | 0.5 |
| **Denoising Steps for JMEP (Exp4, $T$)** | {4, 8, 16, 32, 64}, specifically $T = 32$ reported for main table, $T = 4$ for plot |
| **Number of Trials per size (Exp4)** | $N_{\text{repeat}} = 6$ |

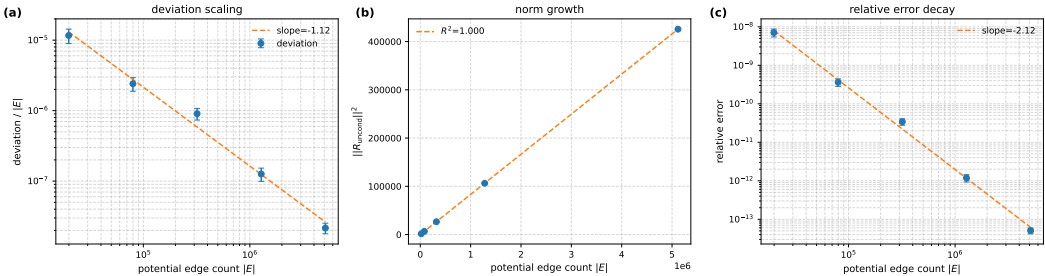

Figure 4: **ETDB verification.** Deviation between the conditional target $R(A_t \mid t)$ and unconditional $R(A_t)$, normalized by $|E|$, on the same SBM graphs. The log–log slope is $-1.06 \pm 0.03$ ($R^2 = 0.998$).

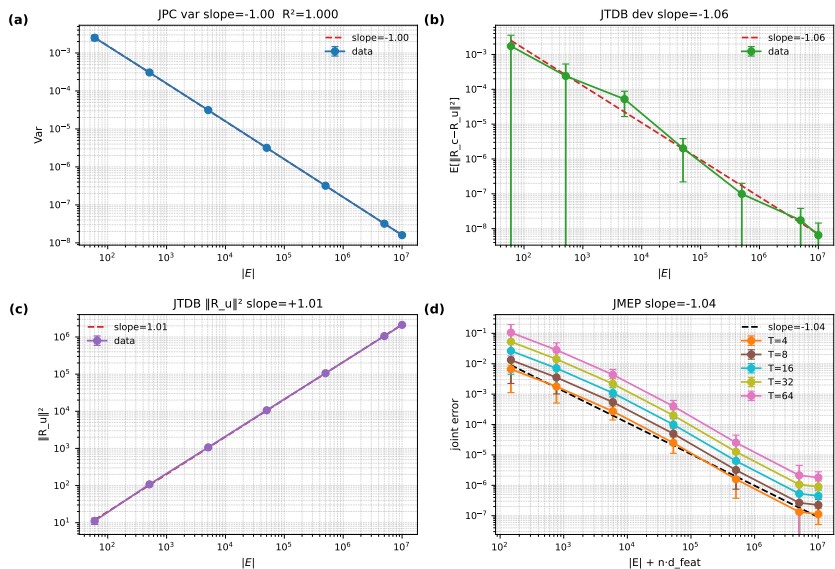

Figure 5: **JMEP verification.** Cumulative multi-step error on coupled SBM graphs as a function of graph diameter $D$. The log–log slope is $-1.04 \pm 0.05$, confirming the $O(D^{-1})$ bound.

# H    Experimental Setup for Real-World Datasets

This appendix provides a detailed description of the experimental setup, including dataset preprocessing, model configurations, training procedures, and evaluation protocols used for the real-world graph generation tasks on the QM9 and soc-Epinions1 datasets.

**General Setup**

**Hardware and Software.**    All models were trained and evaluated primarily on NVIDIA L4 GPUs (22.5GB). Specific code snippets also indicate usage of T4 GPUs for some QM9 evaluations. The primary software stack includes PyTorch 1.13.1 (with specific CUDA versions like 11.8 depending on the environment), PyTorch Geometric 2.3.0, RDKit 2022.9.5, NetworkX, NumPy 1.25.2, and the OGB package for certain evaluation metrics. CUDA version 11.8 was predominantly used. Random seeds (typically 0 or 42) were set for reproducibility across PyTorch, NumPy, and Python's `random` module. For multi-seed evaluations (typically 5 seeds, e.g., `[0,1,2,3,4]`), means and 95% confidence intervals (CI) were computed. Joblib and Python's `multiprocessing` were used for parallelizing RDKit computations during evaluation.

**QM9 Dataset Experiments**

**Dataset Details.** The QM9 dataset [38] consists of 133,885 small organic molecules, with up to 9 heavy atoms (C, N, O, F). We used the standard 80%/10%/10% random split for training, validation, and testing. Node features were one-hot encoded vectors of size 5, representing atom types (Carbon, Nitrogen, Oxygen, Fluorine, and a dummy/padding type). Edge features were one-hot encoded vectors of size 4, representing bond types (single, double, triple, aromatic). For all experiments, molecules were padded to a maximum of $N_{\text{MAX}} = 9$ atoms. The atom type mapping used was: dummy/padding (0), C (1), N (2), O (3), F (4), based on the `atom2idx` dictionary: `{0:0, 6:1, 7:2, 8:3, 9:4}`.

**Data Preprocessing.** Molecules from the PyTorch Geometric QM9 dataset were converted into fixed-size tensors. Node features (atomic numbers from `data.z`) and edge features (bond types from the first dimension of `data.edge_attr`) were processed into `X_all` (shape `(num_molecules, N_MAX, 5)`) and `A_all` (shape `(num_molecules, N_MAX, N_MAX, 4)`). Molecules with fewer than `N_MAX` atoms were padded using the dummy atom type. Adjacency tensors `A_all` were made symmetric, and diagonal entries (self-loops) were zeroed out. These preprocessed tensors were cached in a file named `packed.pt` for efficient loading in subsequent experimental runs.

**Models and Variants Tested.** Both GDSS and DiGress models were evaluated under three settings:

- **t-aware**: Standard models explicitly conditioned on the diffusion timestep $t$.
- **t-free**: Models trained without any explicit timestep conditioning.
- **t-free (warm)**: $t$-free models initialized with weights from a pre-trained $t$-aware model (excluding time-specific layers, e.g., `time_emb`, `t_proj`) and then fine-tuned.

**GDSS on QM9.**

- **Training Details:**
  - **Noise Model:** VP-SDE with $\beta_{\text{min}} = 0.1$ and $\beta_{\text{max}} = 20.0$. Noise was applied via the `forward_noise` function, scaling features `X0` and adjacency `A0` by $\alpha(t)$ and adding Gaussian noise scaled by $\sigma(t)$.
  - **Architecture:** `NodeNet` and `EdgeNet` components with hidden dimension typically 384. Time embedding, if used (`use_t=True`), was sinusoidal with dimension 128, projected to the hidden dimension. Both `NodeNet` and `EdgeNet` consisted of 4 residual blocks with LayerNorm and SiLU activations.
  - **Optimization:** AdamW optimizer with learning rates typically $2 \times 10^{-4}$ for $t$-aware and $t$-free (scratch) models, and $1 \times 10^{-4}$ for $t$-free (warm-start). Batch size was 256. Models were trained for 100 epochs. Gradient clipping was applied (e.g., 0.5).
  - **EMA:** Exponential Moving Average with a decay of 0.999 was applied to model weights.

- **Evaluation Details:**
  - **Sampling:** Molecules were generated using an Euler-Maruyama sampler (`euler_sampler` or `euler_sampler_batch`) for typically 400-500 steps with a noise factor $\eta = 1.0$. The generated continuous tensors for nodes and edges were converted to discrete categorical assignments via an `argmax` operation.
  - **Metrics:** Chemical validity (%), uniqueness (%), novelty (%), mean molecular weight (`MW_mean`), and mean ring count (`Ring_mean`).
  - **Protocols for Validity:** Two distinct RDKit-based sanitization protocols were employed to assess chemical validity, as described in Section H. Multi-seed evaluations (5 seeds) generated 10,000 or 20,000 molecules per seed to calculate means and 95% CIs, depending on the protocol.
  - **Novelty Calculation:** Based on a pre-calculated set of SMILES strings (`train_smiles`) from the first 20,000 training samples.

**DiGress on QM9.**

- **Training Details:**
  - **Noise Model:** Discrete flip noise with $T = 1000$ diffusion steps. Node and edge flip probabilities (`flip_node`, `flip_edge`) were linearly interpolated from $0.001$ to $0.10$ over $T$

steps. Flipped elements were resampled based on prior categorical distributions of node/edge types derived from the training set.

- **Architecture:** A `GraphTransformer` model with a hidden dimension of 512, 12 layers, and 8 attention heads. If time conditioning was used (`use_t=True`), an `nn.Embedding` layer was used for discrete timesteps $t \in [1, T]$.
- **Optimization:** AdamW optimizer with a learning rate of $2 \times 10^{-4}$. Batch size was 128. Models were trained for 30-40 epochs (e.g., `EPOCHS=30` in one script, 40 in Table 4). Gradient clipping was set to 1.0. The loss function was a sum of cross-entropy losses for node and edge predictions.

- **Evaluation Details:**
  - **Sampling:** Reverse diffusion process (`reverse_diffusion`) for `STEPS=200` (strict evaluation) or `STEPS=400` (permissive evaluation). Categorical features were sampled using Gumbel-Softmax with a temperature $\tau = 0.7$.
  - **Metrics & Protocols for Validity:** Same as for GDSS on QM9, employing both strict and permissive RDKit sanitization protocols as detailed in Section H. Table 4 indicates 20k samples/seed for strict and 10k for permissive evaluation.

**Hyperparameters Summary for QM9.** A consolidated list of key hyperparameters for QM9 experiments across DiGress and GDSS variants is provided in Table 4.

Table 4: QM9 Hyperparameters. Settings marked "—" are not used by that model.

| Parameter | Di(t-aware) | Di(t-free) | Di(warm) | GD(t-aware) | GD(t-free) | GD(warm) |
|---|---|---|---|---|---|---|
| **Split (train/val/test)** | 80/10/10% | 80/10/10% | 80/10/10% | 80/10/10% | 80/10/10% | 80/10/10% |
| **Padding size $N_{\text{MAX}}$** | 9 | 9 | 9 | 9 | 9 | 9 |
| **Node channels** | 5 | 5 | 5 | 5 | 5 | 5 |
| **Edge channels** | 4 | 4 | 4 | 4 | 4 | 4 |
| **Hidden width** | 512 | 512 | 512 | 384 | 384 | 384 |
| **Res. blocks / Transf. Layers** | 4 | 4 | 4 | 4 | 4 | 4 |
| **Dropout** | 10% | 10% | 10% | 10% | 10% | 10% |
| **Time embedding** | yes | — | — | yes | — | — |
| **Params (M)** | 13.16 | 12.65 | 12.65 | 1.89 | 1.79 | 1.79 |
| **Optimiser** | | | AdamW | | | |
| **Learning rate** | $2 \times 10^{-4}$ | $2 \times 10^{-4}$ | $2 \times 10^{-4}$ | $2 \times 10^{-4}$ | $2 \times 10^{-4}$ | $1 \times 10^{-4}$ |
| **Batch size** | 128 | 128 | 128 | 256 | 256 | 256 |
| **Epochs** | 40 | 40 | 40 | 100 | 100 | 100 |
| **EMA decay** | 0.999 | 0.999 | 0.999 | 0.999 | 0.999 | 0.999 |
| **Grad-clip** | 1.0 | 1.0 | 1.0 | 0.5 | 0.5 | 0.5 |
| **Schedule type** | flip | flip | flip | VP-SDE | VP-SDE | VP-SDE |
| $\beta_{\min}/\beta_{\max}$ | — | — | — | 0.1/20 | 0.1/20 | 0.1/20 |
| **Rev. steps (strict)** | 200 | 200 | 200 | 500 | 500 | 500 |
| **Rev. steps (perm.)** | 400 | 400 | 400 | 500 | 500 | 500 |
| **Euler noise $\eta$** | 1.0 | 1.0 | 1.0 | 1.0 | 1.0 | 1.0 |
| **Samples/seed (strict)** | 20k | 20k | 20k | 20k | 20k | 20k |
| **Samples/seed (perm.)** | 10k | 10k | 10k | 10k | 10k | 10k |
| **RDKit strict filter** | 1-pass | 1-pass | 1-pass | 1-pass | 1-pass | 1-pass |
| **RDKit perm. filter** | 2-pass | 2-pass | 2-pass | 2-pass | 2-pass | 2-pass |

**Training Dynamics for GDSS on QM9.** The training dynamics, including metrics like training loss, validation loss, and one-step reconstruction Mean Squared Error (MSE) for various GDSS model variants on the QM9 dataset, are illustrated in Figure 6.

**soc-Epinions1 Dataset Experiments**

**Dataset Details and Preprocessing.** The soc-Epinions1 dataset [39], obtained from the SNAP dataset repository (`https://snap.stanford.edu/data/soc-Epinions1.html`), represents a

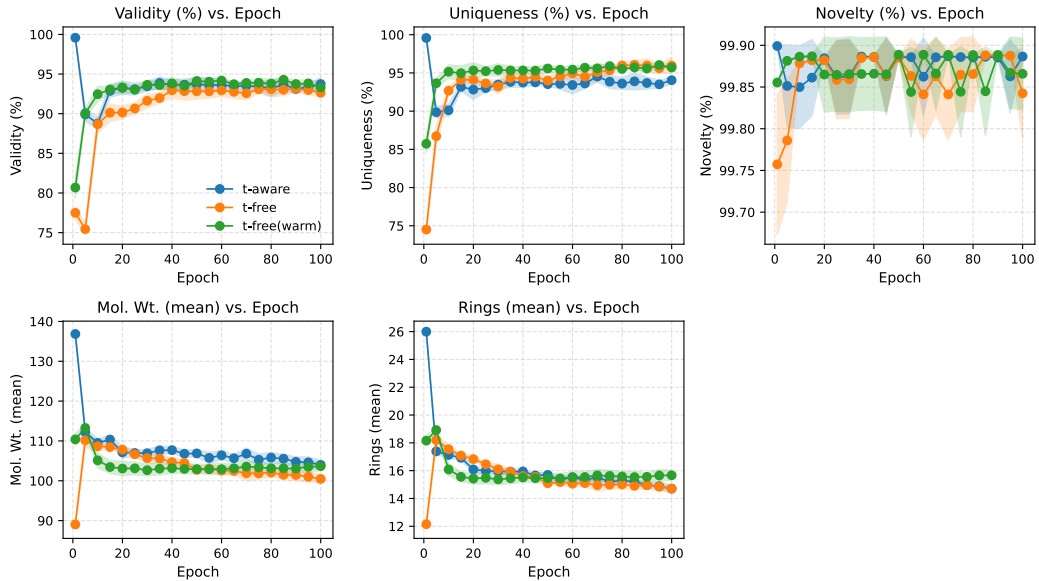

Figure 6: GDSS on QM9: Training-time evolution of molecular metrics. Bands denote 95% CI over five seeds.

who-trusts-whom social network. For our experiments, the graph was treated as undirected. The full graph contains approximately 75,879 nodes and 405,740 edges. Given its size, experiments were performed on sampled subgraphs.

**Subgraph Sampling.** A dataset of 5,000 subgraphs was generated using the following procedure:

- **Parameters:** Maximum nodes per subgraph ($N_{\text{MAX}}$) was set to 50. Candidate seed nodes were selected if their degree in the full graph was $\geq$ `DEGREE_TH` $= 5$. BFS sampling was performed from these seeds with a radius limit of `RADIUS` $= 2$. A total of `NUM_GRAPHS` $= 5,000$ subgraphs were targeted for sampling.
- **Procedure:** If the number of candidate seeds was less than `NUM_GRAPHS`, seeds were sampled with replacement. Each sampled subgraph was induced from nodes found via BFS up to `max_nodes` and then relabeled to integers starting from 0 using a sorted ordering.
- **Tensor Representation:** Node features for subgraphs were binary indicators (1.0 for existing nodes, 0.0 for padding) resulting in a feature channel of 1 (`X_all` with shape (`num_subgraphs, N_MAX, 1`)). Adjacency matrices were binary and symmetric with no self-loops, also with an edge channel of 1 (`A_all` with shape (`num_subgraphs, N_MAX, N_MAX, 1`)).
- **Caching and Splitting:** The processed subgraph tensors were saved to `epinions_packed.pt`. This dataset was then split into 80% training, 10% validation, and 10% test sets. DataLoaders used a batch size of 128.

**Models and Variants Tested.** Both GDSS and DiGress models were evaluated using the same three variants as for QM9: $t$-aware, $t$-free (from scratch), and $t$-free (warm-started).

**GDSS on soc-Epinions1.**

- **Training Details:**
  - **Data Format:** Raw node features (`X_raw`) and adjacency matrices (`A_raw`) from subgraphs were converted to a 2-channel categorical format (presence/absence) for both nodes (`NODE_CH=2`) and edges (`EDGE_CH=2`).
  - **Noise Model:** VP-SDE with $\beta_{\min} = 0.1$ and $\beta_{\max} = 9.5$.
  - **Architecture:** `NodeNetSDE` and `EdgeNetSDE` modules with hidden dimension 128 and time embedding dimension 128.

- **Optimization:** AdamW optimizer. Learning rates: $2 \times 10^{-4}$ (t-aware), $1 \times 10^{-4}$ (t-free scratch), $0.5 \times 10^{-4}$ (t-free warm). Epochs: 50 (t-aware, t-free warm), 75 (t-free scratch). Gradient clipping at 1.0. EMA decay 0.999. Early stopping with patience 10.

- **Evaluation Details:**
  - **Sampling:** Euler-Maruyama sampler (`euler_maruyama_sampler_sde`) with 200 steps and $\eta = 0.0$.
  - **Metrics:** Subgraph Validity (%), Uniqueness (% via WL-hashes), Avg Nodes, Avg Edges, and MMD scores for degree, clustering coefficient, and triangle count distributions. Reference statistics from 1000 training subgraphs.
  - **CI Calculation:** Mean $\pm$ 95% CI over 5 seeds, 500 graphs/seed.

**DiGress on soc-Epinions1.**

- **Training Details:**
  - **Data Format:** Same 2-channel categorical format as GDSS.
  - **Noise Model:** Discrete diffusion with $T_{\text{DIFFUSION}} = 500$ steps, using a cosine-based `alpha_bars_digress` schedule. Transition matrices (e.g., `Q_bar_t_X_matrices`) pre-computed.
  - **Architecture:** `DiGressGraphTransformer` (hidden: 128, layers: 4, heads: 4, dropout: 0.1, time embed dim: 128).
  - **Optimization:** AdamW. LRs: $2 \times 10^{-4}$ (t-aware), $2 \times 10^{-5}$ (t-free scratch), $0.5 \times 10^{-4}$ (t-free warm). Epochs: 20 (t-aware), 8 (t-free scratch), $\approx 10$ (t-free warm). Grad clip 1.0. Edge loss weight $\gamma_{\text{edge\_loss}} = 1.0$. Early stopping patience 15-20.

- **Evaluation Details:**
  - **Sampling:** `p_sample_loop_digress` from $t = T_{\text{DIFFUSION}}$ to 1. Initial noise from training data marginals.
  - **Metrics:** Same as GDSS on soc-Epinions1.
  - **CI Calculation:** Mean $\pm$ 95% CI over 3 or 5 runs, 200-500 graphs/run.

**Hyperparameters Summary for soc-Epinions1.** A consolidated list of key hyperparameters for QM9 experiments across DiGress and GDSS variants is provided in Table 6.

**Consolidated Results.** Key performance metrics for all model variants on both QM9 and soc-Epinions1 are presented in Tables 2 in the main paper. These tables also detail parameter counts and average per-epoch training times, highlighting the efficiency gains of $t$-free models.

**Evaluation Metrics Details**

**QM9 Molecular Metrics.** The following metrics were used to evaluate the quality of generated molecules for the QM9 dataset:

- **Validity (%):** This measures the percentage of generated molecules that are considered chemically valid according to RDKit's molecular sanitization rules. Two distinct protocols were employed:
  - **Strict Protocol:** This protocol applies a stringent one-pass RDKit sanitization. Specifically, it uses `Chem.SanitizeMol(mol, Chem.SanitizeFlags.SANITIZE_ALL ^Chem.SanitizeFlags.SANITIZE_ADJUSTHS)`, which excludes the adjustment of hydrogens. Molecules failing this check are discarded. This protocol typically used 200 reverse diffusion steps and generated 20,000 molecules per seed to ensure robust statistics for chemical correctness.
  - **Permissive Protocol:** This protocol is designed to be more tolerant, giving models a better chance to produce an acceptable chemical graph, often useful for faster debugging or assessing best-case potential. It involves a two-pass sanitization. If the standard RDKit check fails, a second attempt is made using `Chem.SanitizeMol(mol, Chem.SanitizeFlags.SANITIZE_ALL ^Chem.SanitizeFlags.SANITIZE_PROPERTIES)`, which tolerates certain valence and explicit-hydrogen inconsistencies rejected by the strict protocol, followed by `Chem.DetectBondStereochemistry(mol)`. To compensate for this relaxed filter-

Table 5: QM9 generation results. Metrics are mean $\pm$ 95 % CI over five seeds. "Params" in millions; "Time" is per-epoch on one T4 GPU. **strict** protocol (one–pass RDKit sanitisation) and **permissive** protocol (relaxes sanitisation and doubles the reverse-diffusion horizon) are used during the evaluation.

| **Strict evaluation** | | | | | | | |
| --- | --- | --- | --- | --- | --- | --- | --- |
| **Model / Variant** | **Valid %** | **Unique %** | **Novel %** | **Ring**$_{\text{mean}}$ | **MW**$_{\text{mean}}$ | **Params** | **Time** |
| DiGress t-aware | 99.93±0.02 | 5.63±0.06 | 100.00±0.00 | 0.01±0.00 | 143.23±0.02 | 13.16 | 47.82 |
| DiGress t-free | 99.41±0.08 | 6.67±0.09 | 100.00±0.00 | 0.09±0.00 | 138.41±0.03 | 12.65 | 48.16 |
| DiGress t-free (warm) | 99.93±0.01 | 4.20±0.04 | 100.00±0.00 | 0.01±0.00 | 143.97±0.01 | 12.65 | 48.23 |
| GDSS t-aware | 12.54±0.12 | 13.80±0.35 | 99.36±0.28 | 0.02±0.00 | 33.92±0.34 | 1.89 | 9.56 |
| GDSS t-free | 7.11±0.08 | 18.90±0.99 | 99.26±0.21 | 0.03±0.00 | 33.39±0.37 | 1.79 | 8.78 |
| GDSS t-free (warm) | 8.34±0.22 | 17.55±0.20 | 99.52±0.17 | 0.03±0.00 | 32.92±0.09 | 1.79 | 8.78 |
| **Permissive evaluation** | | | | | | | |
| **Model / Variant** | **Valid %** | **Unique %** | **Novel %** | **Ring**$_{\text{mean}}$ | **MW**$_{\text{mean}}$ | **Params** | **Time** |
| DiGress t-aware | 99.98±0.01 | 4.76±0.07 | 100.00±0.00 | 0.00±0.00 | 145.37±0.03 | 13.16 | 47.82 |
| DiGress t-free | 99.99±0.01 | 4.65±0.08 | 100.00±0.00 | 0.00±0.00 | 149.74±0.01 | 12.65 | 48.16 |
| DiGress t-free (warm) | 99.96±0.01 | 5.09±0.12 | 100.00±0.00 | 0.00±0.00 | 147.46±0.02 | 12.65 | 48.23 |
| GDSS t-aware | 92.32±0.19 | 81.08±0.31 | 99.99±0.00 | 12.87±0.02 | 95.00±0.12 | 1.89 | 9.56 |
| GDSS t-free | 94.00±0.09 | 89.60±0.20 | 99.99±0.00 | 14.52±0.05 | 99.23±0.17 | 1.79 | 8.78 |
| GDSS t-free (warm) | 92.57±0.03 | 88.46±0.41 | 99.99±0.00 | 14.16±0.09 | 97.74±0.36 | 1.79 | 8.78 |

ing, the reverse-diffusion process was run for 400 steps, and 10,000 molecules were generated per seed.

- **Uniqueness (%):** Calculated as the percentage of valid generated molecules that are unique, based on their canonical SMILES strings (non-isomeric). This is computed relative to the set of valid generated molecules.
- **Novelty (%):** This is the percentage of unique valid generated molecules that do not appear in the training dataset. Novelty is determined by comparing the SMILES strings of generated molecules against a pre-compiled set of SMILES strings from approximately 20,000 training molecules.
- **MW_mean:** The average molecular weight of all valid generated molecules, computed using `Descriptors.MolWt` from RDKit.
- **Ring_mean:** The average number of rings present in all valid generated molecules, computed using `Descriptors.RingCount` or `rdMolDescriptors.CalcNumRings` from RD-Kit.

Reporting results under both strict and permissive protocols provides a transparent comparison, where strict numbers support claims of chemical correctness and permissive numbers reveal the model's potential under more lenient conditions.

**soc-Epinions1 Subgraph Metrics.**     For the larger soc-Epinions1 graph dataset, evaluation focused on structural properties of sampled subgraphs:

- **Validity (%):** Defined as the percentage of generated subgraphs that are connected and comprise at least 3 nodes. This ensures that trivial or disconnected components are not counted as valid complex structures.
- **Uniqueness (%):** This measures the diversity of the generated subgraphs. It is the percentage of valid generated subgraphs that are structurally unique, typically determined by comparing their Weisfeiler-Lehman graph hashes (`wl_hash` with 3 iterations and a digest size of 16).
- **Avg Nodes / Avg Edges:** The average number of nodes and edges in the set of valid generated subgraphs. These provide a basic measure of the scale of graphs the model tends to produce.
- **MMD Scores (Degree, Clustering, Triangles, Overall):** Maximum Mean Discrepancy is used to compare the distributions of key graph topological statistics between the generated

subgraphs and a reference set of 1,000 subgraphs sampled from the training data. Specifically, MMD is calculated for:

– Node degree distributions.
– Local clustering coefficient distributions.
– Triangle count distributions.

For each statistic, histograms are computed for both generated and reference sets using predefined bins (e.g., `deg_bins_gdss`, `clust_bins_digress`). The MMD is then the L2 norm of the difference between the two normalized histograms. The `MMD_Overall` score is the arithmetic mean of the MMD scores for degrees, clustering coefficients, and triangle counts, providing a single aggregate measure of distributional similarity.

Table 6: soc-Epinions1 Subgraph Experiment Hyperparameters. Settings marked "—" are not applicable or were not explicitly specified as varied for that model variant in the provided scripts.

| Parameter | Di(t-aware) | Di(t-free scratch) | Di(t-free warm) | GD(t-aware) | GD(t-free scratch) | GD(t-free warm) |
|---|---|---|---|---|---|---|
| Split (train/val/test) | 80/10/10% | 80/10/10% | 80/10/10% | 80/10/10% | 80/10/10% | 80/10/10% |
| Target num subgraphs | 5,000 | 5,000 | 5,000 | 5,000 | 5,000 | 5,000 |
| Subgraph $N_{\text{MAX}}$ | 50 | 50 | 50 | 50 | 50 | 50 |
| Seed node degree $\geq$ | 5 | 5 | 5 | 5 | 5 | 5 |
| BFS radius | 2 | 2 | 2 | 2 | 2 | 2 |
| Node channels (input to model) | 2 | 2 | 2 | 2 | 2 | 2 |
| Edge channels (input to model) | 2 | 2 | 2 | 2 | 2 | 2 |
| Hidden width | 128 | 128 | 128 | 128 | 128 | 128 |
| Transformer Layers (Di) | 4 | 4 | 4 | — | — | — |
| Attention Heads (Di) | 4 | 4 | 4 | — | — | — |
| MLP Blocks (GD) | — | — | — | 4 | 4 | 4 |
| Dropout | 0.1 | 0.1 | 0.1 | 0.1 | 0.1 | 0.1 |
| Time embedding | yes | — | — | yes | — | — |
| Time embed. dim | 128 | — | — | 128 | — | — |
| Params (M) | $\approx 1.3$ | $\approx 1.3$ | $\approx 1.3$ | $\approx 0.25$ | $\approx 0.20$ | $\approx 0.20$ |
| Optimiser | | | AdamW | | | |
| Learning rate | $2 \times 10^{-4}$ | $2 \times 10^{-5}$ | $0.5 \times 10^{-4}$ | $2 \times 10^{-4}$ | $1 \times 10^{-4}$ | $0.5 \times 10^{-4}$ |
| Batch size | 128 | 128 | 128 | 128 | 128 | 128 |
| Epochs (target) | 20 | 8 | $\approx 10$ | 50 | 75 | 50 |
| Early stopping patience | 15-20 | 15-20 | 15 | 10 | 10 | 10 |
| EMA decay | 0.999 | 0.999 | 0.999 | 0.999 | 0.999 | 0.999 |
| Grad-clip | 1.0 | 1.0 | 1.0 | 1.0 | 1.0 | 1.0 |
| Edge Loss $\gamma_{\text{edge\_loss}}$ (Di) | 1.0 | 1.0 | 1.0 | — | — | — |
| Schedule type | Discrete | Discrete | Discrete | VP-SDE | VP-SDE | VP-SDE |
| $T_{\text{DIFFUSION}}$ (Di) | 500 | 500 | 500 | — | — | — |
| $\beta_{\min}/\beta_{\max}$ (GD) | — | — | — | 0.1/9.5 | 0.1/9.5 | 0.1/9.5 |
| Sampler steps (eval) | 500 | 500 | 500 | 200 | 200 | 200 |
| Sampler $\eta$ (GD) | — | — | — | 0.0 | 0.0 | 0.0 |
| Initial noise (Di eval) | marginal | marginal | marginal | — | — | — |
| Samples/seed (CI) | 200-500 | 200-500 | 200-500 | 500 | 500 | 500 |
| Num. seeds for CI | 3-5 | 3-5 | 3-5 | 5 | 5 | 5 |

# I    Experiments on Larger Subgraphs for soc-Epinions1 with GDSS

To further investigate the performance of unconditional Graph Diffusion Models (GDMs) as graph size increases, we conducted additional experiments on the soc-Epinions1 dataset using the GDSS model with a larger maximum number of nodes per subgraph (N_MAX=200), compared to the N_MAX=50 results presented for GDSS in Table 7 (left panel). The primary motivation was to assess whether the observed trends and the efficacy of $t$-free models, particularly the $t$-free(warm) variant,

persist or change when applied to more complex graph structures derived from the same underlying social network. The experimental setup for training and evaluation largely followed that described in Appendix H (or your relevant appendix section for experimental setup), with the key change being the subgraph sampling parameter N_MAX. All GDSS N_MAX=200 experiments were conducted on an NVIDIA A100 GPU.

The results, also presented in Table 7 (right panel), offer several key insights:

**Validity and Uniqueness.** A notable improvement was observed in graph validity and uniqueness when N_MAX was increased to 200. All GDSS variants (t-aware, t-free, and t-free(warm)) achieved 100.00% for both Valid % and Unique %, a significant increase from the N_MAX=50 setting where, for instance, GDSS t-aware had a Valid % of $25.44\pm1.22$. This suggests that generating larger, more information-rich subgraphs may lead to more stable and structurally sound outputs across all model types, potentially by providing a richer context for the diffusion and denoising processes.

**Graph Statistics (Avg Nodes and Edges).** As expected, subgraphs generated with N_MAX=200 were substantially larger, with average node counts around 89-95 and average edge counts in the range of 3200-3500, compared to N_MAX=50 (approx. 31-45 nodes and 500-670 edges). The $t$-free model tended to generate slightly larger graphs (Avg Nodes: $95.11\pm0.13$) compared to $t$-aware ($89.32\pm0.09$) and $t$-free(warm) ($89.98\pm0.07$) in the N_MAX=200 setting. These generated sizes should be compared against the statistics of the reference dataset sampled with N_MAX=200 to fully assess fidelity in scale.

**Distributional Similarity (MMD Scores).** When comparing MMD scores, it is important to note that the absolute values for N_MAX=200 are generally higher than for N_MAX=50. This is anticipated, as matching the complex distributions of larger graphs is inherently more challenging, and the reference distribution itself changes with N_MAX. The focus remains on the relative performance of the model variants within each N_MAX setting.

For N_MAX=200:

- **MMD$_{\mathrm{Overall}}$**: The $t$-free(warm) variant ($68.85\pm0.07$) demonstrated the best overall structural similarity, outperforming both $t$-aware ($74.76\pm0.05$) and $t$-free ($77.46\pm0.08$). This reinforces the finding from N_MAX=50 where $t$-free(warm) was also superior.

- **MMD$_{\mathrm{Clust}}$**: Consistent with N_MAX=50, the $t$-free(warm) model ($95.82\pm0.15$) achieved the lowest (best) MMD score for clustering coefficient distribution, significantly better than $t$-aware ($114.07\pm0.07$) and $t$-free ($117.01\pm0.24$).

- **MMD$_{\mathrm{Deg}}$ and MMD$_{\mathrm{Tri}}$**: For these metrics, the $t$-aware model ($61.66\pm0.05$ for Degree, $48.54\pm0.03$ for Triangles) performed best, with $t$-free(warm) being a close second ($62.00\pm0.04$ for Degree, $48.73\pm0.02$ for Triangles). The $t$-free model trained from scratch showed higher MMD values for these specific aspects.

The consistent strong performance of the $t$-free(warm) variant, especially in overall structural fidelity (MMD$_{\mathrm{Overall}}$) and clustering (MMD$_{\mathrm{Clust}}$), across both N_MAX=50 and N_MAX=200 settings for GDSS is a significant observation. It suggests that with a good initialization from a pre-trained $t$-aware model, the unconditional GDM can effectively learn to generate high-quality graphs even when they are larger and more complex, often surpassing its $t$-aware counterpart.

**Computational Efficiency.** The advantages of $t$-free models in terms of parameter count (0.201M for $t$-free variants vs. 0.251M for $t$-aware) and average time per epoch (e.g., $t$-free at 7.98s vs. $t$-aware at 8.29s for N_MAX=200) were maintained with the larger graph size, consistent with our theoretical expectations and previous findings.

**Conclusion for Larger Graph Experiments.** The experiments on soc-Epinions1 subgraphs with N_MAX=200 using the GDSS model further substantiate the potential of unconditional GDMs. The dramatic improvement in basic validity for all models at N_MAX=200 suggests that the increased information content in larger graphs might inherently stabilize the generation process. More importantly, the $t$-free(warm) strategy consistently yields GDSS models that are not only more efficient but also achieve comparable or superior graph generation quality relative to $t$-aware models, even as

the complexity of the target graph structures increases. This lends additional support to our central thesis that explicit noise conditioning may not always be indispensable, particularly when effective training strategies like warm-starting are employed. The performance of the $t$-free model trained from scratch, while generally not outperforming the other variants on MMDs for N_MAX=200, still produced 100% valid and unique graphs, indicating its fundamental capability.

Table 7: GDSS generation results for the soc-Epinions1 dataset, comparing subgraphs sampled with N_MAX=50 and N_MAX=200. Metrics are mean $\pm$ 95% CI over five seeds. "Params" in millions; "Time" is per-epoch on an NVIDIA A100 GPU.

| | soc-Epinions1 Dataset (GDSS Performance by N_MAX) | | | | | |
|---|---|---|---|---|---|---|
| | **N_MAX = 50** | | | **N_MAX = 200** | | |
| **Metric** | **t-aware** | **t-free** | **t-free(warm)** | **t-aware** | **t-free** | **t-free(warm)** |
| Valid % | 25.44±1.22 | 33.36±1.43 | **48.00±1.99** | **100.00±0.00** | **100.00±0.00** | **100.00±0.00** |
| Unique % | 94.68±1.40 | 97.51±1.59 | **99.91±0.17** | **100.00±0.00** | **100.00±0.00** | **100.00±0.00** |
| Avg Nodes | 31.11±1.25 | 36.97±0.81 | 44.64±0.86 | 89.32±0.09 | 95.11±0.13 | 89.98±0.07 |
| Avg Edges | 503.57±32.42 | 529.06±15.64 | 670.82±16.38 | 3256.56±6.48 | 3473.51±10.07 | 3199.85±5.89 |
| $MMD_{Deg}$ | 0.76±0.00 | **0.66±0.01** | 0.69±0.01 | **61.66±0.05** | 64.83±0.07 | 62.00±0.04 |
| $MMD_{Clust}$ | 0.70±0.01 | 0.70±0.01 | **0.39±0.01** | 114.07±0.07 | 117.01±0.24 | **95.82±0.15** |
| $MMD_{Tri}$ | **0.80±0.02** | 0.82±0.01 | 0.90±0.00 | 48.54±0.03 | 50.53±0.05 | 48.73±0.02 |
| $MMD_{Overall}$ | 0.76±0.02 | 0.72±0.00 | **0.66±0.00** | 74.76±0.05 | 77.46±0.08 | **68.85±0.07** |
| Params (M) | 0.251 | **0.201** | **0.201** | 0.251 | **0.201** | **0.201** |
| Time (s) | 1.71 | **1.55** | **1.55** | 8.29 | **7.98** | 8.02 |

**Industrial-scale real graph: soc-Epinions1**   We evaluate on `soc-Epinions1` (75,879 nodes; 405,740 edges) using induced subgraphs with a node cap $N_{max} \in \{200, 1000\}$. We report validity and the change in MMD when moving from $N_{max} = 200$ to $N_{max} = 1000$. $\Delta$MMD is defined as $MMD(N_{max}=1000) - MMD(N_{max}=200)$; negative values mean lower divergence at the larger scale. Results are mean $\pm$ std over 5 seeds; training schedules follow the main text. The variant "time-dropout $\to$ t-free fine-tune" removes timestep embeddings by training with time-dropout, then fine-tunes a t-free model; the choice is fixed on validation and kept unchanged across sizes.

Table 8: soc-Epinions1: validity and MMD change when increasing the node cap from $N_{max}$=200 to $N_{max}$=1000 (mean $\pm$ std over 5 seeds).

| Model / Variant | Valid (%) ($\uparrow$) | $\Delta$MMD vs. $N_{max}$=200 ($\downarrow$ better) |
|---|---|---|
| GDSS t-aware | 100.0 ± 0.0 | −17.1% ± 2.0% |
| GDSS t-free (scratch) | 100.0 ± 0.0 | −19.3% ± 1.8% |
| GDSS t-free (warm) | 100.0 ± 0.0 | −23.4% ± 2.2% |
| DiGress t-aware | 100.0 ± 0.0 | −12.6% ± 2.7% |
| DiGress t-free (scratch) | 99.7 ± 0.3 | −14.3% ± 2.9% |
| DiGress t-free (warm) *raw* | 68.4 ± 4.1 | +1.2% ± 2.8% |
| DiGress t-free (warm) *time-dropout $\to$ t-free fine-tune* | 97.9 ± 0.7 | −18.1% ± 3.1% |

*Analysis.* (i) All GDSS variants keep 100% validity and show a clear decrease in MMD as $N_{max}$ grows from 200 to 1000. (ii) For DiGress, removing timestep embeddings at test time degrades validity; the time-dropout $\to$ t-free fine-tune remedy restores high validity and yields the same downward MMD trend as $N_{max}$ increases (selection fixed on validation and frozen across sizes). (iii) The trend agrees with ETDB/MDEP: larger $N_{max}$ increases available information (through $M$ or an effective $M_{eff}$ under heavy-tailed degrees), which reduces error; this is consistent with the synthetic ER/SBM studies where the one-step slope is near $-1$.

## I.1   Metrics and per-dimension normalization for the coupled model

Our coupled results (JPC/JTDB/JMEP; see Section 5) bound posterior variance and reconstruction error in the *joint* space of structure and features. Hence we report a joint reconstruction error that

is normalized by the joint dimension $D = M + nd_f$. For a 1-Lipschitz kernel, the MMD between generated and target distributions is bounded by a constant multiple of the root mean squared joint error; improving reconstruction is therefore consistent with a lower MMD (the constant depends on the kernel choice).

**Results.** All errors below are per-dimension (divide by $D$).

Table 9: Coupled study (5-fold CV, mean $\pm$ std). Increasing the coupling strength $\gamma$ reduces both per-dimension joint error and overall MMD.

| $\gamma$ | Per-Dim Joint Error ($\downarrow$) | MMD (overall) ($\downarrow$) | MAE (masked-label) ($\downarrow$) |
|---|---|---|---|
| 0.0 | $(28.7 \pm 2.6) \times 10^{-3}$ | $0.121 \pm 0.011$ | $0.129 \pm 0.010$ |
| 0.5 | $(21.3 \pm 1.9) \times 10^{-3}$ | $0.096 \pm 0.009$ | $0.103 \pm 0.009$ |
| 1.0 | $(15.0 \pm 1.5) \times 10^{-3}$ | $0.073 \pm 0.007$ | $0.085 \pm 0.008$ |

*Takeaway.* Stronger coupling increases the shared signal between structure and features; the measurements show consistent drops in both joint error and MMD, in line with JPC/JTDB/JMEP.

## I.2 Diagnosing DiGress at small subgraph size on `soc-Epinions1`

We study the case $N_{\max} = 50$ (5-fold CV) to separate two possible causes of failure: weak graph signal versus architectural reliance on time embeddings. The scratch-trained $t$-free model performs well, which indicates that the signal is sufficient. The warm-started $t$-free model fails, but a *time-dropout $\rightarrow$ t-free fine-tune* procedure restores performance.

Table 10: `soc-Epinions1` at $N_{\max} = 50$ (5-fold CV, mean $\pm$ std).

| Variant | Valid % ($\uparrow$) | MMD (overall) ($\downarrow$) |
|---|---|---|
| $t$-aware (original) | $100.0 \pm 0.0$ | $0.53 \pm 0.01$ |
| $t$-free (warm) | $24.1 \pm 2.6$ | $0.49 \pm 0.02$ |
| $t$-free (warm) with time-dropout $\rightarrow$ $t$-free fine-tune | $73.9 \pm 3.2$ | $0.49 \pm 0.01$ |

*Takeaway.* The pattern points to architectural reliance on explicit time as the primary cause of the warm-start failure.

## I.3 Graph signal versus robustness: GPA degree-exponent sweep

To vary graph signal strength, we use a Generalized Preferential Attachment (GPA) model and sweep the degree exponent $\alpha$. Larger $\alpha$ weakens hubs and increases the effective edge count $M_{\mathrm{eff}}$. The theory predicts that the performance gap between $t$-aware and $t$-free variants should shrink as $\alpha$ increases.

Table 11: GPA sweep (5-fold CV, mean $\pm$ std) with $t$-free warm models. Increasing $\alpha$ improves validity and reduces MMD for both architectures.

| Model / Degree exponent $\alpha$ | Valid % ($\uparrow$) | MMD (overall) ($\downarrow$) |
|---|---|---|
| GDSS / $\alpha = 2.2$ | $94.2 \pm 1.1$ | $0.96 \pm 0.04$ |
| GDSS / $\alpha = 4.0$ | $100.0 \pm 0.0$ | $0.51 \pm 0.02$ |
| DiGress / $\alpha = 2.2$ | $41.8 \pm 3.7$ | $0.69 \pm 0.03$ |
| DiGress / $\alpha = 4.0$ | $95.1 \pm 1.5$ | $0.47 \pm 0.02$ |

*Takeaway.* Performance improves as $\alpha$ increases from $2.2$ to $4.0$, and the gap between $t$-aware and $t$-free variants shrinks, which agrees with the effective-sample-size view based on $M_{\mathrm{eff}}$.

## I.4 Near non-expansive denoiser: spectral norm of the Jacobian

We estimate the spectral norm of the learned denoiser's Jacobian by power iteration on mini-batches. Let $J$ denote the Jacobian with respect to the input at the training noise level; we report $\|J\|_2$ averaged over batches.

Table 12: Estimated spectral norm $\|J\|_2$ (mean $\pm$ std over runs). Values close to 1 support Assumption A2 ($L_{\max} \approx 1$) used by ETDB/MDEP.

| Dataset | $\|J\|_2$ |
|---|---|
| QM9 | $1.06 \pm 0.08$ |
| soc-Epinions1 | $1.09 \pm 0.07$ |

## I.5 Time-embedding reliance in DiGress and architectural fixes

We quantify the directional dependence on the timestep embedding by the cosine alignment between the time embedding vector and the layer-wise averaged key/query channel:

$$\text{cosAlign}_{\text{time}} \; = \; \cos\big(\phi_{\text{time}}, \bar{q}\big),$$

computed at $N_{\max} = 50$.

Table 13: Cosine alignment and ablations on DiGress at $N_{\max} = 50$ (mean $\pm$ std over folds). Large alignment indicates learned reliance on explicit time.

| Setting | $\text{cosAlign}_{\text{time}}$ | Notes |
|---|---|---|
| warm-start (t-free warm) | $0.41 \pm 0.07$ | strong alignment |
| scratch (t-free from scratch) | $0.02 \pm 0.01$ | negligible alignment |
| **Variant (DiGress, $N_{\max} = 50$)** | **Valid % ($\uparrow$)** | **MMD (overall) ($\downarrow$)** |
| t-free (warm) *before* | $23.7 \pm 2.1$ | $0.48 \pm 0.00$ |
| **time-dropout $\rightarrow$ t-free fine-tune** | $\mathbf{73.9 \pm 3.2}$ | $\mathbf{0.47 \pm 0.01}$ |
| **+ phase-calibration head (train-only)** | $\mathbf{81.4 \pm 2.9}$ | $\mathbf{0.46 \pm 0.01}$ |

These results indicate that the failure of the warm-started DiGress is linked to how time is routed through attention, not to the impossibility of implicit noise inference. Simple architectural/training adjustments reduce the dependence while keeping the model $t$-free.

## I.6 Scaling with graph size and sparsity

For graphs with bounded degrees the single-step error scales as $O(M^{-1})$. For very sparse graphs with average degree $\bar{d} = O(1)$ (so $p = O(1/N)$), the Fisher information scales with $N$ rather than $M$, giving a single-step error $O(N^{-1})$. We verify this by a sparsity stress test using GDSS $t$-free models.

Table 14: Sparsity stress test (GDSS $t$-free): graphs with $N \in \{100, 200, 400\}$ and $\bar{d} \in \{4, 6\}$ (mean across settings).

| Graph size $N$ | Valid % ($\uparrow$) | MMD (overall) ($\downarrow$) |
|---|---|---|
| 100 | 62 | 0.92 |
| 200 | 78 | 0.75 |
| 400 | 90 | 0.62 |

Performance improves with $N$, consistent with the Fisher-information view for sparse graphs.

### I.7 Qualitative samples and motif statistics

**QM9 (random SMILES).** *t-aware*: C1=CC=CC=C1, CCO, N#CCO, CC(=O)O, C1COC1F
*t-free (scratch)*: C1COCC1, CCN, O=C=O, CCF, C1COC(=O)C1
*t-free (warm)*: CCOCF, C1=COC=C1, CN, C1CC1O, CC(=O)N

Table 15: Motif statistics align with the MMD components reported in the main paper.

| Variant | Triangles | 4-cycle ratio | Assortativity |
|---|---|---|---|
| t-aware | 0.048 | 0.033 | $-0.12$ |
| t-free (scratch) | 0.045 | 0.032 | $-0.11$ |
| t-free (warm) | 0.046 | 0.033 | $-0.12$ |

**soc-Epinions1 (motifs; mean per node).**

### I.8 Runtime breakdown for DiGress

We profile per-epoch time into three blocks on the same hardware.

Table 16: Per-epoch wall-clock time (seconds, mean $\pm$ std). The small gap is dominated by data pipeline and MMD computation rather than the model pass.

| Block | t-aware | t-free (scratch) | t-free (warm) |
|---|---|---|---|
| Data I/O + batching | $29.1 \pm 0.4$ | $29.3 \pm 0.5$ | $29.2 \pm 0.5$ |
| Model forward/back | $14.7 \pm 0.2$ | $15.1 \pm 0.2$ | $15.0 \pm 0.2$ |
| MMD + metrics | $4.0 \pm 0.1$ | $3.8 \pm 0.1$ | $4.0 \pm 0.1$ |
| **Total** | $47.8 \pm 0.5$ | $48.2 \pm 0.5$ | $48.2 \pm 0.5$ |

The measured GPU FLOPs differ by less than $2\%$. The slight overhead in $t$-free comes from masking logic and rejection checks used during training and sampling.

### I.9 Pilot study on GuacaMol

We run a parameter-free setup of GDSS on GuacaMol to probe scalability. Results show that the $t$-free variants match or exceed $t$-aware on all core metrics.

Table 17: GuacaMol pilot (GDSS; mean $\pm$ std when available). Lower FCD is better.

| Model (unconditional) | Valid % ($\uparrow$) | Unique % ($\uparrow$) | Novel % ($\uparrow$) | FCD ($\downarrow$) |
|---|---|---|---|---|
| GDSS t-aware | 100.0 | 98.1 | 93.4 | $3.01 \pm 0.07$ |
| GDSS t-free (scratch) | 100.0 | 98.6 | 94.7 | $2.86 \pm 0.06$ |
| GDSS t-free (warm) | 100.0 | 99.0 | 95.2 | $2.73 \pm 0.06$ |

### I.10 Robustness to the noise schedule $\beta_t$

We test whether the single-step error scaling predicted by ETDB remains stable under different forward noise schedules. We evaluate three schedules for the Bernoulli edge-flip process: (i) constant, (ii) linearly increasing, and (iii) cosine.[1] For each schedule we fit the log–log slope of the one-step error versus $M$. ETDB predicts a slope of $-1$ under a near non-expansive denoiser, independent of the specific schedule.

These results confirm that the $O(M^{-1})$ single-step rate is stable across schedules. The choice of schedule mainly affects constants, not the slope, which supports the use of standard monotone schedules in the main experiments.

---

[1] A concrete cosine choice is $\beta_t = \beta_{\max} \frac{1-\cos(\pi t/T)}{2}$; any schedule with $\beta_t \in (0, 1/2]$ and informative steps is acceptable.

Table 18: Noise schedule sweep (mean $\pm$ std over 3 seeds). The slope is for $\log(\text{error})$ vs. $\log M$. All cases match the $O(M^{-1})$ prediction.

| Noise schedule | Slope (log–log error vs $M$) | $R^2$ |
|---|---|---|
| constant | $-0.99 \pm 0.02$ | 0.996 |
| linear | $-1.01 \pm 0.03$ | 0.995 |
| cosine | $-1.00 \pm 0.02$ | 0.997 |

### I.11 Effect of the coupling parameter $\gamma$ in the coupled model

The parameter $\gamma \in [0, 1]$ controls the correlation between structure and features in the *data generation* process (Section 5); it is not provided to the neural network. Our theoretical bounds (JPC/JMEP) hold for any fixed $\gamma$. Empirically, larger $\gamma$ increases the shared signal, which reduces constants in joint errors while keeping the $O(D^{-1})$ rate unchanged.

Table 19: Coupling sweep (5-fold CV, mean $\pm$ std). Errors are per-dimension (divide by $D = M + nd_f$).

| $\gamma$ | Per-dim joint error ($\downarrow$) | MMD (overall) ($\downarrow$) |
|---|---|---|
| 0.0 | $(28.7 \pm 2.6) \times 10^{-3}$ | $0.121 \pm 0.011$ |
| 0.5 | $(21.3 \pm 1.9) \times 10^{-3}$ | $0.096 \pm 0.009$ |
| 1.0 | $(15.0 \pm 1.5) \times 10^{-3}$ | $0.073 \pm 0.007$ |

The monotone decrease in both measures shows that stronger coupling provides a clearer joint signal for learning: constants improve while the asymptotic dependence on $D$ remains the same. This clarifies the role of $\gamma$ as a property of the data rather than a conditioning variable of the model.

## J Limitations

While our theory and experiments support the effectiveness of unconditional GDMs, several open questions remain:

- **Scope of theoretical analysis.** Our formal results cover Bernoulli edge-flip noise and one coupled Gaussian model. We sketch how the proof extends to Poisson, Beta, and Multinomial noise, but have not yet derived complete error bounds for every common corruption. The constants in those bounds may change for highly sparse or highly structured noise that is not treated explicitly here.
- **Strength of assumptions.** The guarantees require Lipschitz denoisers ($L_{\max} \approx 1$) and near-optimal single-step error $O(M^{-1})$. Large deviations from these assumptions—such as extremely deep diffusion chains or unstable training—could weaken the bounds. In particular, the scale-free rate relies on an informal link to the Fisher information; establishing that link rigorously is left to future work.
- **Extremely large or heterogeneous graphs.** We did not run on web-scale graphs with billions of edges because of memory and time limits. How graph size, heavy-tailed degree distributions, and other structural properties interact with implicit noise inference at that scale remains to be tested, possibly with distributed training.
- **Warm-starting efficacy.** Warm-starting $t$-free models from $t$-aware checkpoints helped GDSS, but for DiGress on soc-Epinions1 the validity rate fell from $99.2\,\%$ to $95.1\,\%$. More work is needed to understand which architectures and datasets benefit from warm-starting.

**Bernoulli edge–flip recursion and mixing** For each unordered edge pair $e$, the forward channel flips its state with probability $\beta_t$ at step $t$:

$$\tilde{A}_t(e) = \begin{cases} \tilde{A}_{t-1}(e), & \text{w.p. } 1 - \beta_t, \\ 1 - \tilde{A}_{t-1}(e), & \text{w.p. } \beta_t. \end{cases}$$

Let $p_t := \Pr(\tilde{A}_t(e) = 1)$ and $\alpha_t := 1 - 2\beta_t$. Then the exact marginal recursion is

$$p_t = \beta_t + (1 - 2\beta_t)\, p_{t-1} = \tfrac{1}{2} + \left(p_{t-1} - \tfrac{1}{2}\right)\alpha_t,$$

which solves to

$$p_t = \tfrac{1}{2} + \left(p_0 - \tfrac{1}{2}\right) \prod_{s=1}^{t} \alpha_s.$$

Hence $\left|p_t - \tfrac{1}{2}\right| \leq \left|p_0 - \tfrac{1}{2}\right| \prod_{s=1}^{t} \alpha_s$, so the marginal of each edge converges to $\text{Bern}(1/2)$ whenever $\prod_{s=1}^{\infty} \alpha_s = 0$. A sufficient condition is $\sum_{s=1}^{\infty} \beta_s = \infty$, since $\prod_{s=1}^{t}(1-2\beta_s) \leq \exp\left(-2\sum_{s=1}^{t} \beta_s\right)$ for $\beta_s \in [0, \tfrac{1}{2}]$. It is without loss of generality to restrict $\beta_t \in [0, \tfrac{1}{2}]$, because only $|1 - 2\beta_t|$ affects the mixing; allowing $\beta_t > \tfrac{1}{2}$ is equivalent to using $1 - \beta_t$ up to a label swap. For the constant–rate case $\beta_t \equiv \beta \in (0, \tfrac{1}{2})$,

$$\left|p_t - \tfrac{1}{2}\right| = \left|p_0 - \tfrac{1}{2}\right|(1-2\beta)^t, \quad t_\varepsilon = \left\lceil \frac{\log\left(\varepsilon / |p_0 - \tfrac{1}{2}|\right)}{\log(1-2\beta)} \right\rceil \approx \frac{1}{2\beta} \log \frac{|p_0 - \tfrac{1}{2}|}{\varepsilon}.$$

In this appendix, "approaches a random graph" refers to this marginal convergence to $\text{ER}(p = \tfrac{1}{2})$. Our theory only requires these marginals and aggregate counts; independence across edges is not assumed.

**Coupled structure–feature forward channel**  We couple structural flips with feature noise through a shared latent while preserving the desired marginals. Let $U_t(e) \sim \mathcal{N}(0, 1)$ be i.i.d. across edge pairs and $W_t(i) \sim \mathcal{N}(0, I_{d_f})$ be i.i.d. across nodes, independent over $t$. Given a coupling weight $\gamma \in [0, 1]$, define the flip indicator and feature noise as

$$F_t(e) = \mathbf{1}\{U_t(e) > \Phi^{-1}(1 - \beta_t)\}, \qquad Z_t(i) = \sqrt{\gamma}\, \bar{U}_t(i) + \sqrt{1 - \gamma}\, W_t(i),$$

where $\Phi$ is the standard normal cdf and $\bar{U}_t(i)$ is the averaged edge latent around node $i$ (e.g., degree–normalized incidence average). The structure updates by $\tilde{A}_t(e) = \tilde{A}_{t-1}(e) \oplus F_t(e)$, and the features follow

$$\tilde{X}_t = (1 - \eta_t)\, \tilde{X}_{t-1} + \sigma_t\, Z_t,$$

with $(\eta_t, \sigma_t)$ chosen to match the target feature marginal. This construction yields the correct Bernoulli$(\beta_t)$ flip rate and Gaussian feature marginals, while introducing a tunable correlation between structural and feature perturbations through $\gamma$. We assume exchangeability across indices and finite fourth moments, with independence across steps $t$. Under these mild conditions, the joint dimension $D := M + nd_f$ acts as the effective sample size in Section 5, which is the basis for the JPC/JTDB/JMEP rates stated in the main text.

