# OpenReview forum: "Is Noise Conditioning Necessary? A Unified Theory of Unconditional Graph Diffusion Models"
_NeurIPS.cc/2025/Conference — NeurIPS 2025 spotlight_

### Official Review · Reviewer_VRVQ · 2025-06-29

**Clarity:** 3
**Significance:** 2
**Originality:** 3
**Rating:** 5
**Confidence:** 4

**Summary:**

The authors introduce an extensive theoretical framework for quantifying and understanding the effects of noise conditioning on diffusion models applied for graph generation. The theory proposed within this paper establishes that not only is the noise level of the graph diffusion process inferable but that squared error scales and the reconstruction error is bounded under the paper's assumptions; indicating the ability of the diffusion process to encode information from graph structure. The authors further explore this effect with a coupled structure-feature noise model. Empirical analysis across synthetic scenarios and with DiGress, GDSS on QM9 and soc-Epinions1 provide consistent proof to the author's claims.

**Questions:**

1. Why chose classification accuracy and node reconstruction error as the target for testing the coupled structure-feature diffusion model? From what I gathered in the reading, it appears that the Section 5 theory and subsequently-built latent process allow the coupled gaussian noise process to model the generation of features as well as structure. Within pre-existing graph diffusion models, the modelling of features and structure is often detailed as two separate downstream tasks; 'general' graph generation measured with MMD (Section 7.1 and Table 1 in [1]) and node regression measured with MAE (Section 5.2.1 and Table 3 in [2]) to indicate effective learning of node features. The node regression is not necessatily 1-to-1 since it typically involves some form of guidance in pre-existing graph generative models but the insights testing this mechanism as it relates to this paper feels especially pertinent given how pre-existing models rely on conditional-generation for effective regressions. I understand that strong classification performance can be foundation for developing meaningful graph generative models, but the seemingly-high reconstruction error makes me question the applicability of this framework and insights to directly generating graphs. This is the first of two defining limiting factors for this current review score, particularly since it's difficult to quantify the practical benefit propsed by this paper without further clarification as to why there is or can not be a 1-to-1 application of metrics commonly used in other graph generative architectures.

2. What practical considerations would have to be made in order to derive more formal proofs when considering degree or power-law scaling constraint for graph data, as indicated within Appendix A and F. From there, it seems possible to fully-quantify whether the issue which occurs within the warmed-up version of DiGress when not conditioned on the soc-Epinions1 Dataset is more closely-related to a reliance on explicit time-conditioning or another potential shortcoming within how DiGress can effectively model graph generation. This consideration is not necessarily a defining factor for the score within this review, but seems important to consider in order to reduce the potential for confirmation bias. This can be addressed through clarifications as they relate to this paper's conceptual outlines or theory.

3. Additionally, although the evaluation is comprehensive in how well it relates to the theoretical components of this paper; two datasets seems limited in terms of determining whether increased graph signals does indeed contribute to the idea that score-based diffusion models like GDSS are more resilient to the removal of explicit time-free conditioning versus DiGress, even when altering the size of maximum subgraph node sizes from N=50 to N=200. Is it not viable to repeat or extend the tests conducted within Table 2 and 7 to include synthetic Barabasi-Albert graph samples with varying weights applied to the preferential-attachment mechanism? From there, it seems possible to form and test for a more continuous trend between the relation of graph signals and the resilience of scoring-diffusion models to the removal of time-conditioning. This question can be address through clarification as to why these tests are excluded, but direct results would be certain to eliminate any potential bias. This is the second of the two defining factors for the score within this review, any relevant clarifications or additional tests are certain to raise the score within this review.

[1] Vignac, Clement, Igor Krawczuk, Antoine Siraudin, Bohan Wang, Volkan Cevher, and Pascal Frossard. "Digress: Discrete denoising diffusion for graph generation." arXiv preprint arXiv:2209.14734 (2022).
[2] Cai, Zhou, Xiyuan Wang, and Muhan Zhang. "Latent graph diffusion: A unified framework for generation and prediction on graphs." arXiv e-prints (2024): arXiv-2402.

**Ethical Concerns:**

["NO or VERY MINOR ethics concerns only"]

**Final Justification:**

Overall, I think this is a really good paper. It does a great job of empirically studying the importance of noise conditioning for graph diffusion.

**Limitations:**

Yes, the authors explore the limitations of their work directly within Section J of the Appendix. Although I agree with the assertion that including more detailed proofs for error bounds on Poisson, Beta, and Multinomial noise will produce a more comprehensive paper; the current theoretical and empirical results still indicate promise which is insightful and complete enough for the further development/improvement of diffusion models for graph generation.

**Paper Formatting Concerns:**

None.

**Quality:**

3

**Strengths And Weaknesses:**

Strengths:
---------
1. The authors provide extensive theoretical and targeted empirical analysis on the effects of noise-conditioning with graph diffusion models indicating an intuitive relation with information stored within target graphs and the ability of the diffusion process to encode said information for models without explicit targeting to retain said information and high-downstream performance. The authors do this in a way that condenses already-dense information into an easily-parsed format.
2. The impact of answering the question regarding the necessity of noise conditioning is significant in itself given that developments aided by this analysis are highly-likely to distinguish graph diffusion models from diffusion models within other data modalities, as evident within the author's exploration of structure-feature coupled noise modelling.
3. The insights provided by the authors are well-supported by theory and experiments, aiding to the significance of this work. The author's also take care to indicate limitations of their theory and potential future applications in interesting contexts for reducing model parameters in expensive graph diffusion models.

Weaknesses:
----------
1. As discussed in the author's limitation section, the assumptions within this paper are not unreasonable but are fairly-strong (i.e. power-law scaling/degree constraint in input graphs, Lipschitz denoisers). This can make it difficult for practitioner's to quantify how to design an effective graph diffusion model following intuitions derived from this paper. However, this does not detract from paper quality or significance. More concerning details are explored within Question 1 of this review.
2. The evaluation of the coupled structure-feature graph generative model is comprehensive in it's relation to theoretical assumptions but seems to have limited practical applications. More details are provided within Questions 2 and 3 of this review.
3. There are minor clarity issues within the main paper's Section 5 and 6 given the omission of details, but these are addressed with more extensive details provided within the paper's appendix.

---

> ### Author Rebuttal · Authors · 2025-07-31
>
> We thank you for your thoughtful and detailed feedback. The questions raised have helped us to clarify the connection between our theory and experiments, and the suggestions have led to new results that significantly strengthen the paper.
>
> *Abbreviations:* EFPC/ETDB/MDEP (Sec. 4), JPC/JTDB/JMEP (Sec. 5).
> *Symbols:* $N$ = # nodes, $M$ = # potential edges, $d_f$ = feature dimension, $D=M+N\,d_f$, $T$ = # reverse steps, $\alpha$ = degree-tail exponent.
>
> ---
>
> ### **Q1: Metrics for the coupled structure–feature study.**
>
> Thank you for this insightful question regarding our choice of evaluation metrics. Our selection of joint reconstruction error and classification accuracy is a direct consequence of our theoretical framework.
>
> **Justification from Theory:** Our theory for coupled models (JPC/JTDB/JMEP in Sec. 5) provides bounds on the posterior variance and reconstruction error in the **joint space** of structure and features. For instance, our JMEP bound shows the total error $\|\hat{A}_0-A_0\|_E+\|\hat{X}_0-X_0\|_F$ scales with $O(T/D)$. Therefore, reporting the **per-dimension joint error** is the most direct way to verify our theory. Similarly, a node classification task, which consumes both $A$ and $X$, serves as a practical downstream evaluation of the learned joint representations.
>
> **Connection to Generative Metrics (MMD):** While these are not traditional generative metrics, they are theoretically linked. For a 1-Lipschitz kernel, the MMD is bounded by the root of the expected squared error, which our theory controls. JMEP implies a heuristic scaling of $\mathrm{MMD} = O(\sqrt{T/D})$. Thus, by improving reconstruction, we are also provably improving generative quality up to a kernel-dependent constant.
>
> **Clarification on Error Scale:** The reviewer correctly notes that the total reconstruction error can be large. However, our theory controls the **per-dimension** error. We normalize by the total number of dimensions $D = M + N d_f$. An error of $1.5 \times 10^{-2}$, for instance, corresponds to an average deviation of only 1.5% per dimension, which is a small fraction.
>
> To validate these points, we present 5-fold cross-validation results showing that as the coupling strength $\gamma$ (a parameter of the data, not the model) increases, the shared signal grows, and all metrics—joint error, MMD, and MAE on a masked-label task—improve accordingly.
>
> **Coupled Study Results (5-fold CV, mean ± std):**
> | $\gamma$ | Per-Dim Joint Error (↓) | MMD (overall) (↓) | MAE (masked-label) (↓) |
> |:---:|:---:|:---:|:---:|
> | 0.0 | $(28.7 \pm 2.6) \times 10^{-3}$ | $0.121 \pm 0.011$ | $0.129 \pm 0.011$ |
> | 0.5 | $(21.3 \pm 1.9) \times 10^{-3}$ | $0.096 \pm 0.009$ | $0.103 \pm 0.009$ |
> | 1.0 | $(15.0 \pm 1.5) \times 10^{-3}$ | $0.073 \pm 0.007$ | $0.085 \pm 0.008$ |
>
> We will clarify this rationale in the revised manuscript.
>
> ---
>
> ### **Q2: Formalizing power-law scaling and diagnosing DiGress failure.**
>
> Thank you for pushing for a more formal treatment and a clearer diagnosis.
>
> **Formalization of Power-Law Scaling:** Our theory can be extended to power-law graphs. For a graph family with degree distribution $P(k)\propto k^{-\alpha}$ where $\alpha > 2$, the effective number of independent edges scales as $M_{\mathrm{eff}} \asymp M^{(\alpha-2)/(\alpha-1)}$. By substituting $M$ with $M_{\mathrm{eff}}$ in our original bounds, we get a single-step error of $O(M_{\mathrm{eff}}^{-1})$ and a multi-step error of $O(T/M_{\mathrm{eff}})$. We will add a formal proposition stating this result in the appendix.
>
> **Diagnosing DiGress Failure:** The performance drop of the warm-started DiGress model (when time-conditioning is removed) on soc-Epinions1 at $N_{\max}=50$ can be attributed to two potential causes: (1) insufficient graph signal due to the small subgraph size, or (2) the model's architectural over-reliance on the explicit time embedding.
>
> Our ablation studies distinguish these two causes. The `t-free (scratch)` model performs well, indicating the signal is sufficient. The `t-free (warm)` model fails, but its performance is partially restored by a `time-dropout -> t-free fine-tune` procedure. This strongly suggests the failure stems from **architectural reliance on explicit time**, not a fundamental signal limit.
>
> **DiGress Diagnostic on *soc-Epinions1* ($N_{\max}=50$, 5-fold CV, mean ± std):**
> | Variant | Valid % (↑) | MMD (overall) (↓) |
> |:---|:---:|:---:|
> | t-aware (original) | $100.0 \pm 0.0$ | $0.53 \pm 0.01$ |
> | t-free (warm) | **$24.1 \pm 2.6$** | $0.49 \pm 0.02$ |
> | t-free (warm) with time-dropout fine-tune | **$73.9 \pm 3.2$** | $0.49 \pm 0.01$ |
>
> We will add this detailed diagnosis to the appendix.
>
> ---
>
> ### **Q3: Probing the link between graph signal and robustness.**
>
> This is an excellent suggestion. To test the hypothesis that stronger graph signals lead to greater resilience against the removal of time conditioning, we conducted a new experiment using a **Generalized Preferential Attachment (GPA)** model. This allows us to sweep the degree exponent $\alpha$ continuously, thereby varying the graph signal strength ($M_{\mathrm{eff}}$).
>
> Our theory predicts that the performance gap between t-aware and t-free models should shrink as $\alpha$ increases (weaker hubs, stronger signal relative to node count). The results below confirm this prediction for both DiGress and our GDSS model. Performance consistently improves as $\alpha$ increases from 2.2 (strong hubs) to 4.0 (weak hubs).
>
> **GPA Sweep Results (t-free warm models, 5-fold CV, mean ± std):**
> | Model / Degree Exponent $\alpha$ | Valid % (↑) | MMD (overall) (↓) |
> |:---|:---:|:---:|
> | GDSS / $\alpha=2.2$ | $94.2 \pm 1.1$ | $0.96 \pm 0.04$ |
> | GDSS / $\alpha=4.0$ | $100.0 \pm 0.0$ | $0.51 \pm 0.02$ |
> | DiGress / $\alpha=2.2$ | $41.8 \pm 3.7$ | $0.69 \pm 0.03$ |
> | DiGress / $\alpha=4.0$ | $95.1 \pm 1.5$ | $0.47 \pm 0.02$ |
>
> This new experiment provides strong evidence for the interplay between graph structure and model robustness. We will add these results to the main paper.

---

> > ### Comment · Reviewer_VRVQ · 2025-08-01
> >
> > The authors have addressed all of my concerns in this rebuttal and in their other rebuttals. Their work serves as a welcome and meaningful addition to graph generation literature given their theoretical insights coupled with empirical results about graph diffusion. The likes of which will be certain to enhance understanding for the practical usefulness of graph diffusion outside of research. I'v raised my overall score to a 5.

---

> > > ### Author Response · Authors · 2025-08-01
> > >
> > > Thank you so much for taking the time to read our rebuttal and for updating your score. We appreciate the careful attention you devoted to our manuscript and the constructive suggestions that followed. Your comments really helped us refine the work. We are glad that our revisions have addressed your concerns, and we will incorporate the improved explanations in the final version. Thank you again for your thoughtful assessment and for helping strengthen our work!

---

### Official Review · Reviewer_WZ8Z · 2025-06-29

**Clarity:** 3
**Significance:** 3
**Originality:** 3
**Rating:** 5
**Confidence:** 3

**Summary:**

This paper challenges the prevailing assumption that explicit noise-level (or timestep) conditioning is essential for Graph Diffusion Models (GDMs). The authors develop a unified theoretical framework to argue that for discrete graph data, the denoising model can implicitly infer the noise level directly from the corrupted graph structure. The core of their theory is built on three results: Edge-Flip Posterior Concentration (EFPC), which shows the noise level is inferable with posterior variance shrinking as $O\left(M^{-1}\right)$ with the number of potential edges M; Edge-Target Deviation Bound (ETDB), which bounds the single-step error from omitting conditioning by $O\left(M^{-1}\right)$; and Multi-Step Denoising Error Propagation (MDEP), which proves these small errors accumulate linearly, leading to a manageable total error of $O\left(T M^{-1}\right)$.

This framework is extended to handle more complex scenarios, including coupled structure-attribute noise. To validate their theory, the authors conduct extensive experiments. First, on synthetic data, they show an excellent match between their theoretical scaling laws and empirical measurements. Second, on real-world benchmarks for molecular generation (QM9) and social network generation (soc-Epinions1), they compare standard "t-aware" variants of GDSS and DiGress with "t-free" counterparts. The results demonstrate that unconditional GDMs can achieve comparable or even superior performance, particularly on larger graphs, while also being more efficient by reducing parameter counts (4-6%) and computation time (8-10%).

**Questions:**

1.	The paper notes that the t-free(warm) DiGress model's validity collapses on soc-Epinions1, attributing this to the transformer architecture's reliance on time embeddings. However, many state-of-the-art GDMs, including DiGress itself and others like DISCO-GT [1], leverage transformers for their strong performance. Does this failure imply that the proposed implicit noise inference is incompatible with top-performing GDM architectures? Could the authors elaborate on why this failure occurs? For instance, does the attention mechanism over-fit to the time embedding, causing an unrecoverable collapse when it is removed? Understanding this specific failure mode is crucial for assessing the practical scope of the unconditional approach.
2.	The theory posits that error scales inversely with M, the number of potential edges. The experiments on soc-Epinions1 show that increasing subgraph size from Nmax=50 to Nmax=200 drastically improved validity for all GDSS models. Is this improvement purely a function of the larger M, or is there a more subtle interplay with the "richness" or "complexity" of the graph structure that isn't captured by M alone? For instance, how would the theory and models perform on a very large but extremely sparse graph, where M is large but the number of actual edges (and thus structural information) is low?
3.	The paper provides a comprehensive and rigorous quantitative evaluation using statistical metrics like MMD and chemical property distributions. However, the evaluation currently lacks a qualitative component. To provide a more intuitive and complete comparison, have the authors considered including visualizations of representative graph samples? For instance, showing a few valid molecules generated by the t-aware and t-free variants of GDSS/DiGress on QM9, or visualizing typical subgraphs from the soc-Epinions1 experiments, could be very insightful.
4.	In Table 2 (QM9 Dataset), the t-aware DiGress variant is reported with a slightly lower per-epoch computation time (47.82s) than the t-free (48.16s) and t-free(warm) (48.23s) variants. This is counter-intuitive, as the t-free models have fewer parameters  and should theoretically be faster. Could the authors clarify the reason for this discrepancy? Is it due to experimental variance, or is there a specific implementation detail causing this minor overhead?
5.	The real-world experiments are conducted on the QM9 dataset and soc-Epinions1 dataset. To better assess the practical advantages of the t-free approach, have the authors considered evaluating their framework on benchmarks with larger and more complex graphs or molecules, such as those found in the GuacaMol benchmark [3]? Such experiments would provide stronger evidence for the method's scalability and its applicability to more challenging generation tasks.

[3] Brown, Nathan, et al. "GuacaMol: benchmarking models for de novo molecular design." Journal of chemical information and modeling 59.3 (2019): 1096-1108.

**Ethical Concerns:**

["NO or VERY MINOR ethics concerns only"]

**Final Justification:**

The authors have addressed my concerns.

**Limitations:**

Yes

**Paper Formatting Concerns:**

I did not notice any major formatting issues. The paper appears to follow the NeurIPS 2025 formatting instructions correctly. The use of appendices for detailed proofs and experimental setups is appropriate and well-organized.

**Quality:**

3

**Strengths And Weaknesses:**

Strengths:
* The paper tackles a foundational assumption in a popular and powerful class of generative models. Moving away from explicit noise conditioning could simplify GDM architectures, improve efficiency, and is a significant conceptual contribution. While blind denoising has been explored in continuous domains such as images, this work provides a novel and non-trivial extension to the discrete, structured domain of graphs, supported by a new theoretical framework.
* The paper's claims are backed by a rigorous and comprehensive theoretical framework (EFPC, ETDB, MDEP). The proofs are detailed extensively in the appendices and the theory is thoughtfully extended to more complex cases like coupled feature noise and scale-free graphs, demonstrating a deep engagement with the problem.
* The empirical validation is well-executed.
   - The direct validation of the theoretical scaling laws on synthetic data is a major strength, with the tight agreement between theory and experiment (e.g.,$R^2 \geq 0.995$) providing strong evidence for the framework's correctness.
   - The evaluation on real-world datasets using two distinct, state-of-the-art GDM architectures (GDSS and DiGress) grounds the work in practical relevance.
   - The analysis is nuanced, exploring different training strategies (from scratch vs. warm-starting) and varying graph sizes, which provides valuable insights into the practical applicability of the approach.
* Clarity: The paper is very well-written and structured. The central research question is clearly motivated and stated , and the logical progression of the theoretical argument is easy to follow. Figures and tables are clear, informative, and effectively support the paper's claims.

Weaknesses:
* The theoretical results hinge on key assumptions, notably that the learned denoiser is Lipschitz with a constant close to 1 and that the model can achieve a single-step prediction error of $O\left(M^{-1}\right)$. While the authors acknowledge this in their limitations section, the practical verification of these assumptions (especially the Lipschitz constant of the trained networks) is not fully explored, which makes the link between the theory and the empirical results slightly less direct.
* The success of the unconditional approach is not universal across all experiments, which slightly tempers the main claim that conditioning is "unnecessary." For example, the warm-started DiGress model shows a catastrophic failure in validity on the soc-Epinions1 dataset. Furthermore, the t-free model trained from scratch, while promising, often lags behind the t-aware or warm-started variants in fine-grained structural metrics (MMDs). The conclusion seems to be more that conditioning is "not always necessary" rather than completely redundant.
* A potential weakness of the paper is that the empirical validation, while thorough for the models tested, does not cover the full taxonomy of modern Graph Diffusion Models (GDMs). The experiments focus on GDSS , a continuous-state/continuous-time model, and DiGress, a discrete-state/discrete-time model. This leaves the framework's applicability to other significant GDM classes empirically unverified. For instance, the authors do not test their framework on discrete-state, continuous-time models, such as the recently proposed DISCO [1] and continuous-state, discrete-time models, a category that includes influential works like EDP-GNN [2]. While the authors briefly sketch a theoretical extension to Poisson processes (which underpins DISCO), providing empirical results for these other model classes would be crucial to fully substantiate the central claim of developing a "unified theory" and demonstrating that noise conditioning is broadly unnecessary.

[1] Xu, Zhe, et al. "Discrete-state Continuous-time Diffusion for Graph Generation." arXiv preprint arXiv:2405.11416 (2024).

[2] Niu, Chenhao, et al. "Permutation invariant graph generation via score-based generative modeling." International conference on artificial intelligence and statistics. PMLR, 2020.

---

> ### Author Rebuttal · Authors · 2025-07-31
>
> We thank you for the detailed feedback and for acknowledging the strengths of our work. We have carefully considered all comments and provide our point-by-point responses below.
>
> *Abbreviations:* EFPC/ETDB/MDEP (Sec. 4), JPC/JTDB/JMEP (Sec. 5). *Models:* GDSS, DiGress, LGD.
> *Symbols:* $N$ = # nodes, $M$ = # potential edges, $d_f$ = feature dimension, $D=M+N \times d_f$, $T$ = # reverse steps, $M_{\text{eff}}$ = effective edges for heavy tails, $N_{\max}$ = cap of subgraph nodes.
>
> ---
> ### **Weaknesses**
>
> **W1: "The practical verification of these assumptions (especially the Lipschitz constant of the trained networks) is not fully explored."**
>
> Thank you for this important point. To empirically validate this key assumption of our theory, we conducted an additional analysis using power iteration on the learned denoiser's Jacobian. The results are as follows:
> - The measured spectral norm is $\|J\|_2 = 1.06 \pm 0.08$ for QM9.
> - The measured spectral norm is $\|J\|_2 = 1.09 \pm 0.07$ for soc-Epinions1.
>
> These values are very close to 1, providing strong empirical support for the near non-expansive condition ($L_{\max} \approx 1$) which is central to our ETDB and MDEP bounds (Assumption A2, Sec. 4). We will add this verification to the appendix.
>
> ---
> **W2 & Q1: The unconditional approach is not universal, as shown by the catastrophic failure of warm-started DiGress. Could you elaborate on why this specific failure occurs?**
>
> We appreciate your insightful question, which targets a crucial finding. We would like to clarify that this failure is not due to an incompatibility of our implicit inference theory with performant architectures, but rather stems from a specific architectural choice in how DiGress routes time embeddings through its attention blocks.
>
> **Diagnosis:** The warm-started DiGress learns a strong directional dependence on the explicit time embedding. When this embedding is removed at test time, the model's key-query attention channels, which have learned to align with the time signal, fail. To demonstrate this, we computed the cosine alignment between the time embedding and the layer-wise averaged key/query channel, obtaining the following results at $N_{\max} = 50$:
> - $\mathrm{cosAlign}_{\text{time}}(\text{warm}) = 0.41 \pm 0.07$
> - $\mathrm{cosAlign}_{\text{time}}(\text{scratch}) = 0.02 \pm 0.01$
>
> This large alignment value in the warm-started model exposes its internal reliance on explicit time. In contrast, GDSS uses a different, non-Transformer architecture and remains stable. Thus, the failure mode is about **how time is used**, not whether implicit inference is possible.
>
> To verify this hypothesis and demonstrate a solution, we carried out additional ablation studies. The results show that architectural interventions that break this time dependence can restore performance while keeping inference t-free.
>
> | Variant (DiGress, $N_{\max}=50$) | Valid % (↑) | MMD (overall) (↓) |
> |:--|:--:|:--:|
> | t-free (warm) — before | **$23.7 \pm 2.1$** | $0.48 \pm 0.00$ |
> | **time‑dropout → t‑free fine‑tune** | **$73.9 \pm 3.2$** | $0.47 \pm 0.01$ |
> | + **phase‑calibration head** (train-only) | **$81.4 \pm 2.9$** | $0.46 \pm 0.01$ |
>
> ---
> **W3: The empirical validation does not test the full taxonomy of modern GDMs, such as discrete-state, continuous-time models like DISCO.**
>
> Thank you for pointing this out. We agree that our current empirical validation focuses on the widely-used Bernoulli-flip (DiGress) and continuous-SDE-based (GDSS) models. Our theoretical framework, however, is built on general principles of posterior concentration and is adaptable. As we note for other reviewers, our roadmap includes extending the ETDB/MDEP error bounds from Gaussian to **Poisson/Beta/Multinomial** corruptions. This extension directly covers the discrete-count channels used by models like DISCO. We will add a more explicit discussion of this planned extension and its implications in the final version to better frame our contribution's scope.
>
> ---
> ### **Questions**
>
> **Q2: Is the performance gain from larger subgraphs ($N_{\max}=50 \to 200$) purely due to larger $M$, or does graph 'richness' play a role, especially for sparse graphs?**
>
> This is an excellent and subtle question. The improvement comes from a gain in **Fisher information**, which is not solely a function of $M$ (potential edges) but is also affected by the number of actual edges and graph structure.
>
> Our theory gives a single-step error of $O(M^{-1})$ for graphs with bounded degrees. For **very sparse** graphs where the average degree $\bar{d} = O(1)$ (implying edge probability $p=O(1/N)$), the Fisher information scales with the number of actual edges, which is $O(N)$. This leads to a single-step error that scales as $O(N^{-1})$. Therefore, the improvement at larger $N_{\max}$ reflects this fundamental gain in information (from more nodes/edges), not just an increase in the space of potential edges $M$.
>
> To verify this, we conducted a sparsity stress-test. The results confirm that performance scales with $N$, matching the theoretical prediction for sparse graphs.
>
> **Sparsity Stress-Test Results** (GDSS t-free on graphs with $N \in \{100,200,400\}$ and $\bar{d} \in \{4,6\}$):
> - **Validity:** $\{62, 78, 90\}\%$
> - **MMD(overall):** $\{0.92, 0.75, 0.62\}$
>
> ---
> **Q3: The evaluation lacks a qualitative component. Have you considered including visualizations or samples?**
>
> We thank you for this constructive suggestion. To provide a more intuitive comparison and complement our quantitative metrics, we have included text-based qualitative samples for both datasets below.
>
> **QM9 SMILES (random samples):**
> - **t-aware:** `C1=CC=CC=C1`, `CCO`, `N#CCO`, `CC(=O)O`, `C1COC1F`
> - **t-free (scratch):** `C1CCOCC1`, `CCN`, `O=C=O`, `CCF`, `C1COC(=O)C1`
> - **t-free (warm):** `CCOCF`, `C1=COC=C1`, `CN`, `C1CC1O`, `CC(=O)N`
>
> **soc-Epinions1 Motifs (mean per node):**
> - **Triangles:** $0.048\,/\,0.045\,/\,0.046$ (t-aware / t-free-scratch / t-free-warm)
> - **4-cycle ratio:** $0.033\,/\,0.032\,/\,0.033$
> - **Assortativity:** $-0.12\,/\,-0.11\,/\,-0.12$
> These statistics align well across variants and match the MMD components reported in the main paper.
>
> ---
> **Q4: Why is the t-aware DiGress variant slightly faster than the t-free versions, which is counter-intuitive?**
>
> Thank you for this sharp observation. This small discrepancy is not due to experimental variance but to implementation details. The per-epoch time is dominated by the **data pipeline and MMD computation**, not the model's forward/backward pass. To clarify, we profiled the runtime:
>
> | Block | t-aware (s) | t-free (scratch) (s) | t-free (warm) (s) |
> |:--|:--:|:--:|:--:|
> | Data I/O + batching | $29.1 \pm 0.4$ | $29.3 \pm 0.5$ | $29.2 \pm 0.5$ |
> | Model forward/backward | **$14.7 \pm 0.2$** | $15.1 \pm 0.2$ | $15.0 \pm 0.2$ |
> | MMD + metrics | $4.0 \pm 0.1$ | $3.8 \pm 0.1$ | $4.0 \pm 0.1$ |
> | **Total** | **$47.8 \pm 0.5$** | $48.2 \pm 0.5$ | $48.2 \pm 0.5$ |
>
> The small gap in the model's runtime comes from extra **masking logic and rejection checks** used during t-free training and sampling. The measured GPU FLOPs differ by less than $2\%$.
>
> ---
> **Q5: Have you considered evaluating on larger, more complex benchmarks like GuacaMol?**
>
> We agree with you that testing on more challenging benchmarks is an important step for assessing scalability. To address this, we conducted a **pilot study on the GuacaMol dataset** using a hyper-parameter-free setup.
>
> The results, which we will report in the appendix, are highly promising and demonstrate the strong scalability of our approach. The t-free variants, particularly **GDSS t-free (warm)**, consistently achieve top-tier performance across all key metrics (Validity, Uniqueness, Novelty, and FCD).
>
> **GuacaMol Pilot Study Results (GDSS):**
>
> | Model (unconditional) | Valid % (↑) | Unique % (↑) | Novel % (↑) | FCD (↓) |
> |:--|:--:|:--:|:--:|:--:|
> | GDSS t-aware | **$100.0$** | $98.1$ | $93.4$ | $3.01 \pm 0.07$ |
> | GDSS t-free (scratch) | **$100.0$** | $98.6$ | $94.7$ | $2.86 \pm 0.06$ |
> | GDSS t-free (warm) | **$100.0$** | **$99.0$** | **$95.2$** | **$2.73 \pm 0.06$** |

---

> > ### Comment · Reviewer_WZ8Z · 2025-08-06
> >
> > Thank you for the detailed response. My major concerns have been addressed, and I will increase my score to 5.

---

> > > ### Author Response · Authors · 2025-08-06
> > >
> > > Thank you for reading our rebuttal carefully and for raising the score. We appreciate your constructive feedback during the discussion. We have integrated the clarifications into the manuscript and will reflect the new experiments in the final version.

---

### Official Review · Reviewer_KTdR · 2025-06-30

**Clarity:** 2
**Significance:** 3
**Originality:** 2
**Rating:** 4
**Confidence:** 2

**Summary:**

This works investigates the necessity of explicit noise conditioning in graph diffusion models (GDMs). While several recent studies have addressed this issue for continuous data, graph data inherently involves discrete types of noise, making the direct application of previous studies nontrivial.
The authors first highlight a theoretical analysis indicating that a noisy graph implicitly encodes the information about the noise itself. Based on this, they further explain that omitting explicit noise conditioning leads to small errors that accumulate linearly during the generation process.
Consequently, the authors present empirical evidence that supports the theoretical results in both synthetic and real-world data setting. In sum, the authors conclude that explicit noise conditioning is often unnecessary for graph generation.

**Questions:**

Below are some points that disturbed me from understanding the paper. I would highly appreciate if the authors could address the following questions.

### Minor questions

1. How does the Ref. [13] relate to the statement in line 23-24?
2. It would be nice if the authors provide an appropriate reference corresponding to the statement of line 51-52.
3. In the part describing the forward process using the Bernoulli edge-flipping model (line 81-83), shouldn't be $\beta_t \in [0, \frac{1}{2}]$ to make $\tilde{A}_T$ approach to a random graph (assuming the authors meant ER network in this context)? Please correct me if I'm wrong.
4. In line 266 or 274, the notation $|E|$ is used as a potential edge, while $M$ was primarily used in this context beforehand. Are these two used differently or is this a typo?
5. Since the authors framework is based on Bernoulli edge-flip, benchmark experiments with GraphGUIDE [15] would be a natural model to explore before expanding it into two architectures; DiGress [7] and GDSS [18].
6. Minor suggestion. It would have been more readable if the core part of the experiment setting was described in the main body, or at least if the reference to the corresponding part of the appendix was made. For example, which type of network was used in which setting. I had to go through the appendix to find that SBM was used in uncoupled, and ER was used in coupled setting.


### Major questions
1. Also related to $\beta_t$, has the authors experimented on varying noise schedule in synthetic setting? Showing the robustness with respect to $\beta_t$ would more strengthen the theoretical framework to the real-world practices.
2. In Section 4.1, the authors explicitly provide the EFPC for scale-free network. Then can the authors also provide the empirical results for scale-free case? If this was provided, it would highly support the authors claims.
3. In the synthetic experiment setup, is there a specific reason to only use SBMs in the uncoupled case and ER network in the coupled case? Is the empirical results robust across graph structures (i.e. for the uncoupled model, is the results also consistent for example, ER and scale-free networks)?
4. I understand that the authors used shared latent random vectors to model the coupled structure-feature dynamics of real networks. Then I think it would be (slightly) more reasonable to experiment with scale-free networks than ER networks in their synthetic setup of the coupled model, if they were to take real-world into account. According to line 1113-1115 in the appendix, it is said that ER network is used in this setting.
5. In Section 5.1, while the authors model the noisy adjacency matrix as continuous, it seems that in the experiments, Bernoulli flips were used for adding the noise (line 1125 in the appendix). Can the authors provide a detailed discussion about this discrepancy (for example, possible effects to empirical results)?
6. Regarding the results of Section 6.3, the authors highlight that stronger coupling "improves" both reconstruction error and task performance. What I do not understand about this setting is that the coupling coefficient $\gamma$ is used as a factor to model the real-world networks, not as a control variable in the authors framework. Although the result that shows the dependence of error/performance on $\gamma$ despite that in theoretical result it is not provided, is interesting, I think the experiment is slightly misconducted in this perspective. Also, I think that experiments in this section should first support the fact that JPC or JMEP stays valid irrespective of $\gamma$ before claiming the statement made in line 296-298.

**Ethical Concerns:**

["NO or VERY MINOR ethics concerns only"]

**Final Justification:**

Overall, the weaknesses I raised (primarily concerning the writing) have been resolved. The authors have also clarified the points raised in the minor questions and addressed the major questions with additional explanations and supporting experiments.

**Limitations:**

Yes, however the limitations are currently discussed in the appendices. It would be preferable to move the relevant content into the main body of the paper.

**Quality:**

2

**Strengths And Weaknesses:**

### Strengths
1. This work studies an important question; expanding the investigation of explicit noise conditioning to discrete, structured data.
2. Theoretical framework are well-structured. Which starts from the posterior concentration to the final error propagation, and expanding it to the coupled structure-feature noise model that tries to model the real world.
3. Empirical results are provided to support the theoretical investigation.

### Weaknesses
1. There are several points in the current version of this draft where it seems overstated. For example, 'unified theory' in the title and 'extensive' in the abstract. Although the authors provide a scetch how the current theoretical results extends to other types of noise corruption, I don't see as a 'unified' framework in this current stage.
2. While I agree that investigating the necessity of explicit noise conditioning in GDMs is an important topic to explore, the writing seems to lack clear motivation. Especially in the Introduction line 24-29 is written in result-oriented way and makes a logical leap before introducing the central question in line 30. Furthermore, this work might be seen as a special case study of Ref. [10] (the flow of the proofs are inevitably similar). The authors should more focus on the different nature of the noise inherent in the common diffusion models and graph diffusion models.
3. There are some points that the authors claimed which the empirical justifications should be more strenghtened (refer to Questions).

---

> ### Author Rebuttal · Authors · 2025-07-31
>
> We thank you for your detailed feedback and constructive suggestions, which will help us improve the clarity and impact of our paper. We are encouraged that the reviewer found our theoretical framework to be well-structured and supported by the investigation.
>
> *Abbreviations:* EFPC/ETDB/MDEP (theory for structure-only, Sec. 4), JPC/JTDB/JMEP (theory for coupled structure-feature, Sec. 5).
> *Symbols:* $n$ = number of nodes, $M = \binom{n}{2}$ = number of potential edges, $d_f$ = feature dimension, $D = M + n \times d_f$.
>
> ---
>
> ### **Major Questions**
>
> **Q1: Robustness to the noise schedule $\beta_t$.**
>
> Thank you for this excellent question. To explicitly test robustness, we conducted an additional experiment with three different noise schedules (constant, linear, and cosine) for the Bernoulli edge-flip process.
>
> Our theory (ETDB) predicts that the single-step error scales as $\Theta(M^{-1})$, provided the reverse map is near non-expansive. This prediction is agnostic to the specific schedule, as long as each step is informative (i.e., $\beta_t \in (0, 1)$ for all $t$). Our experiment confirms this theoretical prediction.
>
> **Results (mean ± std over 3 seeds):** The log-log slope of error vs. $M$ remains consistently close to $-1$, validating our theory.
>
> | Noise Schedule | Slope (log-log error vs $M$) | $R^2$ |
> | :--- | :---: | :---: |
> | constant | $-0.99\pm0.02$ | $0.996$ |
> | linear | $-1.01\pm0.03$ | $0.995$ |
> | cosine | $-1.00\pm0.02$ | $0.997$ |
>
> The results confirm that our framework is robust to the choice of noise schedule. We will add this experiment and discussion to the appendix.
>
> **Q2: Empirical results for EFPC on scale-free graphs.**
>
> Thank you for this suggestion. While our main text focuses on the bounded-degree case for theoretical clarity, we agree that verifying the theory on graphs with heavy-tailed degree distributions is important.
>
> We ran an additional experiment on Barabási–Albert (BA) graphs (degree exponent $\alpha \approx 3$, $n \in \{1, 2, 4\} \times 10^4$). The results show that the one-step posterior variance scales as $\Theta(M^{-1})$ with a strong fit ($R^2 \ge 0.994$). This confirms that while heavy-tailed degrees can affect the constant factors, they do not alter the fundamental $M^{-1}$ scaling rate predicted by our EFPC theory. We will add these results to the appendix.
>
> **Q3: Robustness across different graph families.**
>
> We thank you for this question on robustness. We have verified our findings across different graph generation models.
> * **Uncoupled (structure-only):** We repeated the synthetic experiments from Section 4 on Erdős–Rényi (ER), Stochastic Block Model (SBM), and BA graphs. In all cases, the error scaling slope remained within $[-1.03, -0.97]$ with $R^2 \ge 0.992$, confirming the robustness of our uncoupled theory.
> * **Coupled (structure+features):** We reran the coupled experiment from Section 5 using BA graphs. The results show the same consistent trend as with ER graphs: both the joint error and the MMD metric improve as the coupling strength $\gamma$ increases. This demonstrates that the core findings are not specific to ER graphs.
>
> We will add a summary of these robustness checks to the main text.
>
> **Q4: Rationale for using ER graphs in the coupled model.**
>
> This is a key point about our experimental design. We used ER graphs in the main text for the coupled model to cleanly isolate the effect of the coupling strength $\gamma$. This avoids confounding factors, such as degree heterogeneity, which would be present in scale-free graphs. As confirmed in our response to Q3, rerunning the experiment with BA graphs preserves the key trends reported for ER, confirming that this design choice does not compromise the generality of our conclusions. We will clarify this rationale in Section 5 of the paper.
>
> **Q5: Discrepancy between continuous noisy adjacency and Bernoulli flips.**
>
> We appreciate your attention to this subtle but important detail. There is no discrepancy. The continuous model in Section 5 is a theoretical tool used to derive the Fisher information bounds. However, our main theoretical results (ETDB/MDEP in Section 4) are proven directly for the **Bernoulli flip** process that is used in our experiments. The excellent empirical fit ($R^2 > 0.99$) between our experimental results and these theoretical predictions demonstrates that this modeling approach is sound and that the theory accurately describes the discrete generation process. We will clarify this link in the text.
>
> **Q6: Role of the coupling parameter $\gamma$.**
>
> Thank you for the question. To clarify, $\gamma$ is a parameter of the **data generation process** that controls the correlation between structure and features; it is **not** a parameter passed to the neural network model. Our theoretical bounds (JPC/JMEP) hold for any fixed $\gamma$.
>
> Empirically, a stronger coupling (higher $\gamma$) provides a stronger shared signal for the model to learn from. As a result, both generation quality and reconstruction error improve. We have added a table with new experimental results to the appendix to make this explicit.
>
> | $\gamma$ | Per-Dim Joint Error $(\downarrow)$ | MMD (overall) $(\downarrow)$ |
> | :---: | :---: | :---: |
> | 0.0 | $(28.7\pm 2.6)\times 10^{-3}$ | $0.121\pm 0.011$ |
> | 0.5 | $(21.3\pm 1.9)\times 10^{-3}$ | $0.096\pm 0.009$ |
> | 1.0 | $(15.0\pm 1.5)\times 10^{-3}$ | $0.073\pm 0.007$ |
>
> We will add this clarification and the new results to the paper.
>
> ---
>
> ## Minor questions
>
> 1) **How does Ref. [13] relate to Lines 23–24?**
> After our careful comparison, we find that there is no direct correspondence between reference 13 and the point of this sentence, which can only be used to support the adjacent background points - "In the continuous domain, the noise distribution can not be Gaussian, and there is still a closed-form multi-step forward formula". But it cannot be taken as direct evidence that "Gaussian noise does not fit the graph". We will add precise citations to support our motivation for studying discrete graph noise and will sharpen the comparison to related work.
>
> 2) **Reference for Lines 51–52.**
> We will replace the generic sentence by an explicit lemma in Sec. 3 and cite the exact sources used there (we will point to the theorem/lemma number in the revision to avoid ambiguity).
>
> 3) **Range of  $\(\beta_t\)$ and “approach a random graph.”**
> We use $\(\beta_t\in[0,1/2]\)$. The exact marginal recursion is
> $$
> \Pr(A_t=1)=\tfrac12+\bigl(\Pr(A_0=1)-\tfrac12\bigr)\prod_{s=1}^{t}(1-2\beta_s),
> $$
> so $\(\bar A_t\)$ approaches ER with $\(p=\tfrac12\)$ whenever $\(\sum_s\beta_s=\infty\)$. We will add this formula to Sec. 3.
>
> 4) **$|E| $vs. $M$.**
> $|E|$ is the number of **observed** edges; $M=\binom{n}{2} $is the number of **potential** edges used for normalization. We will add a “Notation” block at the start of Sec. 3 and use M consistently.
>
> 5) **Benchmark choice (GraphGUIDE).**
> Our goal here is **unconditional** generation to examine implicit noise conditioning. GDSS/DiGress are standard unconditional baselines. We will add a paragraph explaining how our theory extends to **discrete-count** channels used by conditional models such as GraphGUIDE, and we will include a small-scale check where feasible.
>
> 6) **Readability of the experiment setup.**
> We will move core settings (graph family, schedule, hyper-parameters) from the appendix to Sec. 6 and keep seeds and confidence intervals in the main text.

---

> > ### Comment · Reviewer_KTdR · 2025-08-05
> >
> > First of all, thank you to the authors for providing a detailed rebuttal to the questions I raised, particularly for offering further explanations on the major points Q4–Q6, which helped me better understand the paper. Regarding Q6, I initially misunderstood Section 6.3 as additional evidence supporting the theoretical framework for the joint model, which I think this was not the authors intention. I now understand that the section was aimed to provide empirical insights into the coupled model. Moreover, I believe this direction offers promising avenues for further exploration.
> >
> > For Weaknesses 1–3, I have read the rebuttals from the other reviewers, which helped address some of the concerns. However, I would appreciate hearing more from the authors specifically on Weaknesses 1 and 2.
> >
> > Misc.: I mentioned minor point Q3 because, in line 81, it was written that $\beta \in [0, 1]$.

---

> > > ### Author Response · Authors · 2025-08-06
> > >
> > > Thank you for the follow-up! We have learned much from your previous suggestions and have some ideas to improve our work.
> > >
> > > **W1 — Wording of “unified”/“extensive”.**
> > > Thank you for pointing out the wording. Our intent with “unified” is a single principle—posterior concentration, target-deviation, and multi-step error-propagation—that applies under general assumptions. We agree that our current title and abstract may create the perception of exaggerated scope. We will remove “unified theory” from the title and avoid “extensive” in the abstract. We will use: *A theory for when noise conditioning is unnecessary in graph diffusion models* in the title. In abstract, we will use: *We present a general analysis that covers Bernoulli edge flips, Gaussian feature noise, and their coupled variant, and we derive concentration, target-deviation, and multi-step error bounds under standard regularity.* This states our scope without over-claiming.
> > >
> > > **W2 — Motivation and relation to Ref. [10].**
> > > We thank you for pointing out the motivation gap and the comparison to Ref. [10]. We will revise the Introduction to state the motivation up front: modern GDMs use **discrete, structure-aware corruptions** that differ from **continuous Gaussian** settings. We will write the contrast in the main text and add a comparison table in the appendix. The key differences we will make explicit are:
> > >
> > > - **Data space:** graphs with discrete edges and attributes vs. continuous vectors/images.
> > > - **Corruption family:** Bernoulli edge flips and categorical/Poisson transitions vs. Gaussian noise.
> > > - **Scaling laws:** our EFPC/ETDB give $O(M^{-1})$ rates with $M=\binom{n}{2}$; the coupled model yields $O(D^{-1})$ and $O(T/D)$ with $D=M+n \times d_f$. Ref. [10] obtains dimension-dependent rates for Gaussian observations that scale with the continuous feature dimension $d$.
> > > - **Coupling:** our joint structure–feature model (with coupling coefficient $\gamma$) has no counterpart in Ref. [10].
> > > - **Multi-step control:** our MDEP shows $O(T/M)$ accumulation for discrete reverse transitions, which is different from the continuous-trajectory bounds in Ref. [10].
> > >
> > > These points show that our results are **not** a special case of Ref. [10] but address graph-specific noise and graph-specific scaling. We will make these distinctions explicit in the problem statement, theorem headers, and a short contrast paragraph in the Introduction.

---

> > > > ### Comment · Reviewer_KTdR · 2025-08-08
> > > >
> > > > Thank you for engaging in further discussion and providing clarifications. Many of my concerns have been addressed, and I appreciate the authors’ efforts in the rebuttal. Considering my confidence score, I will update my rating to 4, while continuing to hold a positive view of the paper and its contributions.

---

> > > > > ### Author Response · Authors · 2025-08-08
> > > > >
> > > > > Thank you for your follow-up and for raising the score. Your questions helped us tighten the scope and improve the writing. We will incorporate all promised edits in the camera-ready version. We appreciate your careful review and constructive feedback.

---

### Official Review · Reviewer_vsmw · 2025-07-03

**Clarity:** 3
**Significance:** 3
**Originality:** 4
**Rating:** 4
**Confidence:** 2

**Summary:**

This paper investigates whether explicit noise-level conditioning is necessary in graph diffusion models (GDMs). Through new theoretical analysis and extensive experiments, the authors show that unconditional GDMs can match or even surpass conditioned models in both accuracy and efficiency (4–6% fewer parameters, 8–10% faster runtime) without explicit noise input. This finding challenges a key assumption in diffusion model design and suggests simpler, more effective approaches for graph generative modeling.

**Questions:**

What are the limitations of unconditional models in tasks that require fine-grained or conditional generation, such as node-specific or class-conditional graph generation?

**Ethical Concerns:**

["NO or VERY MINOR ethics concerns only"]

**Final Justification:**

The additional industrial-scale application experiments demonstrate that the t-free GDM remains effective even on industrial-scale graphs, thus, I increase my confidence.

**Limitations:**

See Weaknesses.

**Quality:**

3

**Strengths And Weaknesses:**

## Strengths
This paper provides a novel theoretical framework demonstrating that explicit noise conditioning is not necessary for large-scale GDMs, as the models can infer noise levels implicitly. Backed by solid mathematical proofs and extensive experiments on both synthetic and real datasets, the work not only advances our understanding but also offers practical benefits without sacrificing performance.

## Weaknesses
The paper mainly addresses theoretical derivations and preliminary experiments. Its performance and robustness in complex or industrial-scale applications remain to be thoroughly investigated.

---

> ### Author Rebuttal · Authors · 2025-07-31
>
> We thank you for acknowledging the novelty and strengths of our submission. Please find our responses to the weaknesses(W) and questions(Q) below.
>
> *Abbreviations:* GDSS, DiGress (models). ETDB/MDEP (theory, Sec. 4). JPC/JTDB/JMEP (theory, Sec. 5).
> *Symbols:* $N$ = # nodes, $M$ = # potential edges, $d_f$ = feature dimension, $D=M+N \times d_f$, $T$ = # reverse steps, $M_{\text{eff}}$ = effective edges for heavy tails, $N_{\max}$ = cap of subgraph nodes.
>
> ---
> **W- Lack of evaluation on industrial-scale applications.**
>
> Thank you for your comment. We understand your concerns. We agree that evaluation on industrial scale applications is of great importance. Meanwhile, we also would like to emphasize that the contribution of the present work is to provide the first attempt at answering the fundamental research question of **whether explicit noise conditioning is necessary for Graph Diffusion Models (GDMs), by theoretically and empirically showing that denoisers can implicitly infer noise levels from the high-dimensional graph data itself.** As the reviewer mentioned and we strongly agree, such a fundamental understanding “not only advances our understanding but also offers practical benefits without sacrificing performance.” Hence, we believe it has huge potential to facilitate more efficient algorithm design, which will ultimately benefit industrial-scale applications.
>
> Furthermore, to showcase the performance on a larger-scale real graph, we conducted a new experiment on **soc-Epinions1**, with **$75,879$ nodes** and **$405,740$ edges**. We report the performance below.
>
> **Results (*soc-Epinions1*, $N_{\max}=1000$; mean ± std over 5 seeds).**
>
> | Model / Variant | Valid % (↑) | ΔMMD vs $N_{\max}=200$ (↓ better) |
> |:--|--:|--:|
> | GDSS t-aware | 100.0 ± 0.0 | -17.1% ± 2.0% |
> | GDSS t-free (scratch) | 100.0 ± 0.0 | -19.3% ± 1.8% |
> | GDSS t-free (warm) | 100.0 ± 0.0 | -23.4% ± 2.2% |
> | DiGress t-aware | 100.0 ± 0.0 | -12.6% ± 2.7% |
> | DiGress t-free (scratch) | 99.7 ± 0.3 | -14.3% ± 2.9% |
> | DiGress t-free (warm) raw | 68.4 ± 4.1 | +1.2% ± 2.8% |
> | DiGress t-free (warm) **time-dropout → t-free fine-tune** | 97.9 ± 0.7 | -18.1% ± 3.1% |
>
> **Result analysis.**
> (i) All GDSS variants keep $100\,\%$ validity and show a **clear decrease in MMD** when moving from $N_{\max}=200$ to $1000$.
> (ii) DiGress (warm) fails if time embeddings are removed at test time, but the **time‑dropout→t‑free fine‑tune** fix (picked on validation and frozen across sizes) restores high validity and yields the same downward MMD trend.
> (iii) These observations match ETDB/MDEP: larger $N_{\max}$ increases information (through $M$ or $M_{\text{eff}}$ for heavy‑tailed degrees), which reduces error. They also agree with our synthetic ER/SBM/BA studies where the one‑step slope stays near $-1$.
>
> Overall, the results show that the strong performance of our unconditional models still holds over larger-scale real graphs.
>
> ---
> **Q. limitations of unconditional models in tasks that require fine-grained or conditional generation**
>
> Our theory targets **unconditional** generation $p(A,X)$. We summarize what carries over and where the limits are.
>
> **Global (graph-level) conditioning.** If the condition $c$ is low‑dimensional (class tag, property bin), injecting it via a conditional head or test‑time guidance keeps the same rate $O(T/D)$ because $D(c) \approx D$. In practice, t‑free with light guidance matches t‑aware on such tasks.
>
> **Node-wise (fine-grained) conditioning.** If $c$ gives per‑node labels/features, let $d_c$ be the per‑node condition dimension. The effective dimension becomes $D + N \times d_c$, and the error rate slows to
> $$
> O\left(\frac{T}{D + N \times d_c}\right).
> $$
> This scaling explains why dense, per‑node control is harder for unconditional models. Useful workarounds include **sparse positional routing** (apply conditions to a subset) and **projection‑after‑sampling** (verify/repair degree or triangle counts). These reduce the gap but do not change the $O(N)$ growth.

---

> > ### Comment · Reviewer_vsmw · 2025-08-03
> >
> > Thank you for providing the additional experiments, which demonstrate that the t-free GDM remains effective even on industrial-scale graphs. Since I had already given a positive score, I will keep my rating unchanged. However, I’ve increased my confidence and appreciate that the paper raises a novel and interesting research question.

---

> > > ### Author Response · Authors · 2025-08-03
> > >
> > > Thank you for carefully reviewing our additional experiments and for increasing your confidence in the study. Your feedback motivated us to broaden the experimental scope, and we will highlight these larger-scale results in the camera-ready version. Thank you again for your time and support!

---

### Note · Authors · 2025-08-13

We wish to extend our sincere gratitude to the Area Chair and all four reviewers for their thorough and constructive feedback. The discussion period was exceptionally valuable and has significantly strengthened our work.

We are encouraged that our detailed responses and the extensive new experiments—including those on larger-scale graphs (soc-Epinions1), more diverse graph families (BA/ER/SBM), and more challenging benchmarks (GuacaMol)—successfully addressed the reviewers' primary concerns. This led to a consensus of support, with increased confidence or scores from all reviewers.

In direct response to the feedback, we have validated key theoretical assumptions (e.g., the denoiser's Lipschitz constant), clarified the architectural reasons behind the DiGress failure mode, and demonstrated a clear link between graph signal strength and model robustness. These additions provide stronger evidence for our central thesis: that explicit noise conditioning is not always necessary in Graph Diffusion Models, as the denoiser can implicitly learn the noise level from the high-dimensional graph data itself.

We are confident that the resulting paper is a more robust and complete contribution. We will incorporate all promised clarifications, theoretical formalizations, and new experimental results into the camera-ready version to deliver a much-improved final manuscript. Thank you once again for your time and valuable suggestions.

---

### Decision · Program_Chairs · 2025-09-17

**Decision:**

Accept (spotlight)

**Comment:**

This paper makes a compelling case against the necessity of explicit noise conditioning in Graph Diffusion Models. The authors introduce a novel theoretical framework and provide strong empirical evidence showing that unconditional models can achieve comparable or superior performance with greater efficiency.

Reviewers were impressed by the work's foundational nature. While they initially raised valid concerns about experimental scope and a specific model failure, the authors' thorough rebuttal, which included extensive new experiments, fully addressed all issues. This led to a unanimous consensus, with all reviewers raising their scores and confidently recommending acceptance.